# FLOW MATCHING WITH SEMIDISCRETE COUPLINGS

**Alireza Mousavi-Hosseini**[*]
Apple, U. Toronto

**Stephen Zhang**[*]
Apple, U. Melbourne

**Michal Klein**
Apple

**Marco Cuturi**
Apple

## ABSTRACT

Flow models parameterized as time-dependent velocity fields can generate data from noise by integrating an ODE. These models are often trained using flow matching, *i.e.* by sampling random pairs of noise and target points $(\mathbf{x}_0, \mathbf{x}_1)$ and ensuring that the velocity field is aligned, on average, with $\mathbf{x}_1 - \mathbf{x}_0$ when evaluated along a time-indexed segment linking $\mathbf{x}_0$ to $\mathbf{x}_1$. While these noise/data pairs are sampled independently by default, they can also be selected more carefully by matching batches of $n$ noise to $n$ target points using an optimal transport (OT) solver. Although promising in theory, the OT flow matching (OT-FM) approach (Pooladian et al., 2023; Tong et al., 2024) is not widely used in practice. Zhang et al. (2025) pointed out recently that OT-FM truly starts paying off when the batch size $n$ grows significantly, which only a multi-GPU implementation of the Sinkhorn algorithm can handle. Unfortunately, the pre-compute costs of running Sinkhorn can balloon, requiring $O(n^2/\varepsilon^2)$ operations for every $n$ pairs used to fit the velocity field, where $\varepsilon$ is a regularization parameter that should be typically small to yield better results. To fulfill the theoretical promises of OT-FM, we propose to move away from batch-OT and rely instead on a *semidiscrete* formulation that can leverage the fact that the target dataset is usually of finite size $N$. The SD-OT problem is solved by estimating a *dual potential* vector of size $N$ using SGD; using that vector, freshly sampled noise vectors at train time can then be matched with data points at the cost of a maximum inner product search (MIPS) over the dataset with a cost $O(N)$. Semidiscrete FM (SD-FM) removes the quadratic dependency on $n/\varepsilon$ that bottlenecks OT-FM. SD-FM beats both FM and OT-FM on all training metrics and inference budget constraints, across multiple datasets, on unconditional/conditional generation, or when using mean-flow models.

## 1 INTRODUCTION

Flow-based generative models (Rezende & Mohamed, 2015; Dinh et al., 2016; Kingma & Dhariwal, 2018; Grathwohl et al., 2018) can gradually transform noise vectors into structured data. Flow models parameterized as time-dependent velocity fields (Chen et al., 2018) $\boldsymbol{v}_\theta(t, \mathbf{x})$ can be efficiently trained using flow matching (FM) (Lipman et al., 2023; Peluchetti, 2022; Albergo et al., 2023) to yield state-of-the-art performance (Zheng et al., 2024; Esser et al., 2024). The FM training approach proposes to sidestep the need to differentiate through a costly numerical ODE integration at training time by instead minimizing velocity fields locally through a regression loss. In a nutshell, the loss is formed by sampling a pair of noise/data points $(\mathbf{x}_0, \mathbf{x}_1)$ and a random time $t \in [0, 1]$ to evaluate $\|\mathbf{x}_1 - \mathbf{x}_0 - \boldsymbol{v}_\theta(t, \mathbf{x}_t)\|^2$, where $\mathbf{x}_t := (1 - t)\mathbf{x}_0 + t\,\mathbf{x}_1$ is a barycenter in the $[\mathbf{x}_0, \mathbf{x}_1]$ segment.

**Coupling Noise×Data.** A crucial ingredient in the FM methodology lies in choosing the *coupling* of noise/data used to sample pairs. Indeed, given a source and target distribution, there are infinitely many probability paths that can interpolate between them; choosing a different coupling yields a different loss and hence a different flow (Albergo et al., 2023; Liu, 2022). The default formulation of FM (Lipman et al., 2024) relies on the *independent* coupling, where noise and data are sampled independently. While simple and cheap, this approach is known to produce ODE paths with high curvature, which requires increased compute at inference time. In practice, the quality of samples is highly dependent on the number of function evaluations (NFEs) used to integrate the ODE.

---

[*]Joint first authorship, work done while AMH and SZ were interns at Apple

**Correspondence**: mousavi@cs.toronto.edu; syz@syz.id.au; michalk@apple.com; cuturi@apple.com

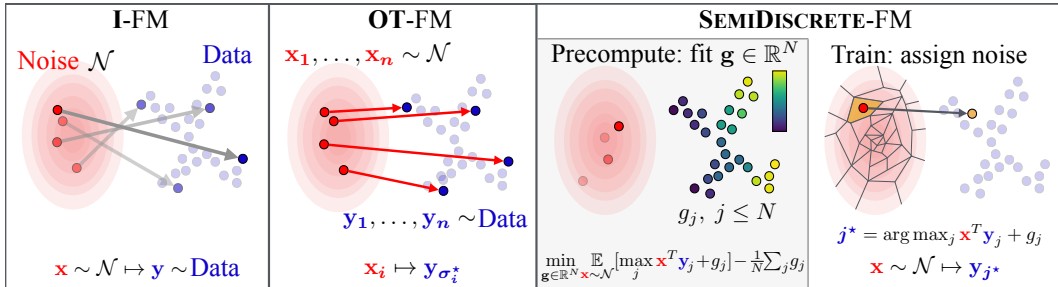

Figure 1: **I**-FM *(left)* assigns noise to data purely at random. **OT**-FM *(middle-left)* samples batches of $n$ noise and $n$ data points and re-aligns them with an optimal matching permutation $\sigma^\star$. These matches are, however, inherently unstable, as $n$ points do not reflect the whole noise distribution nor the dataset. Increasing drastically $n$ can mitigate this issue (Zhang et al., 2025), but at a significant cost. Our method, **SD**-FM *(right)*, solves these issues in two steps: in a precompute phase, the semidiscrete OT problem (parameterized as a vector of size $N$, the dataset size) is solved using SGD. At FM train time , each newly sampled noise is assigned to a data point using a maximum inner product search, *Laguerre* cells (Mérigot, 2011) being illustrated in the plot. Our figure uses no entropic regularization ($\varepsilon = 0$) and a neg-dot-product cost for simplicity, see (4) for more generality.

**Optimal Transport Flow Matching.** Tong et al. (2024) and Pooladian et al. (2023) have proposed using an optimal transport (OT) coupling to select noise/data pairs motivated by the dynamic least-action principle of Benamou & Brenier. Both works advocate sampling $n$ noise and data points, compute an optimal $n$-permutation (or $n \times n$ coupling matrix, from which pairs of indices are sampled from), as illustrated in the middle-left plot of Figure 1. While appealing in theory, OT-FM yields limited improvements for an additional precompute cost. In a follow-up work, Davtyan et al. (2025) have proposed to keep in memory the pairings returned by multiple Hungarian solvers run along training iterations, but this approach is bottlenecked by a price of $O(n^3)$ at each batch. Zhang et al. (2025) hypothesize that the mitigated results of OT-FM are due to the choice of a small $n$: Indeed, optimal matchings on small sample sizes are inherently unstable and cannot reliably approximate matches that would appear for far larger $n$ due to the curse of dimensionality (Hütter & Rigollet, 2021; Chewi et al., 2024). Zhang et al. (2025) propose to instead use the Sinkhorn algorithm with significantly larger $n$ (from 256 used originally to $\approx 10^6$) using a multi-GPU-node implementation, carefully ablating the role of $\varepsilon$ regularization. They argue that these computations are *precompute* costs, happening independently of FM training, and therefore can be cached beforehand. However, that cost grows as $O(n^2/\varepsilon^2)$ (Dvurechensky et al., 2018; Lin et al., 2019) for every $n$ pairs fed to FM training. Their finding that larger $n$ / smaller $\varepsilon$ yield even better results hinders the adoption of OT-FM as currently implemented (see also our discussion in Table 1).

**The Semidiscrete Approach.** Our work proposes an alternative route to OT-guided FM that does go to the costly process of optimally pairing batches of noises to data. We leverage the entropy regularized *semidiscrete* (SD) formulation of OT (Oliker & Prussner, 1989; Mérigot, 2011; Cuturi & Peyré, 2018; An et al., 2020), which studies OT from a *continuous* (noise) to a *discrete* (data) distribution. SD-OT relies on a potential vector of size $N$ of the target, fitted using SGD (Genevay et al., 2016). At FM train time, freshly sampled noise is paired to a point in the dataset in $O(N)$ time, as summarized in Figure 1. Following the presentation of background material in Section 2,

- We propose in Section 3 a new convergence criterion and convergence analysis of SGD for regularized SD-OT that is applicable to both the $\varepsilon = 0$ and $\varepsilon > 0$ cases.

- We introduce in Section 4 our semidiscrete FM (SD-FM) method, comparing it to FM and OT-FM in terms of memory and compute. We propose a generalization of the Tweedie formula (Robbins, 1956) for flows trained using SD-FM, with a correction $\boldsymbol{\delta}_\varepsilon$ that vanishes when $\varepsilon \approx 0$ or $\varepsilon \approx \infty$. We use it to sample from a geometric mixture of distributions using their velocity flows.

- We illustrate in Section 5 in a varied set of (un)conditional generative tasks the advantages of SD-FM over OT-FM and FM. We show that SD-FM results in a better pairing of noise to data (as seen in better metrics) for a negligible computational overhead relative to the cost of FM training.

## 2 BACKGROUND AND RELATED WORK

**Stochastic interpolants and flow matching.** For two probability distributions $\mu, \nu \in \mathcal{P}(\mathbb{R}^d)$, let $\Gamma(\mu, \nu) \subset \mathcal{P}(\mathbb{R}^d \times \mathbb{R}^d)$ be the set of couplings between $\mu$ and $\nu$, i.e. all joint distributions having $\mu$ as $\nu$ as their first and second marginal, respectively. Consider a source and target pair of distributions $\mu_0, \mu_1$ and $\pi \in \Gamma(\mu_0, \mu_1)$ a prescribed coupling between the two. For an interpolant function $(t, \mathbf{x}, \mathbf{y}) \mapsto \varphi_t(\mathbf{x}, \mathbf{y}) \in \mathbb{R}^d$ such that $\varphi_0(\mathbf{x}, \mathbf{y}) = \mathbf{x}$ and $\varphi_1(\mathbf{x}, \mathbf{y}) = \mathbf{y}$, Albergo et al. (2023, Definition 2.1) prove that to a cou-

$$\mathrm{d}\boldsymbol{X}_t = \boldsymbol{v}_t(\boldsymbol{X}_t)\mathrm{d}t, \quad \boldsymbol{X}_0 \sim \mu_0. \quad (1)$$

pling $\pi$ and an interpolant function $\varphi_t$ corresponds a flow generating a continuous *probability path* $\rho_t \in \mathcal{P}(\mathbb{R}^d)$ where $\rho_t := \varphi_t \# \pi$, where $\#$ is the pushforward operator, and $\rho_0 = \mu_0, \rho_1 = \mu_1$. That interpolant can be written equivalently as a *probability flow ODE* with a random initial condition (1), for a suitable time-parameterized velocity field $\boldsymbol{v}_t$. Access to $\boldsymbol{v}_t$ then allows for samples $\boldsymbol{X}_0 \sim \mu_0$ to be transformed into samples $\boldsymbol{X}_1 \sim \mu_1$ through evolution of (1). Training a neural approximation $\boldsymbol{v}_\theta$ for the flow field $\boldsymbol{v}$ can be done by sampling $(\boldsymbol{X}_0, \boldsymbol{X}_1) \sim \pi$ and $t \in [0, 1]$, setting $\boldsymbol{X}_t := \varphi_t(\boldsymbol{X}_0, \boldsymbol{X}_1)$ and minimizing (2).

The *linear* interpolant $\varphi_t(\mathbf{x}, \mathbf{y}) := (1-t)\mathbf{x} + t\mathbf{y}$ and the squared Euclidean distance for $\mathcal{L}$ are commonly used (Lipman et al., 2024, Eq. 2.3).

$$\min_\theta \mathbb{E}_{t, (\boldsymbol{X}_0, \boldsymbol{X}_1) \sim \pi} \mathcal{L}(\boldsymbol{X}_1 - \boldsymbol{X}_0, \boldsymbol{v}_\theta(t, \boldsymbol{X}_t)). \quad (2)$$

Albergo et al. (2023, Thm. 2.7) prove that the global minimum in (2) recovers $\boldsymbol{v}_t$ from (1). Other interpolants or loss functions (Song & Dhariwal, 2024; Kim et al., 2024) as well as penalizations facilitating one-step generation (Boffi et al., 2025, and references therein) have also been proposed.

**Straighter flows...** While the independent product coupling $\pi_I := \mu_0 \otimes \mu_1$ is the most widely used in practice (Lipman et al., 2024, §4.8.3), it may not be efficient in the sense that it induces *curved* flows. This means that adaptive ODE solvers consuming many neural function evaluations (NFEs) must be used to generate data at inference time. This is hardly surprising from an OT perspective (Santambrogio, 2015, §4), because the independent coupling $\pi_I$ is known to incur a high transportation cost, where for any valid coupling $\pi \in \Gamma(\mu_0, \mu_1)$ that cost is usually defined as $\mathcal{C}(\pi) := \mathbb{E}_{(\boldsymbol{X}_0, \boldsymbol{X}_1) \sim \pi_I} \|\boldsymbol{X}_0 - \boldsymbol{X}_1\|^2$. Concretely, although the flow $\boldsymbol{v}_t$ learned from $\pi_I$ might validly link $\mu_0$ to $\mu_1$, its *kinetic energy* $\int_0^1 \|\boldsymbol{v}_t\|^2_{L^2(\rho_t)} \mathrm{d}t$ is high. Benamou & Brenier (2000) show how optimal transport arises naturally as the least-action dynamics that transports $\mu_0$ to $\mu_1$. That map is exactly the Monge map, and the corresponding optimal flow paths are straight lines that would be trivial to integrate. To benefit from this insight, two families of works have emerged.

**... using Reflow.** Liu (2022) proposes to straighten a *pretrained* flow model (Liu et al., 2022) using *Reflow*, a method that forms noise/generated-data pairs, used subsequently to improve that model using FM. For Reflow to work, the pretrained flow model must be good enough to generate approximately the target distribution, and significant compute must be spent on generating at each reflow step (through ODEs) a large number of "virtual" data points. While successful (Liu et al., 2024), Reflow is a costly recursive *post-training* procedure that uses FM training as an *inner* iteration.

**... using pairs sampled from OT couplings.** Pooladian et al. (2023) and Tong et al. (2024) (see also discussion in Lipman et al. 2024, §4.9.2) propose OT-FM, a much lighter approach that simply replaces $\pi_I$ by a coupling $\pi$ that should, ideally, approximate the OT coupling $\pi^\star = \arg\min_{\pi \in \Gamma(\mu_0, \mu_1)} \mathcal{C}(\pi)$. Compared to Reflow, which uses FM as an *inner* step, OT-FM adds a *pre-processing* effort to FM training, to select better noise/data pairs. Because the ground-truth OT coupling $\pi^\star$ is never known, however, they use OT *matrices*: sampling $n \approx 256$ points from both $\mu_0$ and $\mu_1$, they match them using an OT solver (e.g. the Hungarian algorithm, Kuhn 1955) and feed these pairs to the flow loss (2). Davtyan et al. (2025) proposed then LOOM-CFM, a hybrid approach that stores buffers of paired noise/data samples (using the Hungarian algorithm). Unfortunately, OT-FM yields only modest gains. Zhang et al. (2025) posit that this is due to the well-known statistical bias arising when approximating continuous OT couplings using samples (Hütter & Rigollet, 2021). To reduce the effect of this bias, they use significantly higher $n$ handled with a multi-GPU, multi-node implementation of the Sinkhorn algorithm, carefully ablating the role of the $\varepsilon$ regularization parameter. While encouraging, their results show better performance as $n$ grows and $\varepsilon$ shrinks to 0, an explosive cocktail of hyperparameters since the cost of running Sinkhorn (1964) grows as $O(n^2/\varepsilon^2)$. Fundamentally, these approaches are bottlenecked by their reliance on the *discrete* view of OT problems – that is, they rely repeatedly on matching $n$ i.i.d. samples of noise and data.

**Conditional generation using OT.** Chemseddine et al. (2024); Kerrigan et al. (2024); Hosseini et al. (2025); Baptista et al. (2024) provided a noteworthy extension of OT-FM in the context of *conditional* generation, where each data point $\mathbf{x} \in \mathbb{R}^d$ is paired with a corresponding condition $\mathbf{z} \in \mathbb{R}^p$. The *data* is now an *augmented* random variable $(\boldsymbol{X}_1, \boldsymbol{Z}_1) \sim \tilde{\mu}_1 \in \mathcal{P}(\mathbb{R}^{d+p})$, with a marginal distribution $\tilde{\mu}_{1,\boldsymbol{Z}}$ over *conditions*. The goal remains to generate samples from noise in $\mathbb{R}^d$, but conditioned on a vector $\mathbf{z} \in \mathbb{R}^p$ of interest. OT-FM has a natural generalization to this setting, using the notion of condition-preserving *triangular* maps $T : (\mathbf{x}, \mathbf{z}) \mapsto (T_\mathbf{z}(\mathbf{x}), \mathbf{z})$ that couples conditional noise distribution $\tilde{\mu}_0 := \mathcal{N}(0, \mathbf{I}) \otimes \tilde{\mu}_{1,\boldsymbol{Z}} \in \mathcal{P}(\mathbb{R}^{d+p})$ with augmented data in $\tilde{\mu}_1$. Kerrigan et al. (2024, Prop. 1) (see also Baptista et al. 2024, Theorem 2.4) state that such triangular maps can be reduced to transport of distributions supported on the product space $\mathbb{R}^{d+p}$ following exactly OT-FM principles, implemented as ODE trajectories using a *conditional* vector field $\dot{\boldsymbol{X}}_t = \boldsymbol{v}(t, \boldsymbol{X}_t | \mathbf{z})$. OT-FM then consists in sampling $n$ noise and condition vectors, along with $n$ data points and their known condition, which are then optimally paired using an *augmented* cost:

$$c((\mathbf{x}, \mathbf{z}), (\mathbf{x}', \mathbf{z}')) = c_\mathcal{X}(\mathbf{x}, \mathbf{x}') + \beta c_\mathcal{Z}(\mathbf{z}, \mathbf{z}'), \quad \beta > 0. \tag{3}$$

**Semidiscrete optimal transport.** In the scenarios envisioned in this work, the noise measure $\mu_0$ is continuous, while the target measure $\mu_1$ of data is finite. This setting fits the stochastic optimization approach outlined in (Genevay et al., 2016, §2), defined for two probability measures $\mu, \nu \in \mathcal{P}_2(\mathbb{R}^d)$, a cost function $c : \mathbb{R}^d \times \mathbb{R}^d \to \mathbb{R}$ and entropic regularization $\varepsilon \geq 0$ as:

$$\min_{\pi \in \Gamma(\mu, \nu)} \mathcal{C}_\varepsilon(\pi) := \mathbb{E}_{(\boldsymbol{X}, \boldsymbol{Y}) \sim \pi}[c(\boldsymbol{X}, \boldsymbol{Y})] + \varepsilon \mathrm{KL}(\pi | \mu \otimes \nu), \tag{$\mathrm{P}_\varepsilon$}$$

This problem can be written in a *semidual* form (Cuturi & Peyré, 2018), leveraging the fact that $\nu$ is a finite measure, $\nu := \sum_{j=1}^N b_j \delta_{\mathbf{y}_j}$ where $\mathbf{b} = (b_1, \ldots, b_N)$ is a probability vector. For a potential vector $\mathbf{g} = (g_1, \ldots, g_N) \in \mathbb{R}^N$, we define the soft-$c$ transform (Peyré & Cuturi, 2019, §5.3) as:

$$f_{\mathbf{g}, \varepsilon} : \mathbf{x} \in \mathbb{R}^d \mapsto \begin{cases} -\varepsilon \log \left[ \sum_{j=1}^N b_j \exp \left( \frac{g_j - c(\mathbf{x}, \mathbf{y}_j)}{\varepsilon} \right) \right], & \varepsilon > 0 \\ -\max_{j=1}^N [g_j - c(\mathbf{x}, \mathbf{y}_j)], & \varepsilon = 0. \end{cases} \in \mathbb{R} \tag{4}$$

One can solve ($\mathrm{P}_\varepsilon$) by solving the *concave maximization* semidual problem parameterized entirely by $\mathbf{g}$, that Genevay et al. (2016, Alg. 2) proposed to solve using averaged SGD (our Algorithm 1)

$$\max_{\mathbf{g} \in \mathbb{R}^N} F_\varepsilon(\mathbf{g}) := \mathbb{E}_{\boldsymbol{X} \sim \mu}[f_{\mathbf{g}, \varepsilon}(\boldsymbol{X})] + \langle \mathbf{b}, \mathbf{g} \rangle. \tag{$\mathrm{S}_\varepsilon$}$$

For any $\varepsilon \geq 0$, the values of ($\mathrm{P}_\varepsilon$) and ($\mathrm{S}_\varepsilon$) are equal. Moreover for $\varepsilon > 0$, given a maximizing potential $\mathbf{g}^\star = (g_1^\star, \ldots, g_N^\star)$ of ($\mathrm{S}_\varepsilon$), one can recover a minimizer $\pi_\varepsilon^\star$ of ($\mathrm{P}_\varepsilon$) that has density

$$\mathrm{d}\pi_\varepsilon^\star(\mathbf{x}, \mathbf{y}_j) = \exp \left( \frac{f_{\mathbf{g}, \varepsilon}(\mathbf{x}) + g_j^\star - c(\mathbf{x}, \mathbf{y}_j)}{\varepsilon} \right) b_j \, \mathrm{d}\mu(\mathbf{x}). \tag{5}$$

## 3 IMPROVED OPTIMIZATION FOR SEMIDISCRETE OPTIMAL TRANSPORT

We adapt the approach of Genevay et al. (2016) to the much larger scales used in this work through two novel contributions: we propose an unbiased convergence criterion to monitor convergence, and a theoretical analysis to inform the computation of SD-OT that works *also* for the case $\varepsilon = 0$. Genevay et al. (2016) had not considered that case, which turns out to be the most promising when using SD-OT within the SD-FM methodology presented next in §4.

**Marginal Estimation for SD Couplings.** The primal-dual relationship in (5) holds at optimality, but can *also* be used to create a couplingby plugging a vector $\mathbf{g} \in \mathbb{R}^N$ into (5) to define a joint probability that only satisfies the *first* marginal constraint. The exponentiation in (5) can be rewritten using the softmax operator mapping a vector $\mathbf{z} = (z_1, \ldots, z_N)$ to a vector in the simplex $\Delta^N$,

$$\boldsymbol{\sigma}_{\mathbf{b}, \varepsilon}(\mathbf{z}) = \left\{ \left[ \frac{b_j \exp(z_j/\varepsilon)}{\sum_{k=1}^N b_k \exp(z_k/\varepsilon)} \right]_j, \quad \varepsilon > 0, \quad \left[ \frac{\mathbb{1}[j \in \arg\max_\ell z_\ell] b_j}{\sum_{k=1}^N \mathbb{1}[k \in \arg\max_\ell z_\ell] b_k} \right]_j, \quad \varepsilon = 0. \right. \tag{6}$$

This helps to define a coupling $\pi_{\varepsilon, \mathbf{g}}$ that arises from a partial optimization of the dual, as

$$\mathrm{d}\pi_{\varepsilon, \mathbf{g}}(\mathbf{x}, \mathbf{y}_j) := [s_{\varepsilon, \mathbf{g}}(\mathbf{x})]_j \mathrm{d}\mu(\mathbf{x}), \text{ where } \mathbf{s}_{\varepsilon, \mathbf{g}}(\mathbf{x}) := \boldsymbol{\sigma}_{\mathbf{b}, \varepsilon}([g_j - c(\mathbf{x}, \mathbf{y}_j)]_j) \in \Delta^N. \tag{7}$$

By construction, the first marginal of $\pi_{\varepsilon, \mathbf{g}}$ is $\mu$: Indeed, the measure $\pi_{\varepsilon, \mathbf{g}}(\mathbf{x}, \mathbf{y}_j)$ in (7) sums to $\mathrm{d}\mu(\mathbf{x})$ when integrated against the data for a fixed $\mathbf{x}$, since summing w.r.t. index $j$ is akin to summing a weighted soft-max distribution. On the other end, $\pi_{\varepsilon, \mathbf{g}}$ would be an OT coupling if and only if its

second marginal, defined as $\mathbf{m}(\mathbf{g})$ through an integral in (8), coincided

$$\mathbf{m}(\mathbf{g}) := [\pi_{\varepsilon,\mathbf{g}}(\mathbb{R}^d, \mathbf{y}_j)]_j = \int_{\boldsymbol{x} \in \mathbb{R}^d} \mathbf{s}_{\varepsilon,\mathbf{g}}(\mathbf{x}) \mathrm{d}\mu(\mathbf{x}) \in \Delta^N. \quad (8)$$

with $\mathbf{b}$, by analogy to the Sinkhorn algorithm (Peyré & Cuturi, 2019, §4.14). Note that $\pi_{\varepsilon,\mathbf{g}}$ solves $(\mathrm{P}_\varepsilon)$ if $\mathbf{g}$ is a maximizer of $(\mathrm{S}_\varepsilon)$.

**Convergence Criterion.** To our knowledge, no convergence criterion has been considered in the stochastic SD-OT literature (An et al., 2020, Alg.1), (Genevay et al., 2016, Alg.2). Recall that $\mathbf{g}$ is the solution to the dual problem if and only if $\mathbf{m}(\mathbf{g}) = \mathbf{b}$, i.e. the $\nu$-marginal constraint is satisfied.

A good criterion for convergence could be given by a distance between $\mathbf{m}(\mathbf{g})$ and $\mathbf{b}$. A natural candidate could be total variation, $\mathrm{TV}(\mathbf{m}(\mathbf{g}), \mathbf{b}) = \frac{1}{2}\|\mathbf{m}(\mathbf{g}) - \mathbf{b}\|_1$, as often used to track the convergence of the Sinkhorn algorithm. Unfortunately for continuous $\mu$, the expectation used in $\mathbf{m}(\mathbf{g})$ would need to be replaced with an empirical mean, leading to a biased estimate, because that expectation would be inside the norm. Reducing that bias would require a number of samples scaling linearly with $N$, making it prohibitively large. Fortunately, one can efficiently obtain an unbiased estimator for the $\chi^2$ divergence, defined for two vectors $\mathbf{p}, \mathbf{q} \in \Delta^N$ as $\chi^2(\mathbf{p}\|\mathbf{q}) = \sum_j (p_j/q_j)^2 q_j - 1$. That $\chi^2$ divergence can be formulated as an expectation, as highlighted in the fact below, proved in A.1:

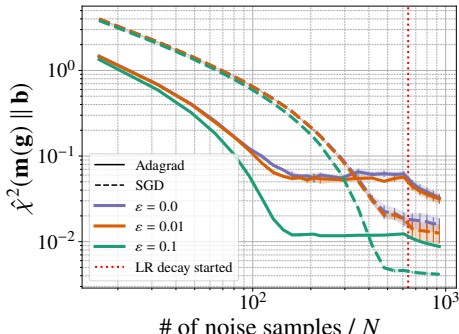

Figure 2: **SD-OT convergence**: $\widehat{\chi}^2$-divergence vs. SGD optimization steps for ImageNet-32, averaged over 3 seeds. See Appendix C for details.

$$\textbf{Fact 1}: \chi^2(\mathbf{m}(\mathbf{g}) \| \mathbf{b}) = \iint_{\mathbf{x},\mathbf{x}' \in \mathbb{R}^d} \sum_j \frac{1}{b_j} [s_{\varepsilon,\mathbf{g}}(\mathbf{x})]_j [s_{\varepsilon,\mathbf{g}}(\mathbf{x}')]_j \mathrm{d}\mu(\mathbf{x})\mathrm{d}\mu(\mathbf{x}') - 1. \quad (9)$$

When $\nu$ is uniform, by Lemma 7, the above $\chi^2$ divergence is also equal to the rescaled squared norm of the semidual gradient. From (9), we propose in (10) an unbiased estimator using a batch of i.i.d. samples $\mathbf{x}_1, \ldots, \mathbf{x}_B \sim \mu$ that can be computed in $O(NB)$ time. Figure 2 shows that we are able to effectively track $\widehat{\chi}^2$ as it decays towards zero along SGD updates.

$$\widehat{\chi}^2(\mathbf{b}(\mathbf{g}) \| \mathbf{b}) = \frac{1}{B(B-1)} \sum_{j=1}^N \frac{1}{b_j} \left( (\sum_i [s_{\varepsilon,\mathbf{g}}(\mathbf{x}_i)]_j)^2 - \sum_i [s_{\varepsilon,\mathbf{g}}(\mathbf{x}_i)]_j^2 \right) - 1. \quad (10)$$

**Convergence Analysis.** Studying the case $\varepsilon = 0$ requires some additional regularity assumptions, that we use to state the convergence guarantee of SGD for SD-OT in Theorem 2.

**Assumption 1.** Suppose $c(\mathbf{x}, \mathbf{y}) = -\mathbf{x}^\top \mathbf{y}$, $\delta := \min_{j \neq j'} \|\mathbf{y}_j - \mathbf{y}_{j'}\| > 0$, and $\mu$ admits a density w.r.t. the Lebesgue measure. Additionally, suppose the maximum $\mu$-surface area of any convex subset of $\mathbb{R}^d$ is bounded by $C_\mu^{\max}$, i.e. $\int_{\partial A} \mu(\mathbf{x}) \mathrm{d}S(\mathbf{x}) \leq C_\mu^{\max}$ for all convex $A \subseteq \mathbb{R}^d$, $S$ being the $d-1$-dimensional Euclidean surface measure. When $\mu = \mathcal{N}(0, \mathbf{I}_d)$, $C_\mu^{\max} \leq 4d^{1/4}$ (Ball, 1993).

**Theorem 2.** *Suppose either $\varepsilon > 0$ or Assumption 1 is satisfied. Let $L_\varepsilon := 1/\varepsilon$ for $\varepsilon > 0$ and $L_0 := C_\mu^{\max}/\delta$ else. For any $K \in \mathbb{N}$, let $\eta_k = \sqrt{\Delta/(L_\varepsilon K)}$ be a constant learning rate, where $\Delta := F_\varepsilon^\star - F_\varepsilon(\mathbf{0})$. Let $\mathbf{g}_k$ denote the SGD iterates with $\mathbf{g}_0 = \mathbf{0}$, and let $t \sim \mathrm{Unif}(\{0, \ldots, K-1\})$. Let $\mathbf{g}_\varepsilon^\star$ be an optimal dual solution to Equation $(\mathrm{S}_\varepsilon)$. Then, for any $\varepsilon \geq 0$, and taking expectations w.r.t. the randomness in noise samples and $t$,*

$$\mathbb{E}[\chi^2(\mathbf{m}(\mathbf{g}_t) \| \mathbf{b})] \lesssim \frac{1}{\min_j b_j} \sqrt{\frac{L_\varepsilon \Delta}{K}} \quad \text{and} \quad \mathbb{E}[\mathcal{C}_\varepsilon(\pi_{\varepsilon,\mathbf{g}_t})] - \mathcal{C}_\varepsilon^\star \lesssim \left( \|\mathbf{g}_\varepsilon^\star\|^2 + \frac{\Delta}{L_\varepsilon} \right)^{1/2} \cdot \left( \frac{L_\varepsilon \Delta}{K} \right)^{1/4}.$$

If the number of iterations is not fixed a priori, one can use the decaying schedule $\eta_k = \sqrt{\Delta/(L_\varepsilon k)}$, which replaces $1/K$ with $\log K/K$ in the bounds above. The above statement shows how the second marginal of $\pi_{\varepsilon,\mathbf{g}}$ approximates the correct marginal $\nu$, and how its (entropic) transport cost approximates the optimal cost as $K$ grows. Interestingly, both entropic $\varepsilon > 0$ and unregularized $\varepsilon = 0$ cases (pending Assumption 1) are valid, supporting our use of $\varepsilon = 0$ throughout the paper.

## 4 SEMIDISCRETE TRANSPORT FOR FLOW MATCHING

We discuss the comparative advantages of SD-FM over OT-FM and I-FM. We follow with a generalized Tweedie identity for flows trained with regularized SD-FM that incorporates a correction term. That term vanishes when either $\varepsilon \to \infty$ (I-FM) but also, more surprisingly, when $\varepsilon \to 0$.

**SD-FM *vs.* OT-FM *vs.* I-FM.** Building on Section 3, the OT problem is solved between Gaussian noise $\mu_0$ and the data measure $\mu_1$ supported on $(\mathbf{x}_1^{(1)}, \ldots, \mathbf{x}_1^{(N)})$. This framework can also accommodate the conditional generation setting, in which observations are paired with a $p$-dimensional condition, see Section 2. We follow Zhang et al. (2025, §3) and use the dot-product cost $c(\mathbf{x}, \mathbf{y}) := -\mathbf{x}^T \mathbf{y}$ (augmenting it with temperature $\beta$ for conditioning (3)) and rescale $\varepsilon$ with the cost matrix standard deviation for a sampled reference batch. As illustrated in Figure 1, SD-FM works in two steps, (1) solve SD-OT between $\mu_0$ and $\mu_1$ *before* flow training, storing $\mathbf{g}^\star \in \mathbb{R}^N$ solving (S$_\varepsilon$); (2) in the FM train loop, pair a freshly sampled noise $\mathbf{x}_0$ to data $\mathbf{x}_1^{(k)}$, where $k \sim s_{\varepsilon, \mathbf{g}^\star}(\mathbf{x}_0) \in \Delta^N$.

As recalled in (7), for $\varepsilon > 0$, this amounts to a *categorical* sampling among $N$ points, while for $\varepsilon = 0$ this reduces to an assignment to the $k^\star$-th point in the dataset, $k^\star = \arg\max_k g_k^\star + \langle \mathbf{x}_0, \mathbf{x}_1^{(k)} \rangle$ (or sample uniformly among ties if they arise). The latter operation is slightly faster than categorical sampling, as observed in Fig. 4. The differences between these three approaches are highlighted in Algorithms 3, 4, 5, 6. In a nutshell, SD-FM uses a simple SGD precompute effort to fit $\mathbf{g}$, with a minor lookup

| Coupling | Memory | Preprocessing | FM Training |
|----------|--------|---------------|-------------|
| **I-FM** | - | - | $NE\Theta$ |
| **OT-FM** | - | - | $NE(\Theta + O(dn/\varepsilon^2))$ |
| **OT-FM\*** | $2NE$ | $O(NEdn/\varepsilon^2)$ | $NE\Theta$ |
| **SD-FM** | $N$ | $NdK$ | $NE(\Theta + dN)$ |

Table 1: **Complexity of I/OT/SD-FM** given $N$: dataset size, $d$: data dimension, $E$: FM training epochs, $n$: Sinkhorn batch size, $\varepsilon$: entropic regularization, $\Theta$: cost of computing FM loss' gradient for a pair. $K$: SD SGD steps. OT-FM\* is the cached variant of OT-FM. $\Theta$ dominates all costs, $N \geq n, \varepsilon \approx 0$.

effort to pair each noise with a datapoint, rather than making potentially costly calls to Sinkhorn in OT-FM. That lookup is a MIPS procedure (Shrivastava & Li, 2014) when $\varepsilon = 0$, solved exactly, incurring a $Nd$ cost (noise $\times$ data dot-products) and a max over $N$ values. This might be sped-up with the use of *approximate* MIPS, left for future work.

**Score Estimation and Guidance** A key property that connects flow matching and diffusion models is that learning the marginal velocity field under independent coupling and Gaussian noise is equivalent to estimating the marginal score along the probability path. Namely, let $\rho_t$ denote the law of $\mathbf{X}_t = (1-t)\mathbf{X}_0 + t\mathbf{X}_1$, where $\mathbf{X}_0 \sim \rho_0 = \mu$ and $\mathbf{X}_1 \sim \rho_1 = \nu$ independently. Recall that,

$$\nabla \log \rho_t(\mathbf{x}) = \mathbb{E}[\nabla_{\mathbf{x}_t} \log \rho_{t|1}(\mathbf{X}_t \mid \mathbf{X}_1) \mid \mathbf{X}_t = \mathbf{x}] = \frac{t\mathbf{v}_t(\mathbf{x}) - \mathbf{x}}{1-t}. \quad (11)$$

Note that the above is Tweedie's formula since the flow matching velocity that transports $\rho_0$ to $\rho_1$ satisfies $(1-t)\mathbf{v}_t(\mathbf{x}) = \mathbb{E}[\mathbf{X}_1 - \mathbf{X}_t \mid \mathbf{X}_t = \mathbf{x}]$. While (11) relates score and velocity, the second equality above relies on the independence of $\mathbf{X}_0$ and $\mathbf{X}_1$; it may no longer hold under general couplings. Surprisingly, we find that for SD-OT couplings (11) still holds approximately for $\varepsilon \approx 0$.

**Proposition 3** (Generalized Tweedie's Formula). *For any* $\mathbf{g} \in \mathbb{R}^N$, *let* $\mathbf{X}_0, \mathbf{X}_1 \sim \pi_{\varepsilon, \mathbf{g}}$, *let* $\rho_t$ *denote the density of* $\mathbf{X}_t = (1-t)\mathbf{X}_0 + t\mathbf{X}_1$ *for* $t < 1$, *and define* $\mathbf{v}_t(\mathbf{x}) := \mathbb{E}[\mathbf{X}_1 - \mathbf{X}_0 \mid \mathbf{X}_t = \mathbf{x}]$. *Suppose* $\rho_0 = \mathcal{N}(0, \mathbf{I}_d)$. *Then,* $\nabla \log \rho_t(\mathbf{x}) = \frac{t\mathbf{v}_t(\mathbf{x}) - \mathbf{x} + (1-t)\boldsymbol{\delta}_\varepsilon}{(1-t)^2}$ *holds for any* $\varepsilon \geq 0$, *where* $\boldsymbol{\delta}_\varepsilon = \frac{1}{\varepsilon}\mathbb{E}[-\nabla_{\mathbf{x}_0} c(\mathbf{X}_0, \mathbf{X}_1) + \mathbb{E}[\nabla_{\mathbf{x}_0} c(\mathbf{X}_0, \mathbf{X}_1) \mid \mathbf{X}_0] \mid \mathbf{X}_t = \mathbf{x}]$ *for* $\varepsilon > 0$, *and* $\boldsymbol{\delta}_0 = \mathbf{0}$.

For small $\varepsilon$ and ignoring other constants, one can show that $\|\boldsymbol{\delta}_\varepsilon\| \lesssim e^{-1/\varepsilon}/\varepsilon$, thus going to zero as $\varepsilon \to 0$. Intuitively, this happens because $\mathbf{X}_1$ becomes increasingly determined by $\mathbf{X}_0$ as $\varepsilon \to 0$. Note that $\varepsilon \to \infty$ recovers the independent coupling, and we obtain the same score as expected.

**Correcting the Guidance.** To see a direct application of Proposition 3, consider the case where we have two flow models $(\rho_{1,t})_{t=0}^1$ and $(\rho_{2,t})_{t=0}^1$ that are generated by velocity fields $\mathbf{v}_{1,t}$ and $\mathbf{v}_{2,t}$ respectively. For simplicity, we consider the case where both flows start from $\rho_0$. Our goal is to sample from $\rho_t^{(\gamma)} = \rho_{1,t}^\gamma \rho_{2,t}^{1-\gamma}/Z_t^{(\gamma)}$ for some $\gamma \geq 0$, where $Z_t^{(\gamma)}$ is the normalizing constant. One example is classifier-free guidance (Ho & Salimans, 2022) where $\rho_1$ is a conditional model and $\rho_2$ is unconditional or, more generally, autoguidance (Karras et al., 2024), where $\rho_2$ is a weaker model compared to $\rho_1$. While in practice we often directly combine the corresponding velocity fields, the samples are not guaranteed to be from $\rho_t^{(\gamma)}$, and we need additional "correction" as laid out below.

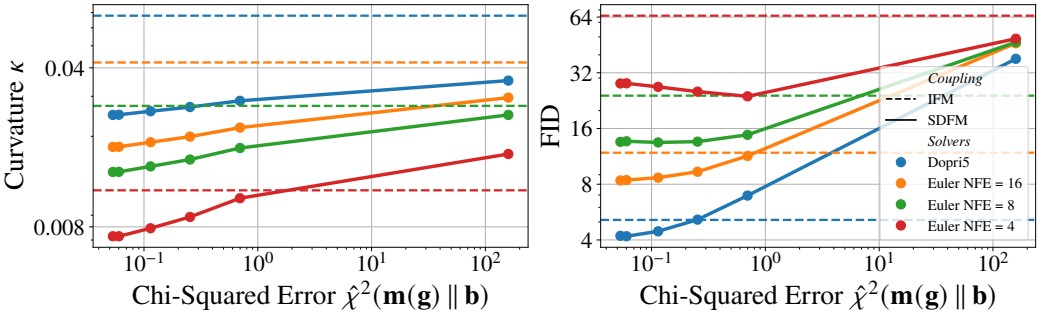

Figure 3: **Better SD Potential Estimation = Better Curvature and FID**: On ImgN32, convergence of dual potential **g** *vs.* SD-FM ($\varepsilon = 0$) curvature and FID; I-FM is shown as lines. Note that curvatures of different solvers are computed on different trajectories, hence they are not comparable.

**Proposition 4** (Informal). *Let $(\boldsymbol{X}_0^{(i)})_{i=1}^r \overset{\text{i.i.d.}}{\sim} \rho_0$, and define $(\boldsymbol{X}_t^{(\gamma,i)})_{t=0}^1$ by the ODE*

$$\mathrm{d}\boldsymbol{X}_t^{(\gamma,i)} = \{\gamma\boldsymbol{v}_{1,t}(\boldsymbol{X}_t^{(\gamma,i)}) + (1-\gamma)\boldsymbol{v}_{2,t}(\boldsymbol{X}_t^{(\gamma,i)})\}\mathrm{d}t, \quad \forall i \in [r].$$

*Furthermore, define the weights*

$$w^{(\gamma,i)} := \gamma(\gamma-1)\int \left\langle \boldsymbol{v}_{1,t}(\boldsymbol{X}_t^{(\gamma,i)}) - \boldsymbol{v}_{2,t}(\boldsymbol{X}_t^{(\gamma,i)}), \nabla\log\rho_{1,t}(\boldsymbol{X}_t^{(\gamma,i)}) - \nabla\log\rho_{2,t}(\boldsymbol{X}_t^{(\gamma,i)})\right\rangle\mathrm{d}t.$$

*Draw $I \sim \mathrm{softmax}(\{w^{(\gamma,i)}\}_{i=1}^r)$, and let $\boldsymbol{X}_t = \boldsymbol{X}_t^{(\gamma,I)}$. Then, $\mathrm{Law}(\boldsymbol{X}_t) \to \rho_t^{(\gamma)}$ as $r \to \infty$.*

This result can be seen as the flow matching variant of correctors used for diffusions (Bradley & Nakkiran, 2024; Skreta et al., 2025). Concretely, to sample from $\rho^\gamma$, one must generate a number of i.i.d. samples using the combined velocity field, and draw the outcome according to the weights. However, calculating weights requires knowledge of the scores $\nabla\log\rho_t$. Thanks to Proposition 3, either for the case of independent coupling ($\varepsilon = \infty$), or unregularized semidiscrete couplings ($\varepsilon = 0$), we can calculate them using the velocity model and obtain

$$w^{(\gamma,i)} = \gamma(\gamma-1)\int \frac{t}{1-t}\left\|\boldsymbol{v}_{1,t}(\boldsymbol{X}_t^{(\gamma,i)}) - \boldsymbol{v}_{2,\gamma}(\boldsymbol{X}_t^{(\gamma,i)})\right\|^2\mathrm{d}t.$$

## 5 EXPERIMENTS

### 5.1 IMAGENET & PETFACE: UNCONDITIONAL AND CLASS-CONDITIONAL GENERATION.

We consider generation in raw pixel space for the ImageNet (ImgN) train fold (Deng et al., 2009), augmented by horizontal flipping ($N$=2.56 M images) in (32,32) $d = 3072$, and (64,64) $d = 12\,288$ resolutions, and the PetFace dataset (Shinoda & Shiohara, 2024) of animal faces ($N$= 1.27 M images) in size (64,64). We measure generation quality using FID (Heusel et al., 2017) and flow curvature (Lee et al., 2023). I-FM, OT-FM and SD-FM training differ *exclusively* in the way noise-data pairs are sampled, all other aspects of FM training stay identical, using Pooladian et al.'s setup.

**FID *vs.* Potential Quality.** We investigate whether spending precompute effort to get better potential **g** for SD-FM translates into better model performance on ImgN-32. Concretely, we record six different iterates of **g** along the SGD iterations in Algorithm 1, tracking their convergence criteria $\widehat{\chi}^2(\mathbf{m}(\mathbf{g})\|\mathbf{b})$. We train six SD-FM models using these six vectors to assign noise to data. We plot in Figure 3 the FID and curvature across 4 ODE solvers. We observe that FID improves and curvature shrinks with lower $\widehat{\chi}^2$, confirming the main hypothesis of our paper that sharper SD-OT provides a more useful coupling for FM, with gains that seem to saturate when $\widehat{\chi}^2 \approx 0.05$. The runtime needed for SGD to converge is of the order of 12 hours on a node of 8×H100 GPUs.

**FID *vs.* Pairing Time Cost.** We consider the FIDs on ImgN returned by I-FM, replicate the setting in (Zhang et al., 2025) for OT-FM using $\varepsilon \in \{0.01, 0.1\}$ and $n \in \{2^{15}, 2^{17}, 2^{19}\}$, and finally SD-FM using a fixed $K$ budget for SGD with $\varepsilon \in \{0, 0.01, 0.1\}$. We show uncurated samples generated by I-FM and SD-FM in Figures 9 and 11. We provide results (second line) in ImgN-32 where *coupling* computations happen in the $k = 500$ principal components of the data, as proposed by (Zhang et al., 2025): this impacts *all* complexities presented in Table 1 by substituting $k$ for $d$. We stress that PCA

| | | | FID ↓ (Unconditional) | | | | FID ↓ (Class-conditional) | | | |
|---|---|---|---|---|---|---|---|---|---|---|
| | | | Euler 4 | Euler 8 | Euler 16 | Dopri5 | Euler 4 | Euler 8 | Euler 16 | Dopri5 |
| ImageNet-64 | I-FM | - | 79.95 | 37.90 | 19.68 | 9.10 | 34.51 | 12.95 | 7.39 | 3.91 |
| | SD-FM (PCA 500) | $\varepsilon = 0$ | **45.62** | **23.75** | **15.02** | **8.42** | 26.04 | **12.06** | **7.32** | **3.63** |
| | | $\varepsilon = 0.01$ | 45.68 | 23.91 | 15.10 | 8.45 | 26.15 | 12.26 | 7.51 | **3.63** |
| | | $\varepsilon = 0.1$ | 45.90 | 24.05 | 15.35 | 8.58 | **25.92** | 12.16 | 7.42 | 3.64 |
| PetFace | I-FM | - | 56.53 | 26.85 | 13.38 | 1.26 | 47.66 | 21.91 | 11.56 | 1.09 |
| | SD-FM (full $d$=12k) | $\varepsilon = 0$ | **20.54** | **12.77** | **7.50** | 1.26 | **19.10** | **12.10** | 7.13 | 1.05 |
| | | $\varepsilon = 0.01$ | 20.58 | 12.91 | 7.67 | 1.23 | 19.32 | **12.10** | **7.06** | **1.04** |
| | | $\varepsilon = 0.1$ | 20.96 | 12.96 | 7.59 | **1.18** | 19.46 | 12.20 | 7.18 | **1.04** |

Table 2: FID for **ImageNet** 64x64 and **PetFace** 64x64 unconditional/class-conditional generation.

is *only* used to speed up noise/data pairings – FM training is *unchanged*. For ImgN-64, the costs of running OTFM for large $n$ is too large in full $d = 12\,288$, we only report the PCA ($k = 500$) setting. SD-FM improves FID compared to I-FM, specially for cheaper fixed-step Euler solvers, for a far smaller time overhead compared to OT-FM. To put these overhead costs in perspective, we display in a red-dashed line the average time $\Theta$ (see Table 1) needed to compute the gradient of the FM loss on one pair. In summary, for a pairing overhead cost negligible w.r.t. $\Theta$, SD-FM delivers dramatically better performance than I-FM, especially for small inference budgets.

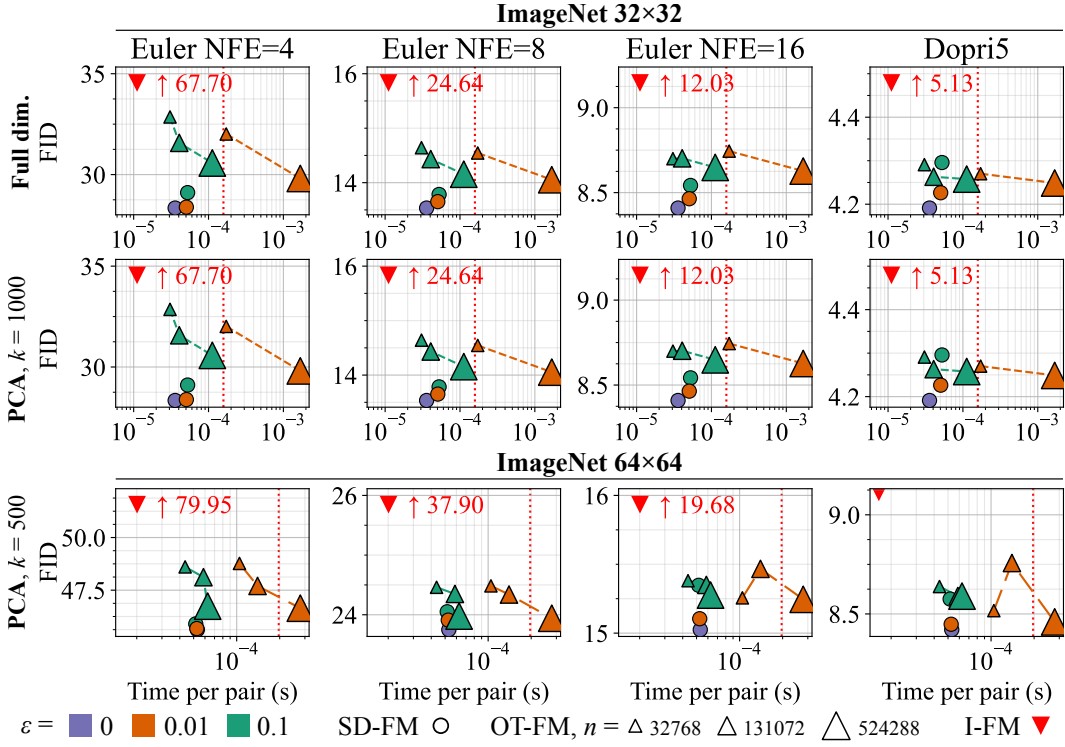

Figure 4: **FID** vs. **time needed to form a pair**, when training **I**-FM, **OT**-FM (varying batch sizes $n$) and **SD**-FM. We use $\varepsilon = 0$ (SD only), $0.01, 0.1$. Couplings are computed using full $d$ or PCA space. Red-dashed lines show the per-sample time $\Theta$ needed to compute the gradient of the loss for one pair. SD-FM yields significant improvements for a negligible overhead.

**Class-conditional ImgN and PetFace.** We train class-conditional flow models using SD-FM, following the principles presented in Section 4, taking the condition **z** to be a one-hot vector. ImgN and PetFace consist respectively of 1000 and 13 classes. As in the unconditional case and Figure 4, we show FID for ImgN as a function of time-per-pair for generation in Figures 7 and 8. Compared to the unconditional setting, we find that the gap between OT-FM and SD-FM is even more pronounced, owing to the slower convergence of Sinkhorn. At 64x64 resolution, the cost of matching points for OT-FM at large $n$ sizes becomes too great (over 10 days) and is therefore not shown. See results in

Table 2. Generated samples are shown in Figures 10, 12, 14. Class-conditional results in Table 2 are consistent with our observations for ImgN, with SD-FM outperforming I-FM in all cases.

## 5.2 CONTINUOUS-CONDITIONAL GENERATION: CELEBA SUPER-RESOLUTION

As an example application to *continuous-valued* conditions, we apply SD-FM to train flow models for conditional sampling in image super-resolution (SR) problems. We use the CelebA dataset (Liu et al., 2015) rescaled to 64x64 resolution, containing 162k train images. We downsize them to 16x16 (4x SR) or 8x8 (8x SR) and apply Gaussian noise with standard deviation $\sigma \in \{0.1, 0.2\}$ to form a condition $\mathbf{z}$ and

|  |  | Noise $\sigma = 0.1$ | | | Noise $\sigma = 0.2$ | | |
|  |  | I-FM | SD-FM | | I-FM | SD-FM | |
|  | $\varepsilon$ |  | 0 | 0.01 |  | 0 | 0.01 |
| 4x SR | PSNR (↑) | 21.17 | 21.38 | **21.41** | 20.04 | **20.44** | 20.42 |
|  | SSIM (↑) | 0.67 | **0.69** | **0.69** | 0.61 | **0.63** | **0.63** |
| 8x SR | PSNR (↑) | 17.52 | 17.87 | **17.94** | 16.50 | **17.35** | 17.34 |
|  | SSIM (↑) | 0.50 | 0.51 | **0.52** | 0.44 | **0.48** | **0.48** |

Table 3: **CelebA** super-resolution results.

run SD-FM with continuous data and conditions as described in Section 4. We compare to I-FM trained by sampling $\mathbf{x}_0 \sim \mathcal{N}(0, \mathbf{I}_d) \otimes \tilde{\mu}_{1,\mathbf{z}}$ and from the augmented data distribution of images $\mathbf{x}$ paired with their corrupted version $\mathbf{z}$. We sample from the trained model, conditioned on corrupted images from the CelebA validation set to obtain a high-resolution reconstruction.

Since the ground-truth image $\mathbf{x}$ is known, we rely on the peak signal-to-noise ratio (PSNR) and structural similarity index measure (SSIM, Wang et al. (2004)) to measure reconstruction accuracy. Results are summarized in Table 3 and we show sampled reconstructions from the validation set in Figure 16.

## 5.3 GUIDANCE: IMGN32

For ImgN-32, Figure 5 shows the effect of $r$, where $r = 1$ is the CFG baseline without any correction and $r = 0$ is generating from the conditional model without guidance. Increasing $r$ improves generation quality (higher precision) but degrades diversity (lower recall). We use the precision and recall calculation given by Kynkäänniemi et al. (2019).

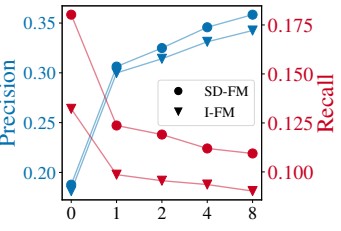

Figure 5: Precision & recall on ImgN-32 *vs.* # of guidance samples $r$. $\gamma = 2$, NFE $= 4$, $\varepsilon = 0$.

## 5.4 ONE-STEP GENERATION WITH MEAN-FLOW: LATENT SPACE IMGN256

Recently, consistency models (models that incorporate trajectory consistency regularizers in their training objective) have become a popular choice for few-step generation with diffusions or flows (Song et al., 2023; Song & Dhariwal, 2024). We demonstrate that the benefits of SD-OT couplings go beyond FM by testing them on the MeanFlow (MF) model of Geng et al. (2025), noting that different consistency formulations can be unified and treated similarly (Boffi et al., 2025; Sabour et al., 2025). Figure 6 compares SD-OT and independent couplings for training MF; we use the same setup as Geng et al. (2025) to train an unconditional DiT-XL/2 model for 300 epochs on ImgN-256x256 in latent space and M/2 for 40

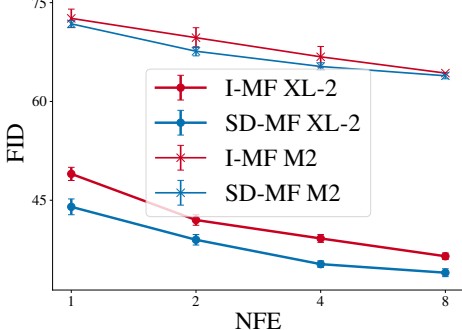

Figure 6: FID on ImgN-256 vs. NFE for mean-flows.

epochs. Note that SDOT in latent space may be easier than pixel space owing to the Gaussianity promoted in auto-encoder latents, see Fig.18.

### LIMITATIONS AND DISCUSSION

Some aspects of the SD-FM pipeline presented here leave room for improvement. Figure 2 shows that computing the SD-OT potential to a reasonable degree of precision impacts the performance of SD-FM. That SD-OT preprocessing cost occurs only once prior to FM training; it only depends on the noise distribution and data, and is typically much cheaper than the FM training cost itself. However, if SD-FM were to be applied to datasets of billions $N$ of points, this could become a challenge (though still less so than FM training). Because SD-OT is a concave maximization problem, one could leverage recipes such as batching, momentum or $\varepsilon$-tempering. For $\varepsilon > 0$, categorical sampling over a $N$-sized softmax vector would be costly, which is why we highlighted

the $\varepsilon = 0$ case that can leverage MIPS and fast retrieval tools (both to train $\mathbf{g}$ and to pair fresh noise). For these reasons, and due to only very minor differences in performace, we would recommend that users use SD-FM with $\varepsilon = 0$. Finally, having in mind dataloader efficiency, one may want to flip the tables in our sampling approach: Start from the $j$-th data point in memory, and match it with a Gaussian noise restricted (Wu & Gardner, 2024) to be in its Laguerre cell $\mathcal{L}_j = \{\mathbf{x} : \mathbf{x}^T(\mathbf{x}_1^{(j)} - \mathbf{x}_1^{(k)}) + g_j - g_k \geq 0, \forall k \neq i\}$. While we have shown successful applications of the SD coupling preprocessing approach to extended FM methods such as mean-flow or guidance, SD couplings could be of course used *within* even more complex approaches building upon FM, such as Reflow. Our focus was on ablating the impact of the SD coupling over the independent coupling in the most widely used FM setup. Assessing how SD couplings interact with more advanced or orthogonal methods is left for future work.

In concurrent work, Kong et al. (2026) also demonstrate the utility of SD-OT for improving flow models, focusing on the unconditional and class-conditional settings. In comparison, we introduce an efficient convergence criterion for using stochastic optimization to solve SD-OT, provide a theoretical convergence analysis that handles both $\varepsilon = 0$ and $\varepsilon > 0$, and generalize Tweedie's formula to SD-OT couplings, providing a score estimator that can be used in guidance techniques.

### CONCLUSION

OT-FM can been used to guide the training of flow models by optimally pairing batches of $n$ noise $n$ data points, with results only truly paying off for very large $n$. We proposed the more cost-efficient semidiscrete OT paradigm, building on a two-step approach (fit potential vector as a one-off cost, assign noise when training through a MIPS) that is not bottlenecked by a batch size $n$. Our experiments in various settings, both in unconditioned and conditioned scenarios, show much improved metrics over I-FM and orders of magnitude cheaper compute compared to OT-FM.

### REPRODUCIBILITY STATEMENT

We explain all the necessary details to reproduce our results in Section 5 and in Appendix C. We provide the code needed to compute and sample from semidiscrete couplings as an addition to the existing open source OTT-JAX toolbox (Cuturi et al., 2022) available at https://github.com/ott-jax/ott.

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

## A    OMITTED PROOFS

In this section, we provide missing derivations and proofs from the main text.

### A.1    PROPERTIES OF SEMIDISCRETE COUPLINGS

We begin by stating a quick proof of Equation (9)

*Proof of Equation* (9). By definition

$$\chi^2(\mathbf{m}(\mathbf{g}) \| \mathbf{b}) = \sum_{j=1}^{N} \left( \frac{m(\mathbf{g})_i}{b_i} \right)^2 b_i - 1.$$

$$= \sum_{j=1}^{N} \frac{1}{b_j} \left( \int s_{\varepsilon,\mathbf{g}}(\mathbf{x})_j \mathrm{d}\mu(\mathbf{x}) \right)^2 - 1$$

$$= \sum_{j=1}^{N} \int s_{\varepsilon,\mathbf{g}}(\mathbf{x})_j s_{\varepsilon,\mathbf{g}}(\mathbf{x}')_j \mathrm{d}\mu(\mathbf{x}) \mathrm{d}\mu(\mathbf{x}') - 1.$$

$\square$

Next, we present the proof of the score formula for our semidiscrete couplings.

*Proof of Proposition 3.* We first consider the case $\varepsilon > 0$. In this case, the conditional law $\boldsymbol{X}_0 \,|\, \boldsymbol{X}_1 = \mathbf{x}_1$ admits a density. We use the general denoising score-matching identity

$$\nabla \log \rho_t(\mathbf{x}) = \mathbb{E}[\nabla_{\mathbf{x}_t} \log \rho_{t|1}(\boldsymbol{X}_t \,|\, \boldsymbol{X}_1) \,|\, \boldsymbol{X}_t = \mathbf{x}], \tag{12}$$

which does not depend on the $\boldsymbol{X}_0, \boldsymbol{X}_1$ coupling. For the sake of completeness, we present the derivation of the above identity:

$$\nabla \log \rho_t(\mathbf{x}) = \frac{\nabla \rho_t(\mathbf{x})}{\rho_t(\mathbf{x})} = \frac{1}{\rho_t(\mathbf{x})} \int \nabla \rho_{t|1}(\mathbf{x} \,|\, \mathbf{x}_1) \rho_1(\mathrm{d}\mathbf{x}_1)$$

$$= \int \nabla \log \rho_{t|1}(\mathbf{x} \,|\, \mathbf{x}_1) \frac{\rho_{t|1}(\mathbf{x} \,|\, \mathbf{x}_1) \rho_1(\mathrm{d}\mathbf{x}_1)}{\rho_t(\mathbf{x})}$$

$$= \int \nabla \log \rho_{t|1}(\mathbf{x} \,|\, \mathbf{x}_1) \rho_{1|t}(\mathrm{d}\mathbf{x}_1 \,|\, \mathbf{x}).$$

Now, using the definition $\boldsymbol{X}_t := (1 - t)\boldsymbol{X}_0 + t\boldsymbol{X}_1$ and the change of variables formula, we have

$$\rho_{t|1}(\mathbf{x} \,|\, \mathbf{x}_1) = \frac{1}{(1 - t)^d} \rho_{0|1} \left( \frac{\mathbf{x} - t\mathbf{x}_1}{1 - t} \,|\, \mathbf{x}_1 \right).$$

As a result,

$$\nabla \log \rho_{t|1}(\mathbf{x} \,|\, \mathbf{x}_1) = \frac{1}{1 - t} \nabla \log \rho_{0|1} \left( \frac{\mathbf{x} - t\mathbf{x}_1}{1 - t} \,|\, \mathbf{x}_1 \right).$$

Consequently,

$$x \nabla \log \rho_t(\mathbf{x}) = \frac{1}{1 - t} \mathbb{E} \left[ \nabla_{\mathbf{x}_0} \log \rho_{0|1}(\boldsymbol{X}_0 \,|\, \boldsymbol{X}_1) \,|\, \boldsymbol{X}_t = \mathbf{x} \right].$$

Let us define $\nu(\boldsymbol{X}_1) = b_I$ and $g(\boldsymbol{X}_1) = g_I$ where $I$ is the index of the random vector $\boldsymbol{X}_1$, i.e. $\boldsymbol{X}_1 = \mathbf{x}_1^{(I)}$. Recall that for $\varepsilon > 0$, the coupling is given by

$$\rho_{0,1}(\boldsymbol{X}_0, \boldsymbol{X}_1) = \frac{e^{\frac{g(\boldsymbol{X}_1) - c(\boldsymbol{X}_0, \boldsymbol{X}_1)}{\varepsilon}}}{\sum_{\mathbf{x}_1} e^{\frac{g(\mathbf{x}_1) - c(\boldsymbol{X}_0, \mathbf{x}_1)}{\varepsilon}} \nu(\mathbf{x}_1)} \mu(\boldsymbol{X}_0) \nu(\boldsymbol{X}_1).$$

Thus, for $\mu = \mathcal{N}(0, \boldsymbol{I}_d)$, we have

$$\nabla_{\mathbf{x}_0} \log \rho_{0|1}(\boldsymbol{X}_0 \,|\, \boldsymbol{X}_1) = -\boldsymbol{X}_0 - \frac{1}{\varepsilon}\nabla_{\mathbf{x}_0} c(\boldsymbol{X}_0, \boldsymbol{X}_1) + \frac{1}{\varepsilon} \cdot \sum_{\mathbf{x}_1'} \frac{\nabla_{\mathbf{x}_0} c(\boldsymbol{X}_0, \mathbf{x}_1') e^{\frac{g(\mathbf{x}_1') - c(\boldsymbol{X}_0, \mathbf{x}_1')}{\varepsilon}} \nu(\mathbf{x}_1')}{\sum_{\mathbf{x}_1} e^{\frac{g(\mathbf{x}_1) - c(\boldsymbol{X}_0, \mathbf{x}_1)}{\varepsilon}} \nu(\mathbf{x}_1)}$$

$$= -\boldsymbol{X}_0 - \frac{1}{\varepsilon}\nabla_{\mathbf{x}_0} c(\boldsymbol{X}_0, \boldsymbol{X}_1) + \frac{1}{\varepsilon}\mathbb{E}\left[\nabla_{\mathbf{x}_0} c(\boldsymbol{X}_0, \boldsymbol{X}_1) \,|\, \boldsymbol{X}_0\right].$$

Plugging this back into (12), we obtain

$$\nabla \log \rho_t(\mathbf{x}) = -\frac{\mathbb{E}[\boldsymbol{X}_0 \,|\, \boldsymbol{X}_t = \mathbf{x}]}{1 - t} + \frac{1}{\varepsilon(1 - t)}\mathbb{E}\left[-\nabla_{\mathbf{x}_0} c(\boldsymbol{X}_0, \boldsymbol{X}_1) + \mathbb{E}[\nabla_{\mathbf{x}_0} c(\boldsymbol{X}_0, \boldsymbol{X}_1) \,|\, \boldsymbol{X}_0] \,|\, \boldsymbol{X}_t = \mathbf{x}\right]$$

$$= \frac{-\mathbb{E}[\boldsymbol{X}_0 \,|\, \boldsymbol{X}_t = \mathbf{x}] + \boldsymbol{\delta}_\varepsilon}{1 - t}.$$

Moreover, since the optimal velocity field in flow matching is given by the expectation of the conditional velocity field (Lipman et al., 2023), we have

$$\boldsymbol{v}_t(\mathbf{x}) = \mathbb{E}\left[\frac{\boldsymbol{X}_t - \boldsymbol{X}_0}{t} \,|\, \boldsymbol{X}_t = \mathbf{x}\right] = \frac{\mathbf{x} - \mathbb{E}[\boldsymbol{X}_0 \,|\, \boldsymbol{X}_t = \mathbf{x}]}{t}.$$

Combining the two identities above finishes the proof for the $\varepsilon > 0$ case.

For the $\varepsilon = 0$ case, we define $\boldsymbol{T}_1(\mathbf{x}_0) = \arg\max_{\mathbf{x}_1 \in \{\mathbf{x}_1^{(i)}\}_{i=1}^N} g(\mathbf{x}_1) - c(\mathbf{x}_0, \mathbf{x}_1)$, where we can arbitrarily break ties, by e.g. picking the maximizer with the smallest index. Then, with probability 1 over the draw of $\boldsymbol{X}_0$, the samples $\boldsymbol{X}_0, \boldsymbol{X}_1 \sim \rho_{0,1}$ are given by $\boldsymbol{X}_1 = \boldsymbol{T}_1(\boldsymbol{X}_0)$. As a result, we can write

$$\boldsymbol{X}_t = \boldsymbol{T}_t(\boldsymbol{X}_0) := (1 - t)\boldsymbol{X}_0 + t\boldsymbol{T}_1(\boldsymbol{X}_0).$$

Recall that by Brenier's theorem, $\boldsymbol{T}_1$ can be written as a gradient of a convex function, therefore $\boldsymbol{T}_t$ is strictly monotone and we can write $\boldsymbol{X}_0 = \boldsymbol{T}_t^{-1}(\boldsymbol{X}_t)$ for all $t < 1$. Further, we observe that $\boldsymbol{T}_1(\boldsymbol{X}_0)$ is a piecewise constant function, thus its Jacobian is $\boldsymbol{0}$ almost everywhere, i.e. we have $D\boldsymbol{T}_t(\boldsymbol{X}_0) = (1 - t)\boldsymbol{I}$ almost surely, where $D$ denotes Jacobian.

For any $\mathbf{x}_t$, define $\mathbf{x}_0 = \boldsymbol{T}_t^{-1}(\mathbf{x}_t)$. We can now use the change of variables formula

$$\rho_t(\mathbf{x}_t) = \frac{\rho_0(\mathbf{x}_0)}{|\det D\boldsymbol{T}_t(\mathbf{x}_0)|} = \frac{\rho_0(\boldsymbol{T}_t^{-1}(\mathbf{x}_t))}{(1 - t)^d}.$$

As a result,

$$\nabla \log \rho_t(\mathbf{x}_t) = D\boldsymbol{T}_t^{-1}(\mathbf{x}_t)\nabla \log \rho_0(\boldsymbol{T}_t^{-1}(\mathbf{x}_t)) = (D\boldsymbol{T}_t(\mathbf{x}_0))^{-1}\nabla \log \rho_0(\mathbf{x}_0)$$

$$= \frac{1}{1 - t}\nabla \log \rho_0(\mathbf{x}_0)$$

$$= -\frac{\mathbf{x}_0}{1 - t},$$

where we plugged in $\rho_0 = \mu = \mathcal{N}(0, \boldsymbol{I}_d)$ in the last step. Furthermore, since $\boldsymbol{X}_t$ is $\boldsymbol{X}_0$-measurable, we have

$$\boldsymbol{v}_t(\mathbf{x}_t) = \mathbb{E}\left[\frac{\boldsymbol{X}_t - \boldsymbol{X}_0}{t} \,|\, \boldsymbol{X}_t = \mathbf{x}_t\right] = \frac{\mathbf{x}_t - \boldsymbol{T}_t^{-1}(\mathbf{x}_t)}{t} = \frac{\mathbf{x}_t - \mathbf{x}_0}{t}.$$

Combining the above equations completes the proof. □

The following is the formal statement of Proposition 4.

**Proposition 5.** *Suppose $\boldsymbol{v}_{1,t}$ and $\boldsymbol{v}_{2,t}$ generate the probability paths $\rho_{1,t}$ and $\rho_{2,t}$ respectively. Let*

$$\boldsymbol{v}_{1,t}^{(\gamma)}(\mathbf{x}) := \gamma\boldsymbol{v}_{1,t}(\mathbf{x}) + (1 - \gamma)\boldsymbol{v}_{2,t}(\mathbf{x}),$$

*and define $\psi_t^{(\gamma)}(\mathbf{x})$ as the solution of the ODE $\frac{d\psi_t^{(\gamma)}(\mathbf{x})}{dt} = \boldsymbol{v}_t^{(\gamma)}(\mathbf{x})$ with the initial condition $\psi_0^{(\gamma)}(\mathbf{x}) = \mathbf{x}$, i.e. $\psi_t^{(\gamma)}(\mathbf{x})$ is the position of a particle initialized from $\mathbf{x}_0 = \mathbf{x}$, moving along the velocity field $\boldsymbol{v}_t^{(\gamma)}$. Define the weight*

$$w_t(\mathbf{x}) := \gamma(\gamma - 1)\int_{s=0}^t \left(\boldsymbol{v}_{1,s}(\psi_s^{(\gamma)}(\mathbf{x})) - \boldsymbol{v}_{2,t}(\psi_s^{(\gamma)}(\mathbf{x}))\right)^\top \left(\nabla \log \rho_{1,s}(\psi_s^{(\gamma)}(\mathbf{x})) - \nabla \log \rho_{2,s}(\psi_s^{(\gamma)}(\mathbf{x}))\right)ds.$$

*Then, for any test function $\phi : \mathbb{R}^d \to \mathbb{R}$,*

$$\int \phi(\mathbf{x}) \mathrm{d}\rho_t^{(\gamma)}(\mathbf{x}) = \frac{\int e^{w_t(\mathbf{x})} \phi(\psi_t^{(\gamma)}(\mathbf{x})) \mathrm{d}\rho_0(\mathbf{x})}{\int e^{w_t(\mathbf{x})} \mathrm{d}\rho_0(\mathbf{x})}.$$

Before presenting the proof of Proposition 5, we comment on its informal interpretation, Proposition 4. Let $\rho_0^{(r)}$ denote the stochastic empirical law of $k$ i.i.d. random vectors $(\mathbf{X}_0^{(i)})_{i=1}^r$ from $\rho_0$. Then, if $\rho_0$ is sufficiently concentrated, we have almost surely for all test functions

$$\frac{\int e^{w_t(\mathbf{x})} \phi(\psi_t^{(\gamma)}(\mathbf{x})) \mathrm{d}\rho_0^{(r)}(\mathbf{x})}{\int e^{w_t(\mathbf{x})} \mathrm{d}\rho_0^{(r)}(\mathbf{x})} \to \frac{\int e^{w_t(\mathbf{x})} \phi(\psi_t^{(\gamma)}(\mathbf{x})) \mathrm{d}\rho_0(\mathbf{x})}{\int e^{w_t(\mathbf{x})} \mathrm{d}\rho_0(\mathbf{x})} \tag{13}$$

as $r \to \infty$. Define $\rho_t^{(r,\gamma)}$ as the measure that satisfies

$$\int \phi(\mathbf{x}) \mathrm{d}\rho_t^{(r,\gamma)}(\mathbf{x}) = \frac{\int e^{w_t(\mathbf{x})} \phi(\psi_t^{(\gamma)}(\mathbf{x})) \mathrm{d}\rho_0^{(r)}(\mathbf{x})}{\int e^{w_t(\mathbf{x})} \mathrm{d}\rho_0^{(r)}(\mathbf{x})},$$

for all test functions $\phi$. Then, sampling from $\rho_t^{(r,\gamma)}$ is exactly equivalent to drawing from $(\psi_t^{(\gamma)}(\mathbf{X}_0^{(i)}))_{i=1}^r$ according to (unnormalized) weights $e^{w_t(\mathbf{X}_0^{(i)})}$. Finally, as $r \to \infty$, we have

$$\int \phi(\mathbf{x}) \mathrm{d}\rho_t^{(r,\gamma)}(\mathbf{x}) = \frac{\int e^{w_t(\mathbf{x})} \phi(\psi_t^{(\gamma)}(\mathbf{x})) \mathrm{d}\rho_0^{(r)}(\mathbf{x})}{\int e^{w_t(\mathbf{x})} \mathrm{d}\rho_0^{(r)}(\mathbf{x})} \to \frac{\int e^{w_t(\mathbf{x})} \phi(\psi_t^{(\gamma)}(\mathbf{x})) \mathrm{d}\rho_0(\mathbf{x})}{\int e^{w_t(\mathbf{x})} \mathrm{d}\rho_0(\mathbf{x})} = \int \phi(\mathbf{x}) \mathrm{d}\rho_t^{(\gamma)}(\mathbf{x})$$

as $r \to \infty$. Therefore, $\rho_t^{(r,\gamma)}$ converges almost surely, weakly to $\rho_t^{(\gamma)}$. This is the sense in which the convergence of Proposition 4 holds. We keep the argument informal since we do not study the exact concentration properties required for (13) to hold.

To prove Proposition 5, we recall this result about the solution of an advection-reaction PDE.
**Lemma 6.** *Suppose $(\rho_t)_{t \geq 0}$ satisfies*

$$\partial_t \rho_t = -\nabla \cdot (\rho_t \boldsymbol{v}_t) + \rho_t(\alpha_t - \int \alpha_t \mathrm{d}\rho_t), \tag{14}$$

*where $\boldsymbol{v}_t : \mathbb{R}^d \to \mathbb{R}^d$ and $\alpha_t : \mathbb{R}^d \to \mathbb{R}$. Let $\psi_t(\mathbf{x})$ be the solution to the ODE $\frac{\mathrm{d}\psi_t(\mathbf{x})}{\mathrm{d}t} = \boldsymbol{v}_t(\psi_t(\mathbf{x}))$ with initial condition $\psi_0(\mathbf{x}) = \mathbf{x}$. Further define*

$$w_t(\mathbf{x}) := \int_{s=0}^t \alpha_s(\psi_s(\mathbf{x})) \mathrm{d}s. \tag{15}$$

*Then, for any $t \geq 0$ and any test function $\phi : \mathbb{R}^d \to \mathbb{R}$,*

$$\int \phi(\mathbf{x}) \mathrm{d}\rho_t(\mathbf{x}) = \frac{\int e^{w_t(\mathbf{x})} \phi(\psi_t(\mathbf{x})) \mathrm{d}\rho_0(\mathbf{x})}{\int e^{w_t(\mathbf{x})} \mathrm{d}\rho_0(\mathbf{x})}. \tag{16}$$

*Proof.* We proceed by differentiating (16) against time to obtain (14). We have

$$\partial_t \int \phi(\mathbf{x}) \mathrm{d}\rho_t(\mathbf{x}) = \frac{\int e^{w_t(\mathbf{x})} \langle \nabla\phi(\psi_t(\mathbf{x})), \boldsymbol{v}_t(\psi_t(\mathbf{x})) \rangle \mathrm{d}\rho_0(\mathbf{x})}{\int e^{w_t(\mathbf{x})} \mathrm{d}\rho_0(\mathbf{x})} + \frac{\int e^{w_t(\mathbf{x})} \alpha_t(\psi_t(\mathbf{x})) \phi(\psi_t(\mathbf{x}))}{\int e^{w_t(\mathbf{x})} \mathrm{d}\rho_0(\mathbf{x})}$$

$$- \frac{\int e^{w_t(\mathbf{x})} \phi(\psi_t(\mathbf{x})) \mathrm{d}\rho_0(\mathbf{x})}{\int e^{w_t(\mathbf{x})} \mathrm{d}\rho_0(\mathbf{x})} \cdot \frac{\int e^{w_t(\mathbf{x})} \alpha_t(\psi_t(\mathbf{x})) \mathrm{d}\rho_0(\mathbf{x})}{\int e^{w_t(\mathbf{x})} \mathrm{d}\rho_0(\mathbf{x})}$$

$$= \int \langle \nabla\phi(\mathbf{x}), \boldsymbol{v}_t(\mathbf{x}) \rangle \mathrm{d}\rho_t(\mathbf{x}) + \int \phi(\mathbf{x}) \left( \alpha_t(\mathbf{x}) - \int \alpha_t \mathrm{d}\rho_t \right) \mathrm{d}\rho_t(\mathbf{x}),$$

where the last equality holds by the change of variables formula. This is exactly the weak sense of (14). $\square$

As a result, to prove Proposition 5, we only need to show that $\rho_t^{(\gamma)}$ satisfies a PDE of the form (14).

*Proof of Proposition 5.* Let $\tilde{\rho}_t^{(\gamma)} := \rho_{1,t}^{\gamma}\rho_{2,t}^{1-\gamma}$ be the unnormalized measure, and recall $Z_t^{(\gamma)} = \int \tilde{\rho}_t^{(\gamma)}(\mathbf{x})d\mathbf{x}$ is the normalizing constant. We have

$$
\begin{aligned}
\partial_t\tilde{\rho}_t^{(\gamma)} =& \gamma\rho_{1,t}^{\gamma-1}\rho_{2,t}^{1-\gamma}\frac{\partial_t\rho_{1,t}}{\partial t} + (1-\gamma)\rho_{1,t}^{\gamma}\rho_{2,t}^{-\gamma}\frac{\partial\rho_{2,t}}{\partial t} \\
=& -\gamma\rho_{1,t}^{\gamma-1}\rho_{2,t}^{1-\gamma}\nabla\cdot(\rho_{1,t}\boldsymbol{v}_{1,t}) - (1-\gamma)\rho_{1,t}^{\gamma}\rho_{2,t}^{-\gamma}\nabla\cdot(\rho_{2,t}\boldsymbol{v}_{2,t}) \\
=& -\nabla\cdot(\tilde{\rho}_t^{(\gamma)}\boldsymbol{v}_t^{(\gamma)}) + \gamma(\gamma-1)\tilde{\rho}_t^{(\gamma)}\langle\boldsymbol{v}_{1,t} - \boldsymbol{v}_{2,t}, \nabla\log\rho_{1,t} - \nabla\log\rho_{2,t}\rangle,
\end{aligned}
$$

where we recall $\boldsymbol{v}_t^{(\gamma)} = \gamma\boldsymbol{v}_{1,t} + (1-\gamma)\boldsymbol{v}_{2,t}$, and the third identity follows from the product rule $\nabla\cdot(m\boldsymbol{v}) = \nabla m\cdot\boldsymbol{v} + m\nabla\cdot\boldsymbol{v}$. Define

$$\alpha_t(\mathbf{x}) := \gamma(\gamma-1)\langle\boldsymbol{v}_{1,t}(\mathbf{x}) - \boldsymbol{v}_{2,t}(\mathbf{x}), \nabla\log\rho_{1,t}(\mathbf{x}) - \nabla\log\rho_{2,t}(\mathbf{x})\rangle.$$

Then,

$$\partial_t\rho_t^{(\gamma)}(\mathbf{x}) = \frac{1}{Z_t^{(\gamma)}}\partial_t\tilde{\rho}_t^{(\gamma)} - \frac{\rho_t^{(\gamma)}}{Z_t^{(\gamma)}}\partial_tZ_t^{(\gamma)}.$$

Note that $Z_t^{(\gamma)}$ is constant in $\mathbf{x}$, and $\partial_t\int\rho_t^{(\gamma)}(\mathbf{x})d\mathbf{x} = 0$. Therefore,

$$\partial_tZ_t^{(\gamma)} = \partial_t\int\tilde{\rho}_t^{(\gamma)}(\mathbf{x})d\mathbf{x} = Z_t^{(\gamma)}\int\alpha_t(\mathbf{x})d\rho_t(\mathbf{x}).$$

Therefore,

$$\partial_t\rho_t^{(\gamma)} = -\nabla\cdot(\rho_t^{(\gamma)}\boldsymbol{v}_t^{(\gamma)}) + \rho_t^{(\gamma)}\left(\alpha_t - \int\alpha_td\rho_t\right).$$

We have obtained our desired PDE, which concludes the proof. □

## A.2 Properties of the Semidual Loss

Since we are interested in the semidiscrete case, the optimal transport plan always admits a density with respect to the product measure $\mu\otimes\nu$, regardless of entropic regularization. Therefore, we restrict the optimization problem to those that admit a density, and abuse the notation by using $\pi$ as the density of a coupling with respect to $\mu\otimes\nu$. Recall $d\pi_{\varepsilon,\mathbf{g}}(\mathbf{x},\mathbf{y}_j) = s_{\varepsilon,\mathbf{g}}(\mathbf{x})_j$. Therefore, we can rewrite $\mathcal{C}_{c,\varepsilon}(\pi)$ as

$$\mathcal{C}_\varepsilon(\pi_{\varepsilon,\mathbf{g}}) = \sum_{j=1}^N\int c(\mathbf{x},\mathbf{y}_j)s_{\varepsilon,\mathbf{g}}(\mathbf{x})_j d\mu(\mathbf{x}) + \varepsilon\sum_{j=1}^N\int s_{\varepsilon,\mathbf{g}}(\mathbf{x})_j\log\left(\frac{s_{\varepsilon,\mathbf{g}}(\mathbf{x})_j}{b_j}\right)d\mu(\mathbf{x}).$$

We first relate the dual objective $F_\varepsilon$ to the distance between $\nu_\varepsilon$ and $\nu$.

**Lemma 7.** *For any* $\mathbf{g}\in\mathbb{R}^N$ *and* $\varepsilon\geq 0$, *we have* $\nabla F_\varepsilon(\mathbf{g}) = \mathbf{b} - \mathbf{m}(\mathbf{g})$ *(we use subgradient for* $\varepsilon = 0$*). As a result,*

$$\chi^2(\mathbf{m}(\mathbf{g})\,\|\,\mathbf{b}) = \sum_{j=1}^N\frac{1}{b_j}(\partial_{g_j}F_\varepsilon(\mathbf{g}))^2 \quad \text{and} \quad \mathrm{TV}(\mathbf{m}(\mathbf{g}),\nu) = \frac{1}{2}\|\nabla F_\varepsilon(\mathbf{g})\|_1.$$

*In particular, for* $\mathbf{b} = \mathbf{1}_N/N$ *we have* $\chi^2(\nu_\varepsilon[\mathbf{g}]\,\|\,\mathbf{b}) = N\|\nabla F_\varepsilon(\mathbf{g})\|^2$.

Recall the Pinsker inequality $\mathrm{TV}\leq\sqrt{\frac{1}{2}\mathrm{KL}}$ and the fact that $\mathrm{KL}\leq\log(1+\chi^2)$. Thus we can always upper bound total variation using the chi-squared divergence.

*Proof of Lemma 7.* Note that the Chi-Squared and total variation bounds follow immediately from

$$\partial_{g_j}F_\varepsilon(\mathbf{g}) = b_j - m(\mathbf{g})_j, \tag{17}$$

since then

$$\sum_{j=1}^N\frac{1}{b_j}(\partial_{g_j}F_\varepsilon(\mathbf{g}))^2 = \sum_{j=1}^N\left(\frac{m(\mathbf{g})_j}{b_j}\right)^2 b_j - 1 = \chi^2(\mathbf{m}(\mathbf{g})\,\|\,\mathbf{b}),$$

and

$$\|\nabla F_\varepsilon(\mathbf{g})\|_1 = \sum_{j=1}^{N} |m(\mathbf{g})_j - b_j| = 2\mathrm{TV}(\mathbf{m}(\mathbf{g}), \mathbf{b}).$$

Thus, we turn to showing (17). For $\varepsilon > 0$,

$$\partial_{g_j} F_\varepsilon(\mathbf{g}) = b_j - \frac{e^{\frac{g_j - c(\mathbf{x}, \mathbf{y}_j)}{\varepsilon}} b_j}{\sum_k e^{\frac{g_k - c(\mathbf{x}, \mathbf{y}_k)}{\varepsilon}} b_k} = b_j - \int s_{\varepsilon,\mathbf{g}}(\mathbf{x})_j \mathrm{d}\mu(\mathbf{x}) = b_j - m(\mathbf{g})_j.$$

In fact, one could show the above identity directly using duality and the envelope theorem, as $\mathbf{b} - \mathbf{m}(\mathbf{g}) = \mathbf{0}$ is a constraint of the minimization problem.

For $\varepsilon = 0$, we similarly have

$$\partial_{g_j} F_0(\mathbf{g}) = b_j - \int \frac{\mathbb{1}[j \in \arg\max_l g_l - c(\mathbf{x}, \mathbf{y}_l)]}{\sum_{k=1}^{N} \mathbb{1}[k \in \arg\max_l g_l - c(\mathbf{x}, \mathbf{y}_l)]} \mathrm{d}\mu(\mathbf{x}) = b_j - \int s_{0,\mathbf{g}}(\mathbf{x})_j \mathrm{d}\mu(\mathbf{x}),$$

which completes the proof.

$\square$

The following allows us to turn an approximate stationary point (maximizer) of the dual into an approximate optimal transport plan.

**Lemma 8.** *For any* $\mathbf{g} \in \mathbb{R}^N$ *we have* $\mathcal{C}_\varepsilon(\pi_{\varepsilon,\mathbf{g}}) \leq \mathcal{C}_\varepsilon^* + \|\mathbf{g}\|\|\nabla F_\varepsilon(\mathbf{g})\|.$

*Proof.* Note that since $\nu$ is a discrete measure, any coupling with the correct marginals admits a density with respect to $\mu \otimes \nu$. Hence, we can use the change of variables $\mathrm{d}\pi(\mathbf{x}, \mathbf{y}) = \mathbf{s}(\mathbf{x})\mathrm{d}\mu(\mathbf{x})$. Then, we can write the OT cost as

$$\mathcal{C}_{c,\varepsilon}^* = \min_{\substack{\mathbf{s}(\mathbf{x}) \geq \mathbf{0} \\ \int \mathbf{s}(\mathbf{x})\mathrm{d}\mu(\mathbf{x}) = \mathbf{b} \\ \sum_{j=1}^{N} s(\mathbf{x})_j = 1}} \sum_{j=1}^{N} \int s(\mathbf{x})_j \left[ c(\mathbf{x}, \mathbf{y}_j) + \varepsilon \log\left(\frac{s(\mathbf{x})_j}{b_j}\right) - \varepsilon \right] \mathrm{d}\mu(\mathbf{x}) + \varepsilon,$$

.

When $\varepsilon > 0$, we can drop the constraint $\pi \geq 0$ as it is satisfied by entropy regularization. The Lagrangian is then given by

$$\mathcal{L}_\varepsilon(\mathbf{s}, f, \mathbf{g}) := \sum_{j=1}^{N} \int s(\mathbf{x})_j \left[ c(\mathbf{x}, \mathbf{y}_j) + \varepsilon \log\left(\frac{s(\mathbf{x})_j}{b_j}\right) - \varepsilon \right] \mathrm{d}\mu(\mathbf{x}) + \varepsilon$$

$$+ \int f(\mathbf{x})\left(1 - \sum_{j=1}^{N} s(\mathbf{x})_j\right)\mathrm{d}\mu(\mathbf{x}) + \left\langle \mathbf{g}, \mathbf{b} - \int \mathbf{s}(\mathbf{x})\mathrm{d}\mu(\mathbf{x}) \right\rangle.$$

Maximizing over $\mathbf{s}$ yields $s(\mathbf{x})_j = e^{(f(\mathbf{x}) + g_j - c(\mathbf{x}, \mathbf{y}_j))/\varepsilon}$, and leads to the following dual problem

$$\max_{f,g} \mathcal{D}_\varepsilon(f, \mathbf{g}) := \int f(\mathbf{x})\mathrm{d}\mu(\mathbf{x}) + \langle \mathbf{g}, \mathbf{b} \rangle - \varepsilon \sum_{j=1}^{N} \int \exp\left(\frac{f(\mathbf{x}) + g_j - c(\mathbf{x}, \mathbf{y}_j)}{\varepsilon}\right) b_j \mathrm{d}\mu(\mathbf{x}) + \varepsilon,$$

This dual is maximized by

$$f_{\mathbf{g},\varepsilon}(\mathbf{x}_0) = -\varepsilon \log \sum_{j=1}^{N} \exp\left(\frac{g_j - c(\mathbf{x}, \mathbf{y}_j)}{\varepsilon}\right) b_j.$$

Note that at $f = f_{\mathbf{g},\varepsilon}$, $s_{f,\mathbf{g},\mathbf{g}}$ coincides with $s_{\varepsilon,\mathbf{g}}$ of (7). The semidual is defined as $F_\varepsilon(\mathbf{g}) := \max_f \mathcal{D}_\varepsilon(f, \mathbf{g})$, and given by

$$F_\varepsilon(\mathbf{g}) = \langle \mathbf{g}, \mathbf{b} \rangle - \varepsilon \int \left[ \log \sum_{j=1}^{N} \exp\left(\frac{g_j - c(\mathbf{x}, \mathbf{y}_j)}{\varepsilon}\right) b_j \right] \mathrm{d}\mu(\mathbf{x}).$$

for $\varepsilon > 0$. Thus, we derived the semidual ($S_\varepsilon$) for $\varepsilon > 0$. Importantly, one can verify

$$F_\varepsilon(\mathbf{g}) = \mathcal{L}_{c,\varepsilon}(\mathbf{s}_{\varepsilon,\mathbf{g}}, f_{\mathbf{g},\varepsilon}, \mathbf{g}) = \mathcal{C}_\varepsilon(\pi_{\varepsilon,\mathbf{g}}) + \left\langle \mathbf{g}, \mathbf{b} - \int \mathbf{s}_{\varepsilon,\mathbf{g}}(\mathbf{x})\mathrm{d}\mu(\mathbf{x}) \right\rangle. \tag{18}$$

We now turn to the case of $\varepsilon = 0$. The Kantorovich dual in this case is given by

$$\max_{\{f,g\,:\,f\oplus g\leq c\}} \mathcal{D}_0(f,g) := \int f(\mathbf{x})\mathrm{d}\mu(\mathbf{x}) + \langle \mathbf{g}, \mathbf{b} \rangle$$

when $\varepsilon = 0$, where $f \oplus g \leq c$ means $f(\mathbf{x}) + g_j \leq c(\mathbf{x}, \mathbf{y}_j)$ for $(\mu \otimes \nu)$-a.e. $(\mathbf{x}, \mathbf{y}_j)$. This is achieved by

$$f_{\mathbf{g},0}(\mathbf{x}) = \min_j c(\mathbf{x}, \mathbf{y}_j) - g_j.$$

This yields the semidual

$$F_0(\mathbf{g}) = \langle \mathbf{g}, \mathbf{b} \rangle + \int [\min_j c(\mathbf{x}, \mathbf{y}_j) - g_j]\mathrm{d}\mu(\mathbf{x}).$$

We once again derived the semidual ($S_\varepsilon$). Using the definition of $\pi_{0,\mathbf{g}}$, we have

$$F_0(\mathbf{g}) = \langle \mathbf{g}, \mathbf{b} \rangle + \sum_{j=1}^N \int [c(\mathbf{x}, \mathbf{y}_j) - g_j]s_{0,\mathbf{g}}(\mathbf{x})_j\mathrm{d}\mu(\mathbf{x})$$

$$= \sum_{j=1}^N \int c(\mathbf{x}, \mathbf{y}_j)s_{0,\mathbf{g}}(\mathbf{x})_j\mathrm{d}\mu(\mathbf{x}) + \left\langle \mathbf{g}, \mathbf{b} - \int \mathbf{s}_{0,\mathbf{g}}(\mathbf{x})\mathrm{d}\mu(\mathbf{x}) \right\rangle.$$

Once again, we obtain

$$F_0(\mathbf{g}) = \mathcal{C}_0(\pi_{0,\mathbf{g}}) + \left\langle \mathbf{g}, \mathbf{b} - \int \mathbf{s}_{\varepsilon,\mathbf{g}}(\mathbf{x})\mathrm{d}\mu(\mathbf{x}) \right\rangle,$$

i.e. we can extend (18) to $\varepsilon = 0$. By Lemma 7, for any $\varepsilon \geq 0$ we have

$$\nabla F_\varepsilon(\mathbf{g}) = \mathbf{b} - \int \mathbf{s}_{\varepsilon,\mathbf{g}}\mathrm{d}\mu(\mathbf{x}_0).$$

The above formula directly implies the total variation bound. Therefore,

$$F_\varepsilon(\mathbf{g}) = \mathcal{C}_\varepsilon(\pi_{\varepsilon,\mathbf{g}}) + \langle \mathbf{g}, \nabla F_\varepsilon(\mathbf{g}) \rangle.$$

By duality, we additionally have $F_\varepsilon(\mathbf{g}) \leq \mathcal{C}_\varepsilon^*$. Combined with the Cauchy-Schwartz inequality, we obtain $\mathcal{C}_\varepsilon(\pi_{\varepsilon,\mathbf{g}}) \leq \mathcal{C}_\varepsilon^* + \|\mathbf{g}\|\|\nabla F_\varepsilon(\mathbf{g})\|$, completing the proof. $\qquad\square$

### A.3 Convergence of SGD

We begin by recalling the following classical result from convex optimization, and we include its proof for completeness.

**Proposition 9.** *Suppose $f : \mathbb{R}^N \to \mathbb{R}$ is a stochastic convex function with subgradient $\nabla f$, and there exists a constant $G$ such that $\mathbb{E}[\|\nabla f(\mathbf{g})\|^2] \leq G^2$ for all $\mathbf{g} \in \mathbb{R}^N$. Define $F : \mathbb{R}^N \to \mathbb{R}$ via $F(\cdot) = \mathbb{E}[f(\cdot)]$, where expectation is over the stochasticity of $f$. Let $\mathbf{g}^*$ denote any minimizer of $F$, and $F^* = F(\mathbf{g}^*)$. Let $\{\mathbf{g}_t\}_{t=0}^{T-1}$ denote the iterates of SGD with the sequence of step size $\{\eta_t\}_{t=0}^{T-1}$. Define $\bar{\mathbf{g}}_T = \frac{\sum_{t=0}^{T-1} \eta_t \mathbf{g}_t}{\sum_{t=0}^{T-1} \eta_t}$. Then,*

$$\mathbb{E}[\|\mathbf{g}_t - \mathbf{g}^*\|^2] \leq \|\mathbf{g}_0 - \mathbf{g}^*\|^2 + G^2 \sum_{t=0}^{T-1} \eta_t^2, \quad \forall t \in [T].$$

*If additionally $F$ is $L$-smooth, i.e. $\|\nabla^2 F\|_{\mathrm{op}} \leq L$. Then*

$$\frac{\sum_{t=0}^{T-1} \eta_t \mathbb{E}[\|\nabla F(\mathbf{g}_t)\|^2]}{\sum_{t=0}^{T-1} \eta_t} \leq \frac{F(\mathbf{g}_0) - F^*}{\sum_{t=0}^{T-1} \eta_t} + \frac{\sum_{t=0}^{T-1} \eta_t^2 LG^2}{2\sum_{t=0}^{T-1} \eta_t}.$$

*Proof.* By the convexity of $F$, for any $\mathbf{g}$ we have $F^* \geq F(\mathbf{g}) + \nabla F(\mathbf{g})^\top(\mathbf{g}^* - \mathbf{g})$. With this property, for any $l$ we can write

$$
\begin{aligned}
\mathbb{E}[\|\mathbf{g}_{l+1} - \mathbf{g}^*\|^2 \mid \mathbf{g}_t] &= \|\mathbf{g}_t - \mathbf{g}^*\|^2 - 2\eta_l \mathbb{E}[\nabla f(\mathbf{g}_l) \mid \mathbf{g}_l]^\top(\mathbf{g}_l - \mathbf{g}^*) + \eta_t^2 \mathbb{E}[\|\nabla f(\mathbf{g}_l)\|^2 \mid \mathbf{g}_l] \\
&= \|\mathbf{g}_l - \mathbf{g}^*\|^2 - 2\eta_l \nabla F(\mathbf{g}_l)^\top(\mathbf{g}_l - \mathbf{g}^*) + \eta_l^2 \mathbb{E}[\|\nabla f(\mathbf{g}_l)\|^2 \mid \mathbf{g}_t] \\
&\leq \|\mathbf{g}_l - \mathbf{g}^*\|^2 + 2\eta_l(F^* - F(\mathbf{g}_l)) + \eta_l^2 \mathbb{E}[\|\nabla f(\mathbf{g}_l)\|^2 \mid \mathbf{g}_l].
\end{aligned}
$$

Taking an expectation over $\mathbf{g}_l$, we obtain

$$
\mathbb{E}[\|\mathbf{g}_{l+1} - \mathbf{g}^*\|^2] \leq \mathbb{E}[\|\mathbf{g}_l - \mathbf{g}^*\|^2] - 2\eta_l \mathbb{E}[F^* - F(\mathbf{g}_l)] + \eta_l^2 G^2.
$$

By summing both sides from $l = 0$ to $t - 1$, we have

$$
\begin{aligned}
\mathbb{E}[\|\mathbf{g}_{t+1} - \mathbf{g}^*\|^2] &\leq \|\mathbf{g}_0 - \mathbf{g}^*\|^2 + \sum_{l=0}^{t-1} \eta_l \mathbb{E}[F^* - F(\mathbf{g}_l)] + G^2 \sum_{l=0}^{t-1} \eta_l^2 \\
&\leq \|\mathbf{g}_0 - \mathbf{g}^*\|^2 + G^2 \sum_{l=0}^{T-1} \eta_l^2,
\end{aligned}
$$

which proves the first inequality.

For the second inequality, we start with the smoothness lemma

$$
F(\mathbf{g}_{t+1}) \leq F(\mathbf{g}_t) + \nabla F(\mathbf{g}_t)^\top(\mathbf{g}_{t+1} - \mathbf{g}_t) + \frac{L}{2}\|\mathbf{g}_{t+1} - \mathbf{g}_t\|^2.
$$

Taking expectation conditioned on $\mathbf{g}_t$, we obtain

$$
\mathbb{E}[F(\mathbf{g}_{t+1}) \mid \mathbf{g}_t] \leq F(\mathbf{g}_t) - \nabla\|\nabla F(\mathbf{g}_t)\|^2 + \frac{\eta_t^2 L}{2}\mathbb{E}[\|\nabla f(\mathbf{g}_t)\|^2 \mid \mathbf{g}_t].
$$

Taking another expectation, we obtain

$$
\mathbb{E}[F(\mathbf{g}_{t+1})] - \mathbb{E}[F(\mathbf{g}_t)] \leq -\eta_t \mathbb{E}[\|\nabla F(\mathbf{g}_t)\|]^2 + \frac{\eta_t^2 L G^2}{2}.
$$

By rearranging the terms, summing over $t$ from 0 to $T - 1$, and noticing that $F^* \leq \mathbb{E}[F(\mathbf{g}_T)]$, we obtain

$$
\sum_{t=0}^{T-1} \eta_t \mathbb{E}[\|\nabla F(\mathbf{g}_t)\|^2] \leq F(\mathbf{g}_0) - F^* + \frac{1}{2}LG^2 \sum_{t=0}^{T-1} \eta_t^2,
$$

which completes the proof. $\square$

To apply Proposition 9, we need to estimate the smoothness constant of the dual, achieved by the following lemma.

**Lemma 10.** *Consider the semidual $F_\varepsilon$ defined in ($S_\varepsilon$). For any cost $c : \mathbb{R}^d \times \mathbb{R}^d \to \mathbb{R}$ and any $\varepsilon > 0$, we have $\left\|\nabla^2 F_\varepsilon\right\|_{\mathrm{op}} \leq 1/\varepsilon$. Furthermore, if $c(\mathbf{x}, \mathbf{y}) = -\mathbf{x}^\top\mathbf{y}$ and $\mu$ admits a density, then for $\varepsilon = 0$ we have $\left\|\nabla^2 F_0\right\|_{\mathrm{op}} \leq 2C_\mu^{\max}/\delta$, where $\delta = \min_{i \neq j}\left\|\mathbf{x}_1^{(i)} - \mathbf{x}_1^{(j)}\right\|$, and $C_\mu^{\max}$ is the maximum $\mu$-surface area of any convex set in $\mathbb{R}^d$.*

*Proof.* By Lemma 7, we have

$$
\nabla F_\varepsilon(\mathbf{g}) = \mathbf{b} - \int \mathbf{s}_\varepsilon(\mathbf{g}, \mathbf{x}_0)\mathrm{d}\mu(\mathbf{x}_0),
$$

where we notice that for $\varepsilon = 0$ we can write

$$
s_{0,\mathbf{g}}(\mathbf{x}_0)_i = \mathbb{1}[g_i + \mathbf{x}^\top\mathbf{y}_i \geq g_j + \mathbf{x}^\top\mathbf{y}_j, \,\forall j],
$$

since $\mu$ admits density, and consequently, ties in the maximum occur with probability zero. Thus, for $\varepsilon > 0$, by a direct calculation we obtain

$$
\nabla^2 F_\varepsilon(\mathbf{g}) = \frac{1}{\varepsilon}\int [\mathbf{s}_{\varepsilon,\mathbf{g}}(\mathbf{x})\mathbf{s}_{\varepsilon,\mathbf{g}}(\mathbf{x})^\top - \mathrm{diag}(\mathbf{s}_{\varepsilon,\mathbf{g}}(\mathbf{x}))]\mathrm{d}\mu(\mathbf{x}).
$$

Using this calculation, one immediately obtains $\left\|\nabla^2 F_\varepsilon\right\|_{\mathrm{op}} \leq \frac{1}{\varepsilon}$, which in fact does not depend on the cost function (the argument above holds for any other cost $c$).

For $\varepsilon = 0$, by Kitagawa et al. (2019, Theorem 1.3), we have

$$\nabla^2 F_0(\mathbf{g}) = \boldsymbol{W} - \text{diag}(\boldsymbol{d}),$$

where $d_i = \sum_{j \neq i} W_{ij}$

$$W_{ii} = 0, \quad \text{and}, \quad W_{ij} = \frac{1}{\|\mathbf{y}_i - \mathbf{y}_j\|} \int_{A(\mathbf{g},i) \cap A(\mathbf{g},j)} \mu(\mathbf{x}_0) dS(\mathbf{x}), \forall i \neq j$$

where $A(\mathbf{g}, i) = \{\mathbf{x} : g_i + \mathbf{x}^\top \mathbf{y}_i \geq g_j + \mathbf{x}^\top \mathbf{y}_j, \forall j\}$ and $dS$ denotes the $d - 1$-dimensional surface measure in a Euclidean space. Note that $-\nabla^2 F_0$ is the Laplacian of a weighted graph with weights $W_{ij} \geq 0$, therefore $\left\| \nabla^2 F_0(\mathbf{g}) \right\| \leq \max_{i=1}^N \sum_{j \neq i} W_{ij}$. Further,

$$\sum_{j \neq i} W_{ij} = \sum_{j \neq i} \frac{1}{\|\mathbf{y}_i - \mathbf{y}_j\|} \int_{A(\mathbf{g},i) \cap A(\mathbf{g},j)} \mu(\mathbf{x}) dS(\mathbf{x}) \leq \frac{1}{\delta} \int_{\partial A(\mathbf{g},i)} \mu(\mathbf{x}) dS(\mathbf{x}) \leq \frac{C_\mu^{\max}}{\delta},$$

which finishes the proof. $\qquad\square$

We are now in a position to state the proof of Theorem 2.

*Proof of Theorem 2.* Let $F = -F_\varepsilon$ in Proposition 9. We begin by noting that for constant learning rate $\eta_k = \eta := \sqrt{\frac{\Delta}{L_\varepsilon K}}$ and $t \sim \text{Unif}(\{0, \ldots, K-1\})$, we can interpret the gradient norm bound of Proposition 9 as

$$\mathbb{E}[\|\nabla F_\varepsilon(\mathbf{g}_t)\|^2] \leq \frac{\Delta}{\eta K} + \frac{\eta L_\varepsilon G^2}{2},$$

where we recall from Lemma 10 that $\left\| \nabla^2 F_\varepsilon \right\|_{\text{op}} \leq L_\varepsilon$. Moreover, let

$$\nabla \hat{F}_\varepsilon(\mathbf{g}) = \mathbf{b} - \frac{1}{B} \sum_{i=1}^B \mathbf{s}_{\varepsilon,\mathbf{g}}(\mathbf{x}^{(i)}),$$

denote the stochastic gradient where $(\mathbf{x}^{(i)}) \overset{\text{i.i.d.}}{\sim} \mu$. We can define the probability distribution

$$\hat{\mathbf{m}}(\mathbf{g}) := \frac{1}{B} \sum_{i=1}^B \mathbf{s}_{\varepsilon,\mathbf{g}}(\mathbf{x}_0^{(i)}) \in \Delta^N.$$

Then,

$$\left\| \nabla \hat{F}_\varepsilon(\mathbf{g}) \right\| \leq \left\| \nabla \hat{F}_\varepsilon(\mathbf{g}) \right\|_1 = 2\text{TV}(\mathbf{b}, \hat{\mathbf{m}}(\mathbf{g})) \leq 2.$$

As a result, $G = 2$ in Proposition 9, and we obtain

$$\mathbb{E}[\|\nabla F_\varepsilon(\mathbf{g}_t)\|^2] \leq 3\sqrt{\frac{L_\varepsilon \Delta}{K}}. \qquad (19)$$

Let $\nu_{\min} := \min_j b_j$ for simplicity. Combined with Lemma 7, we conclude that

$$\chi^2(\mathbf{m}(\mathbf{g}) \| \mathbf{b}) \leq \frac{1}{\nu_{\min}} \|\nabla F_\varepsilon(\mathbf{g})\|^2 \leq \frac{3}{\nu_{\min}} \cdot \sqrt{\frac{L_\varepsilon \Delta}{K}},$$

which concludes the first part of the proof.

For the second part, we use Lemma 8 and the Cauchy-Schwartz inequality to obtain

$$\mathbb{E}[\mathcal{C}_\varepsilon(\pi_{\varepsilon,\mathbf{g}_t})] \leq \mathcal{C}_\varepsilon^* + \mathbb{E}[\|\mathbf{g}_t\|^2]^{1/2} \mathbb{E}[\|\nabla F_\varepsilon(\mathbf{g}_t)\|^2]^{1/2}.$$

From Proposition 9, we have for any fixed $k \in [K]$

$$\mathbb{E}[\|\mathbf{g}_k\|^2] \leq \|\mathbf{g}^\star\|^2 + K\eta^2 G^2 \leq \|\mathbf{g}^\star\|^2 + \frac{4\Delta}{L_\varepsilon}.$$

As a result, the above bound also holds for $\mathbb{E}[\|\mathbf{g}_t\|]$. Combining this with (19) the proof. $\qquad\square$

# B  ALGORITHMS

We provide the details of all algorithms introduced in the paper in this section.

We begin by detailing the optimization algorithm for solving SD-OT. If $\eta_k$ only depends on $k$, Algorithm 1 is SGD on the semidual loss ($S_\varepsilon$). For SGD, the typical choice of learning rate is $\eta_k \propto 1/\sqrt{k}$ after an optional constant learning rate phase. By allowing $\eta_k$ to depend on the gradients, Algorithm 1 becomes adaptive stochastic optimization on the semidual loss. We found AdaGrad (Duchi et al., 2011) to be a particularly effective choice of learning rate schedule, and provide details on the schedule of SGD and Adagrad in Appendix C.

---

**Algorithm 1** SOLVESDOT

> **Input**: data $\mathbf{y}_1, \ldots, \mathbf{y}_N \in \mathbb{R}^d, \mathbf{b} \in \Delta^N, \varepsilon \geq 0$, threshold $\tau$, step sizes $\eta_t$, batch size $M$.
> **Output**: Semidiscrete dual potential $\boldsymbol{g}$.
> 1: $\mathbf{g} \leftarrow \mathbf{0}, \bar{\mathbf{g}} \leftarrow \mathbf{0}, k \leftarrow 0$.
> 2: **while** $\widehat{\chi}^2(\mathbf{m}(\mathbf{g})|\mathbf{b}) > \tau$ **do**
> 3:      Sample $\mathbf{x}_0, \ldots, \mathbf{x}_M \overset{\text{i.i.d}}{\sim} \mu_0^{\otimes M}$
> 4:      $\mathbf{g} \leftarrow \mathbf{g} + \eta_k \left( \mathbf{b} - \frac{1}{M} \sum_{\ell=1}^M s_{\varepsilon, \mathbf{g}}(\mathbf{x}_\ell) \right)$
> 5:
> 6:      $\bar{\mathbf{g}} \leftarrow \frac{1}{k+1}\mathbf{g} + \frac{k}{k+1}\bar{\mathbf{g}}$
> 7:      $k \leftarrow k + 1$
> 8: **end while**
> 9: **return** $\bar{\mathbf{g}}$

---

Once dual potentials are obtained from Algorithm 1, we can use the following algorithm to pair noise $\mathbf{x}_0$ with data $\mathbf{x}_1$, which can be used to train flow-based models.

---

**Algorithm 2** ASSIGN

> **Input**: noise $\mathbf{x}_0$, data $\mathbf{x}_1^{(1)}, \ldots, \mathbf{x}_1^{(N)}, \mathbf{g}, \varepsilon \geq 0$.
> **Output**: Coupled noise-data pair $(\mathbf{x}_0, \mathbf{x}_1')$
> 1: **return** $\mathbf{x}_1^{(k)}, k \sim s_{\varepsilon, \mathbf{g}}(\mathbf{x}_0) \in \Delta^N$.

---

Next, to better see the contrast between I-FM and OT-FM/SD-FM, we recall the usual I-FM algorithm for training flow models.

---

**Algorithm 3** I-FM

> **Input**: Noise $\mu_0$, Data $\mu_1 = \frac{1}{N} \sum_{j=1}^N \delta_{\mathbf{x}_1^{(j)}}$, FM batch size $B$, flow model $\boldsymbol{v}_\theta(t, \mathbf{x})$, Epochs $E$.
> 1: **for** $\ell = 1, \ldots, NE/B$ **do**                                   $\triangleright NE\Theta$
> 2:      Draw $(\mathbf{x}_0^{(i)})_{i=1}^B \sim \mu_0^{\otimes B}$
> 3:      Draw $(\mathbf{x}_1^{(i)})_{i=1}^B \sim \mu_1^{\otimes B}$
> 4:      $\theta \leftarrow \text{FMSTEP}(\theta, (\mathbf{x}_0^{(i)})_{i=1}^B, (\mathbf{x}_1^{(i)})_{i=1}^B)$.                 $\triangleright \Theta \times B$
> 5: **end for**

---

We now present OT-FM and SD-FM. Steps specific to OT-FM are highlighted in blue, while those specific to SD-FM are highlighted in red.

---

**Algorithm 4** OT-FM

---

1: **Input**: Noise $\mu_0$, Data $\mu_1 = \frac{1}{N}\sum_{j=1}^N \delta_{\mathbf{x}_1^{(j)}}$, FM batch size $B$, flow model $\boldsymbol{v}_\theta(t, \mathbf{x})$, Epochs $E$,
2: Regularization $\varepsilon > 0$, OT batch-size $n$.
3: **for** $\ell = 1, \ldots, NE/n$ **do** $\qquad\qquad\qquad\qquad \triangleright NE\Theta + NE/n \times O(dn^2/\varepsilon^2) = NE\Theta + O(NEn/\varepsilon^2)$
4: $\quad$ Draw $(\mathbf{x}_0^{(i)})_{i=1}^n \sim \mu_0^{\otimes n}$
5: $\quad$ Draw $(\mathbf{x}_1^{(j)})_{j=1}^n \sim \mu_1^{\otimes n}$
6: $\quad$ $\mathbf{P} \leftarrow \textsc{Sinkhorn}((\mathbf{x}_0^{(i)})_{i=1}^n, (\mathbf{x}_1^{(j)})_{j=1}^n, \varepsilon) \in \Delta^{n \times n}$ $\qquad\qquad\qquad \triangleright O(dn^2/\varepsilon^2)$
7: $\quad$ Sample from coupling matrix: $\tilde{\mathbf{x}}_1^{(i)} \leftarrow \mathbf{x}_1^{(k)}, k \sim B\mathbf{P}_{i\cdot}$, for every $i \leq n$. $\quad \triangleright O(n^2)$ (neglected)
8: $\quad$ **for** $k = 0, .., n/B - 1$ **do** $\qquad\qquad\qquad\qquad\qquad\qquad\qquad\qquad \triangleright \Theta \times n$
9: $\qquad$ $\theta \leftarrow \textsc{FMStep}(\theta, (\mathbf{x}_0^{(i)})_{i=kB+1}^{(k+1)B}, (\tilde{\mathbf{x}}_1^{(i)})_{i=kB+1}^{(k+1)B})$. $\qquad\qquad\qquad \triangleright \Theta \times B$
10: $\quad$ **end for**
11: **end for**

---

**Algorithm 5** OT-FM*, Cached Sinkhorn computations.

---

1: **Input**: Noise $\mu_0$, Data $\mu_1 = \frac{1}{N}\sum_{j=1}^N \delta_{\mathbf{x}_1^{(j)}}$, FM batch size $B$, flow model $\boldsymbol{v}_\theta(t, \mathbf{x})$, Epochs $E$,
2: Regularization $\varepsilon > 0$, OT batch-size $n$.
3: **for** $\ell = 1, \ldots, NE/n$ **do** $\qquad\qquad \triangleright O(NE/n \times dn^2/\varepsilon^2) = O(NEdn/\varepsilon^2)$ — Memory: $2NE$
4: $\quad$ Store $X_\ell = (\mathbf{x}_0^{(i)})_{i=1}^n \sim \mu_0^{\otimes n}$ $\qquad\qquad\qquad\qquad \triangleright$ Memory: $n$ (storing rng)
5: $\quad$ Draw $(\mathbf{x}_1^{(j)})_{j=1}^B \sim \mu_1^{\otimes n}$
6: $\quad$ $\mathbf{P} \leftarrow \textsc{Sinkhorn}((\mathbf{x}_0^{(i)})_{i=1}^n, (\mathbf{x}_1^{(j)})_{j=1}^n, \varepsilon) \in \Delta^{n \times n}$ $\qquad\qquad\qquad \triangleright O(dn^2/\varepsilon^2)$
7: $\quad$ Sample from coupling matrix: $k_\ell[i] \leftarrow k \sim B\mathbf{P}_{i\cdot}$, for every $i \leq n$. $\qquad \triangleright O(dn^2)$ — Memory: $n$
8: **end for**
9: **for** $\ell = 1, \ldots, NE/n$ **do** $\qquad\qquad\qquad\qquad\qquad\qquad\qquad\qquad\qquad \triangleright NE\Theta$
10: $\quad$ Load $(\mathbf{x}_0^{(i)})_{i=1}^n = X_\ell$
11: $\quad$ Load paired $\tilde{\mathbf{x}}_1^{(i)} = \mathbf{x}_1^{(k_\ell[i])}$, for $i \leq n$
12: $\quad$ **for** $k = 0, .., n/B - 1$ **do**
13: $\qquad$ $\theta \leftarrow \textsc{FMStep}(\theta, (\mathbf{x}_0^{(i)})_{i=kB+1}^{(k+1)B}, (\tilde{\mathbf{x}}_1^{(i)})_{i=kB+1}^{(k+1)B})$. $\qquad\qquad\qquad \triangleright \Theta \times B$
14: $\quad$ **end for**
15: **end for**

---

**Algorithm 6** SD-FM

---

**Input**: Noise $\mu_0$, Data $\mu_1 = \frac{1}{N}\sum_{j=1}^N \delta_{\mathbf{x}_1^{(j)}}$, FM batch size $B$, flow model $\boldsymbol{v}_\theta(t, \mathbf{x})$, Epochs $E$,
1: Regularization $\varepsilon \geq 0$, Threshold $\tau$
2: $\mathbf{g} \leftarrow \textsc{SolveSDOT}(\mu_0, \mu_1, \varepsilon, \tau)$ $\qquad\qquad\qquad\qquad\qquad\qquad \triangleright NdK$ — Memory: $N$
3: **for** $\ell = 1, \ldots, NE/B$ **do** $\qquad\qquad\qquad\qquad\qquad\qquad\qquad\qquad \triangleright NE(\Theta + Nd)$
4: $\quad$ Draw $(\mathbf{x}_0^{(i)})_{i=1}^B \sim \mu^{\otimes B}$
5: $\quad$ $\tilde{\mathbf{x}}_1^{(i)} \leftarrow \textsc{Assign}(\mathbf{x}_0^{(i)}, \mathbf{g}, \varepsilon)$ for $i \leq B$. $\qquad\qquad\qquad\qquad\qquad \triangleright dNB$
6: $\quad$ $\theta \leftarrow \textsc{FMStep}(\theta, (\mathbf{x}_0^{(i)})_{i=1}^B, (\tilde{\mathbf{x}}_1^{(i)})_{i=1}^B)$. $\qquad\qquad\qquad\qquad\qquad \triangleright \Theta \times B$
7: **end for**

---

## C  EXPERIMENT DETAILS

### C.1  SEMIDISCRETE OPTIMAL TRANSPORT

To solve for the semidiscrete dual potential $\mathbf{g}$ in practice, we implement Algorithm 1 using JAX and utilize built-in data parallelism to scale to handle large datasets over multiple GPUs and nodes. Since the problem ($\text{S}_\varepsilon$) is concave (and strictly, for $\varepsilon > 0$) one can use a variety of different optimizers and obtain a suitable convergence behavior. We use the dot-product cost and scale $\varepsilon$ with the standard deviation of the cost, as proposed by Zhang et al. (2025). In the class-conditional setting, we choose $\beta = 10^2$ for ImageNet-32 and $\beta = 2 \times 10^2$ for ImageNet-64 and PetFace.

In all cases, we opt to use AdaGrad with a constant learning rate for $200\,000$ iterations followed by inverse-square root decay for an additional $100\,000$ iterations and perform iterate averaging over the

final $50\,000$ iterates. To choose the learning rate, we notice that typically $\|\mathbf{g}\|$ is of order $\sqrt{N}$ while $\|\nabla F_\varepsilon(\mathbf{g})\|$ is of order $1/\sqrt{N}$. Thus, we expect $L$, Lipschitz constant of the gradient, to be of the order $1/N$, which is smaller than the conservative estimation $L_\varepsilon$ used in Theorem 2. As Theorem 2 suggests a learning rate that scales with $\sqrt{1/L}$ (the typical scaling under bounded gradient norm) , we find $\sqrt{N}$ to be a suitable heuristic for setting the learning rate. As $N$ for our datasets is of order $10^6$, we use $10^3$ as the initial learning rate to optimize the potential. Once computed, the optimal dual potential $\mathbf{g}^\star$ can be cached and reused. Optimal dual potentials for our ImageNet and PetFace experiments will be made public upon publication.

To estimate $\chi^2(\mathbf{m}(\mathbf{g}) \,\|\, \mathbf{b})$, we use a total of $2^{20}$ noise samples, which we divide into batches of size $2^{13}$ that can further be sharded along multiple devices, and average (10) over these batches. For the ImageNet-32x32 dataset, this calculation takes less than 2 minutes on a node of 8 H100 GPUs.

## C.2 FLOW MODEL TRAINING

| | ImageNet (32x32) | ImageNet (64x64), PetFace |
|---|---|---|
| Channels | 256 | 192 |
| Depth | 3 | 3 |
| Channels multiple | 1,2,2,2 | 1,2,3,4 |
| Heads | 4 | 4 |
| Heads Channels | 64 | 64 |
| Attention resolution | 4 | 8 |
| Dropout | 0.0 | 0.1 |
| Batch size / GPU | 256 | 50 |
| GPUs | 4 | 16 |
| Effective Batch size | 1024 | 800 |
| Epochs | 350 | 575 |
| Effective Iterations | $438\,000$ | $957\,000$ |
| Learning Rate | 0.0001 | 0.0001 |
| Learning Rate Scheduler | Polynomial Decay | Constant |
| Warmup Steps | $20\,000$ | - |

Table 4: Hyperparameters used for flow model training, adapted from Pooladian et al. (2023).

**ImageNet**  In all cases, we use the same U-Net architecture and hyperparameter choices as described in Pooladian et al. (2023, Appendix E) and reproduced in Table 4 for completeness. For SD-FM we sample noise-data pairs using Algorithm 2. Flow matching models are trained on a single 8 x NVIDIA H100 node for 32x32 resolution and two 8 x NVIDIA H100 nodes for 64x64 resolution.

**PetFace**  Images from PetFace are resized to 64x64 resolution. We use the same U-Net architecture and hyperparameter choices as used for ImageNet at 64x64 resolution. Flow matching models are trained on two 8 x NVIDIA H100 nodes.

**CelebA**  Images from CelebA are first rescaled to 64x64 resolution. From each image $\mathbf{x}$, low-resolution, noisy images are constructed by first downscaling to 16x16 (4x SR) or 8x8 (8x SR) and adding Gaussian noise with standard deviation $\sigma = 0.1$ or $0.2$, i.e. $\mathbf{z} = \texttt{downscale}(\mathbf{x}) + \eta$, $\eta = \mathcal{N}(0, \sigma^2\,\mathbf{I})$. The sampled corrupted image $\mathbf{z}$ is treated as the condition paired with $\mathbf{x}$, and we solve conditional optimal transport with $\beta = 25$. Flow matching models are then trained to generate samples of 64x64 images $\mathbf{x}$ conditional on downscaled, noisy observations $\mathbf{z}$. We parameterize the flow with a U-Net with the same hyperparameter choices as for 64x64 ImageNet, and conditions are handled by resizing the low-resolution 8x8 or 16x16 image to 64x64 and stacking together with the input to the U-Net. A batch size of 128 and a learning rate of 0.0002 are used for FM training for a total of $500\,000$ iterations.

## D ADDITIONAL EXPERIMENTAL RESULTS

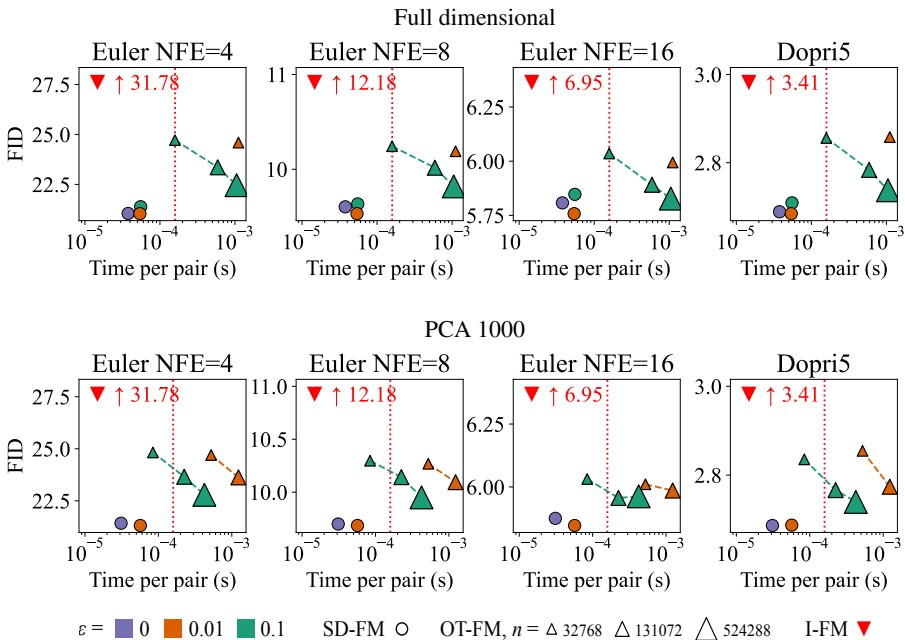

Figure 7: **FID vs total training time for class-conditional ImageNet 32x32.**

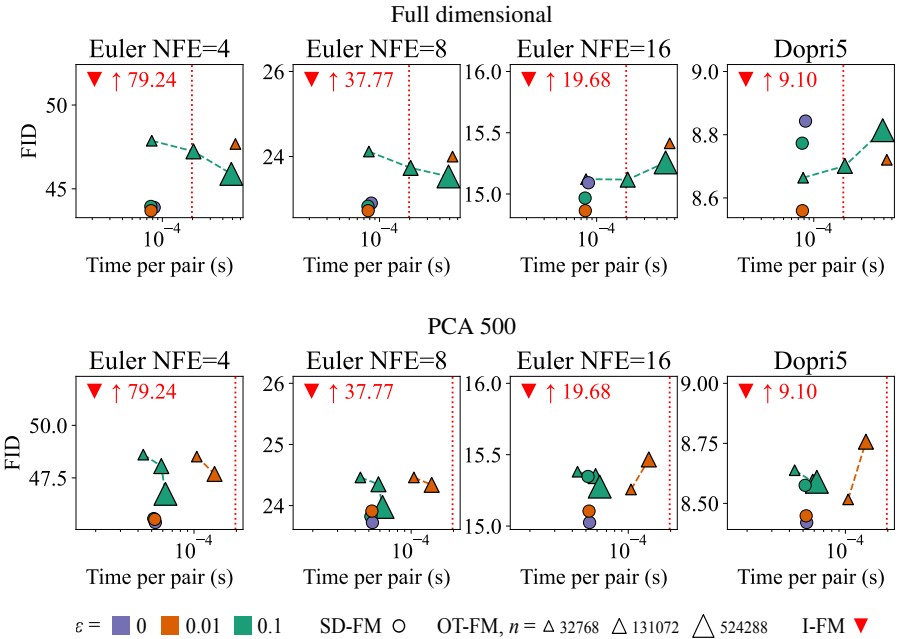

Figure 8: **FID vs time-per-pair for ImageNet 64x64.** We report time-per-pair for training flow matching using SD-FM compared to I-FM and OT-FM with batch sizes $2^{15}, 2^{17}, 2^{19}$.

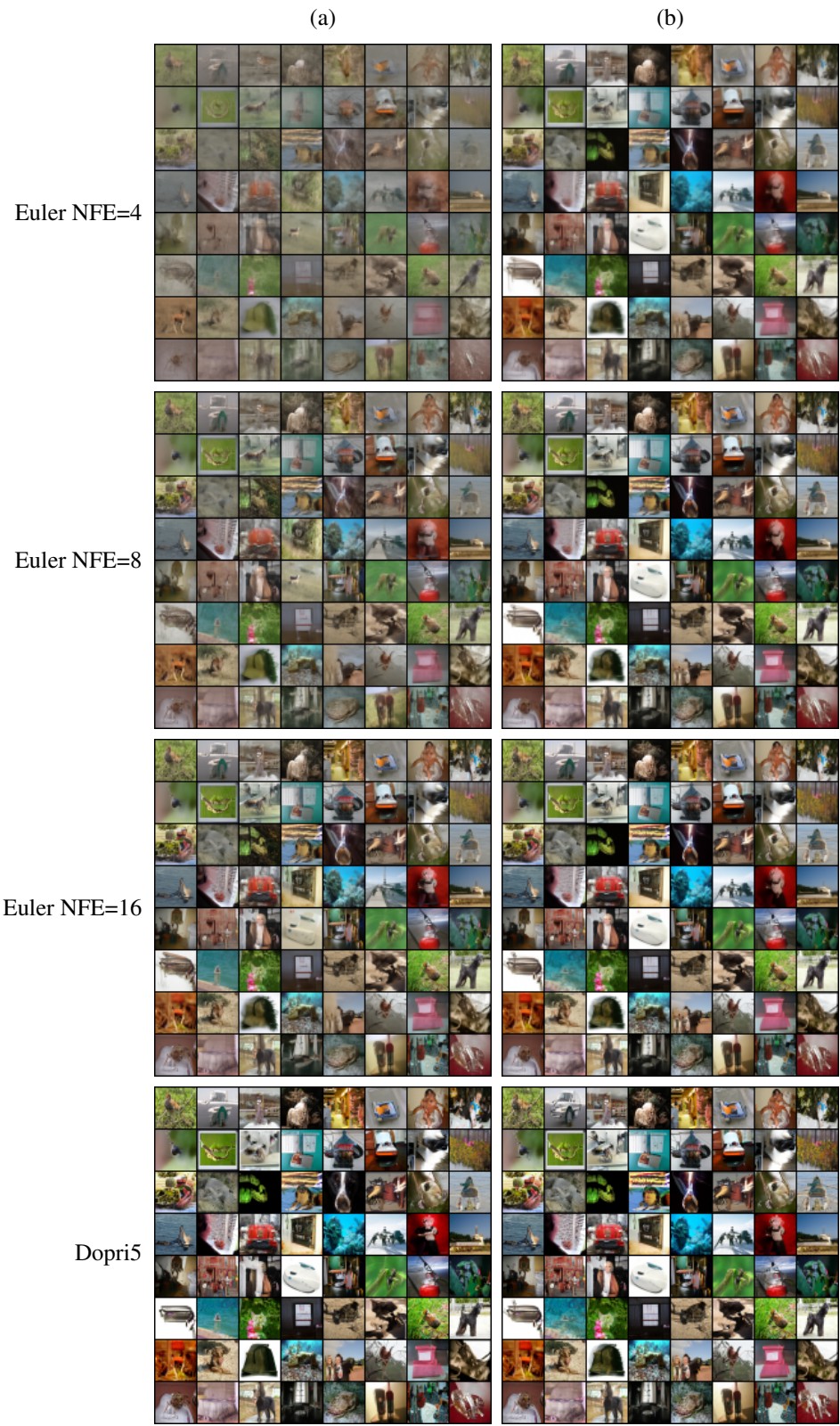

Figure 9: Images generated from unconditional models trained on **ImageNet (32x32)**. (a) denotes independent coupling while (b) denotes semidiscrete OT coupling with $\varepsilon = 0$ used for training the model.

(a)  (b)

Euler NFE=4

Euler NFE=8

Euler NFE=16

Dopri5

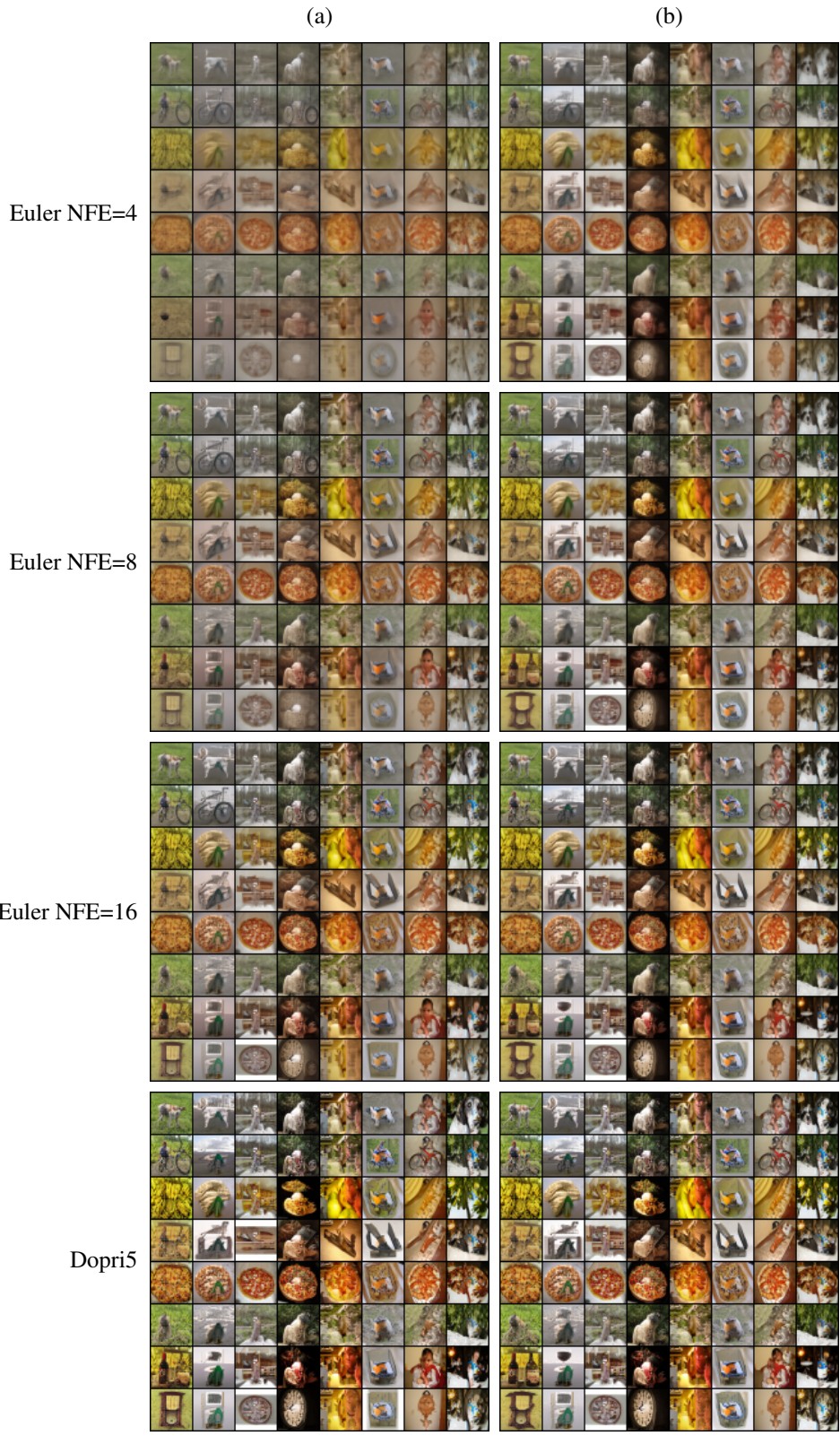

Figure 10: Images generated from conditional models trained on **ImageNet (32x32)**. **(a)** denotes independent coupling while **(b)** denotes semidiscrete OT coupling with $\varepsilon = 0$ used for training the model. Rows are selected classes: English setter, mountain bike, banana, plane, pizza, marmot, red wine, wall clock, cheeseburger, king penguin.

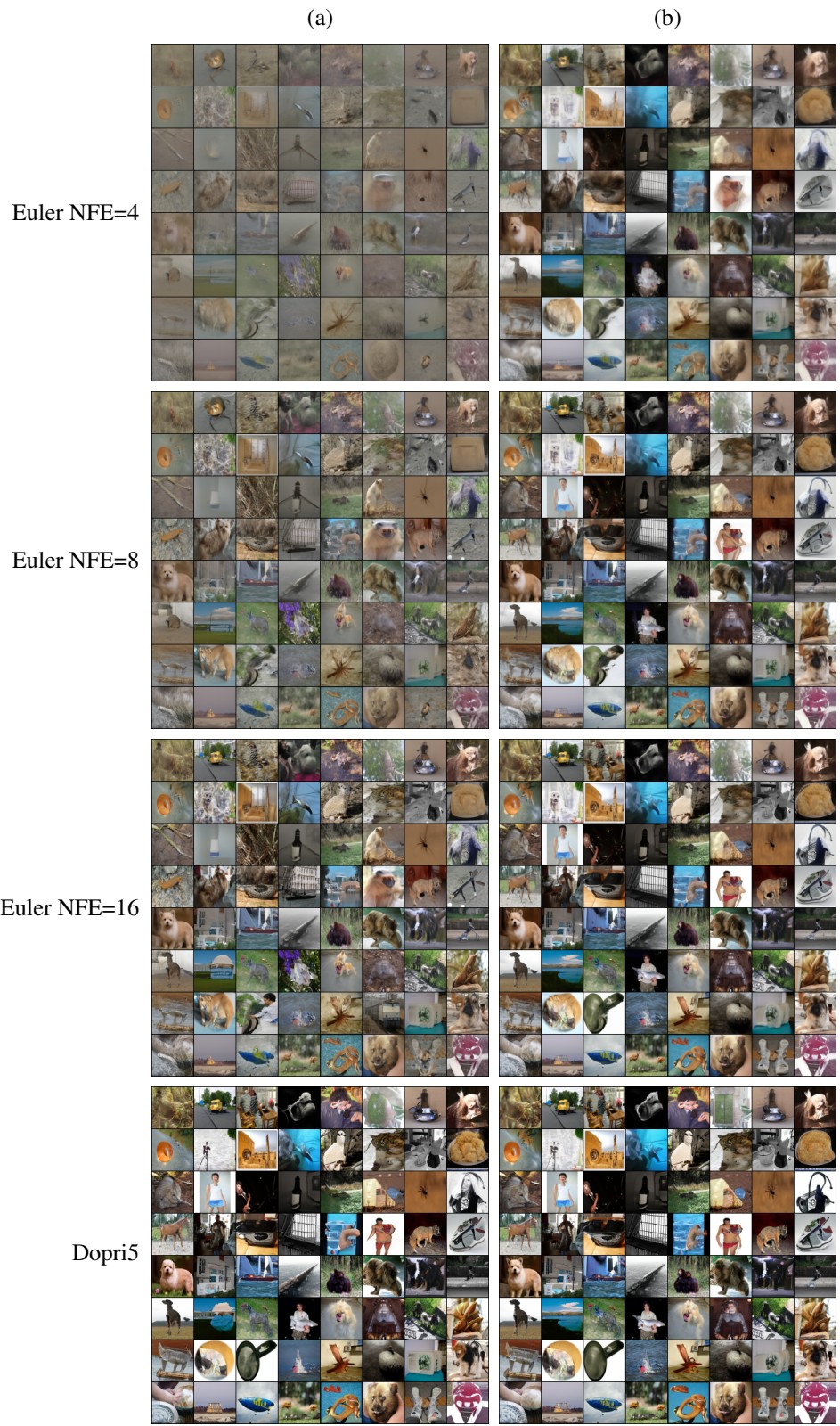

Figure 11: Images generated from unconditional models trained on **ImageNet (64x64)**. **(a)** denotes independent coupling while **(b)** denotes semidiscrete OT coupling with $\varepsilon = 0$ used for training the model.

(a)      (b)

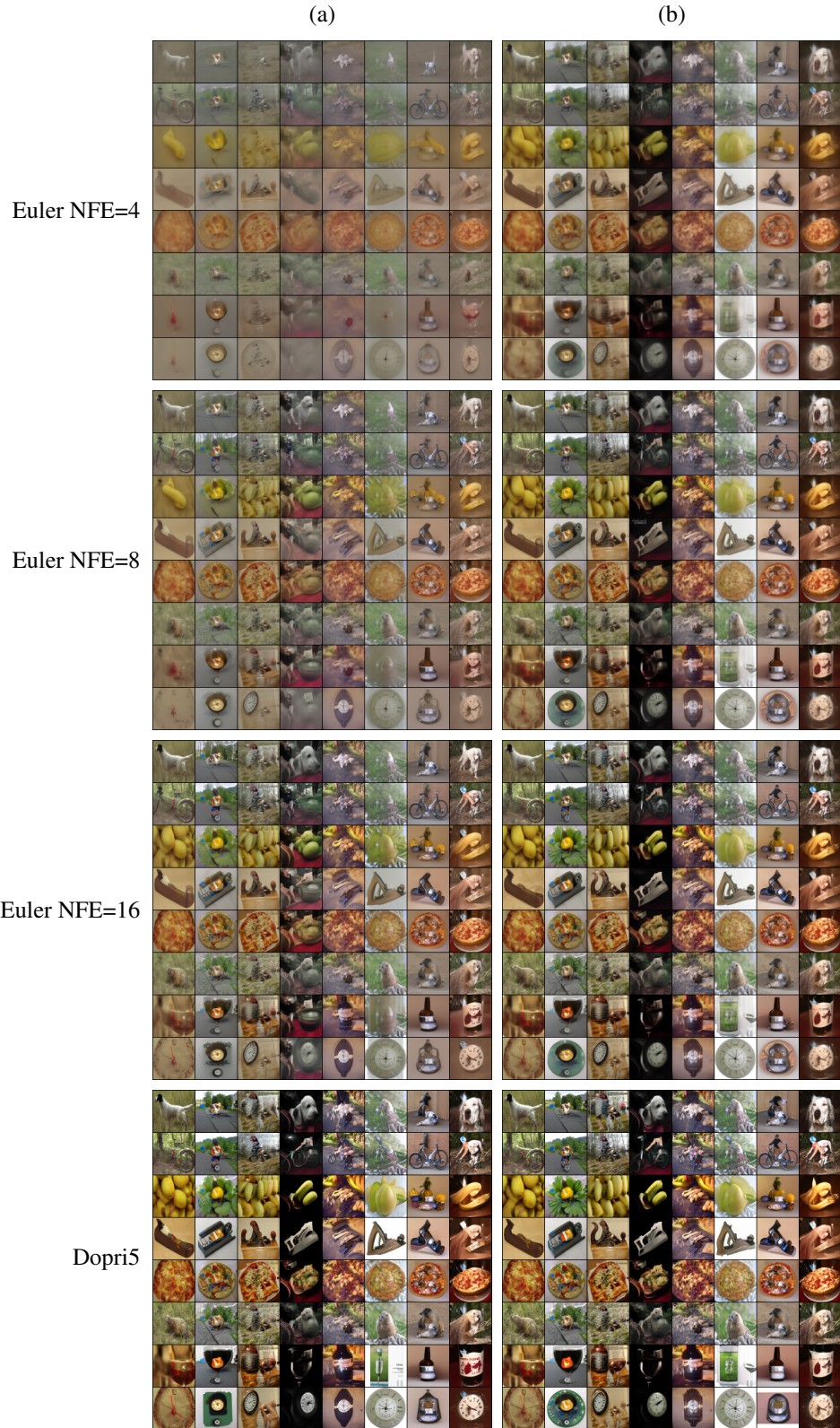

Euler NFE=4

Euler NFE=8

Euler NFE=16

Dopri5

Figure 12: Images generated from conditional models trained on **ImageNet (64x64)**. **(a)** denotes independent coupling while **(b)** denotes semidiscrete OT coupling with $\varepsilon = 0$ used for training the model. Classes are the same as in Figure 10.

(a)  (b)

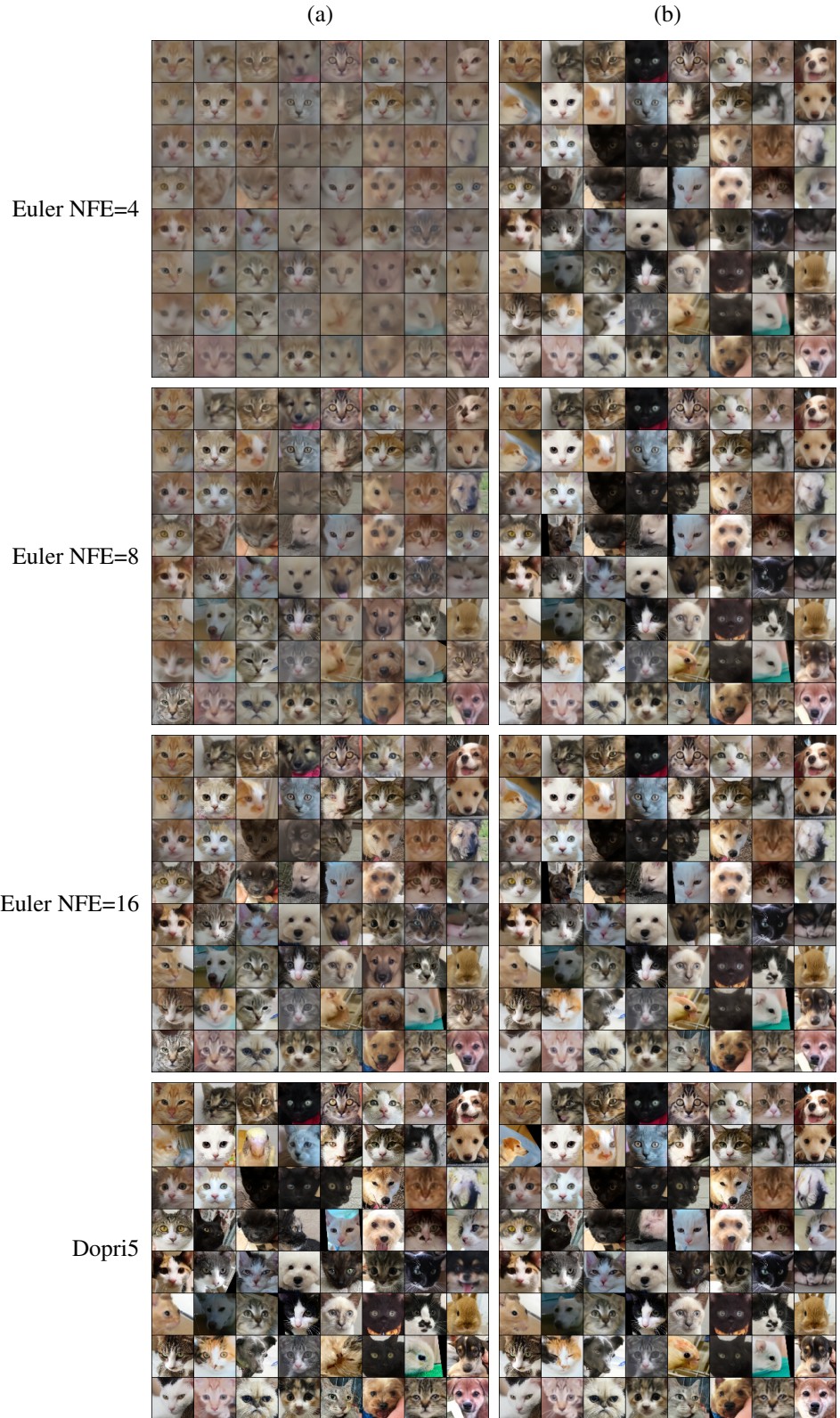

Euler NFE=4

Euler NFE=8

Euler NFE=16

Dopri5

Figure 13: Images generated from unconditional models trained on **PetFace (64x64)**.

(a) (b)

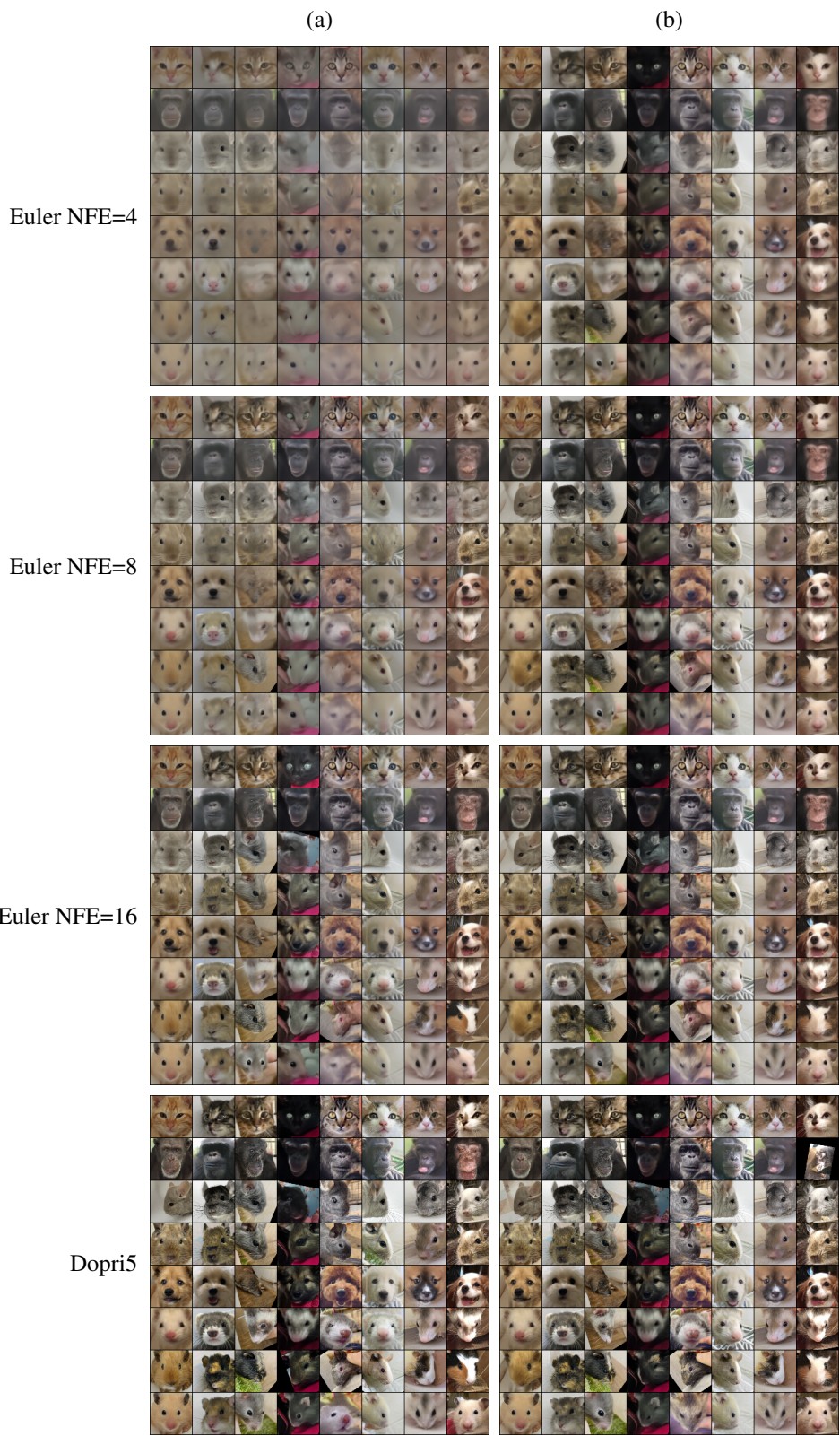

Euler NFE=4

Euler NFE=8

Euler NFE=16

Dopri5

Figure 14: Images generated from conditional models trained on **PetFace (64x64)**. Rows correspond to the following classes: cat, chimp, chinchilla, degus, dog, ferret, guineapig, hamster.

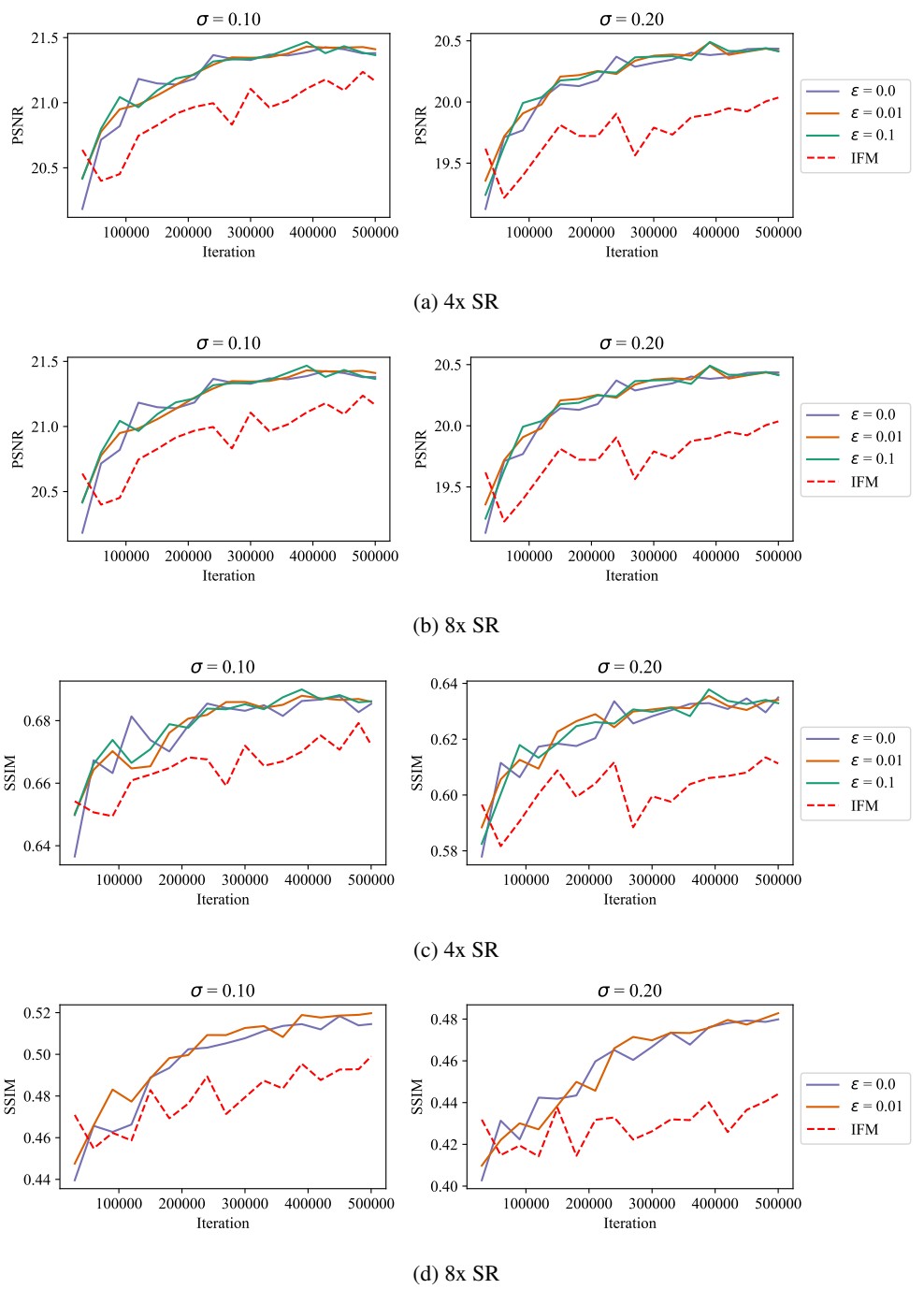

Figure 15: Evolution of **PSNR** and **SSIM** over the course of training for 4x and 8x image super-resolution on **CelebA**.

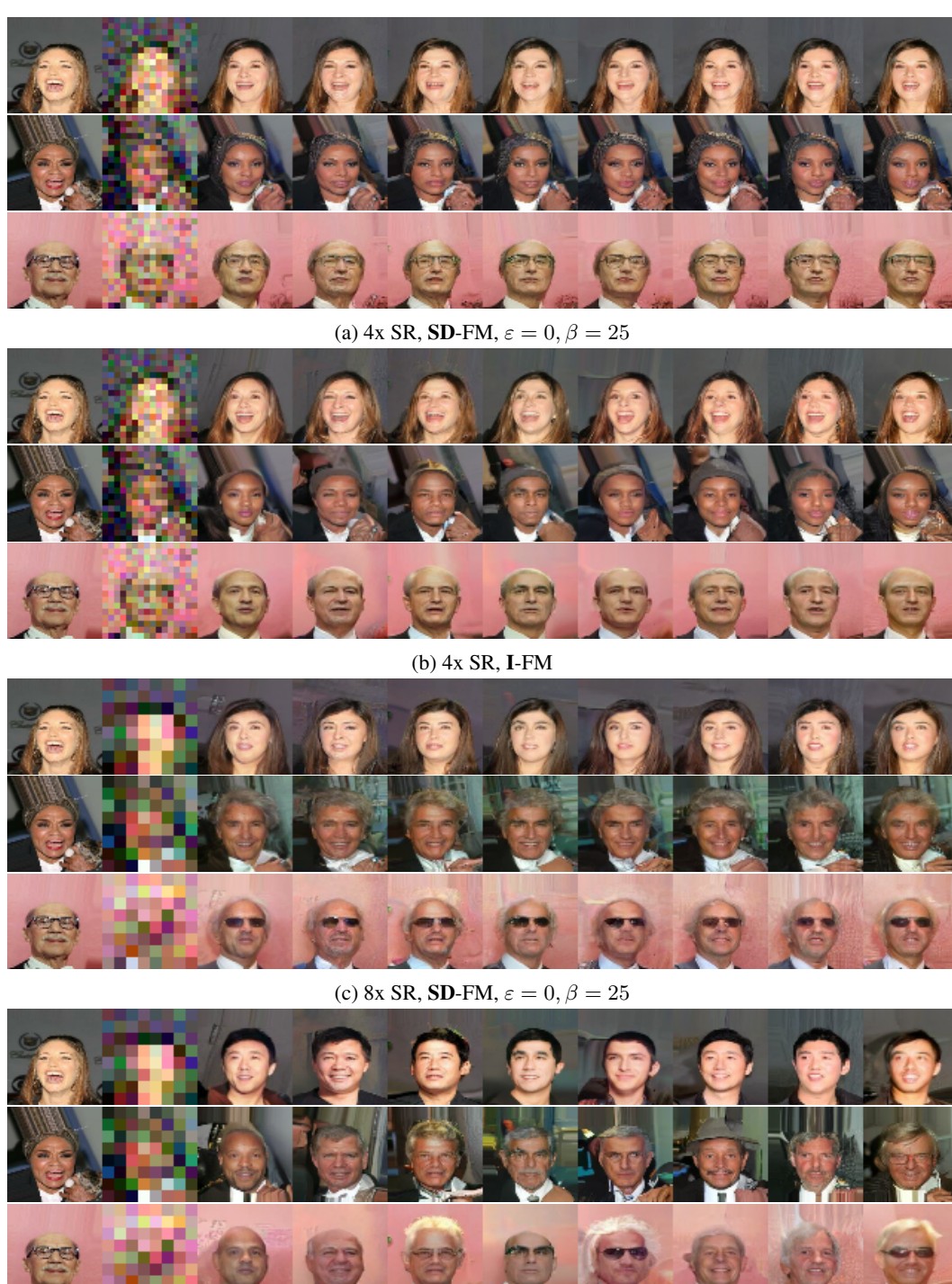

(a) 4x SR, **SD**-FM, $\varepsilon = 0, \beta = 25$

(b) 4x SR, **I**-FM

(c) 8x SR, **SD**-FM, $\varepsilon = 0, \beta = 25$

(d) 8x SR, **I**-FM

Figure 16: Sample images for **CelebA** 4x and 8x super-resolution with noise level $\sigma = 0.2$, for models trained with **SD-FM** and **I-FM**. Columns correspond to the original clean image, downscaled and noisy image, respectively, followed by 8 samples from the flow model.

## E  SENSITIVITY OF DUAL POTENTIAL TO $\varepsilon$

In 17, we provide a fine-grained picture of the impact of $\varepsilon$ when computing $\mathbf{g}$ on the same ImgN64 dataset (in full dimensions).

## F  SD-FM: ABLATIONS ON IMGN64

### F.1  DISTRIBUTION OF MIPS SAMPLING AS A FUNCTION OF $\tau$

Figure 18 proposes to study the distribution of selected images in the dataset of $N$ ImgN64 (full-dimension) as SD-FM training occurs. This pictures provides a complementary view to Fig. 3 showing why the FID metric improve as the $\chi_2$ metric decreases.

### F.2  SD-FM METRICS A FUNCTION OF $\tau, \varepsilon$ AND COST ABLATION

We simultaneously study in the following experiments three different factors that lie at the heart of SD-FM:

- does it matter that the potential $\mathbf{g}_\varepsilon^\star$ for a certain $\varepsilon$ is truly optimal, in the sense that it respects marginals and guarantees that every point in the dataset is properly sampled? In other words, does low $\tau$ always correlate with better FID and curvature metrics?

- does it matter for SD-FM to work to narrow down on the *optimal transport* as defined using the "traditional" $\ell_2$ cost $c(x, y) = -\langle x, y \rangle$? Would results be different if one were to select different costs? To answer this we consider three other costs: the "anti-OT" cost $c(x, y) = \langle x, y \rangle$ that promotes, on the contrary, the longest paths, and two random corruptions of the $c_\sigma(x, y) = -\langle \sigma \circ x, y \rangle$. The $\sigma$ holds the realizations of $d$ random (Rademacher) variable in $-1, 1$ with probability 30% or 70% of taking the value $-1$. This value is sampled only once per experiment.

- does it matter to have a deterministic association ($\varepsilon = 0$) or is it better to inject more randomness ($\varepsilon > 0$), having in mind that infinite $\varepsilon$ is equivalent to IFM.

While all of Figures 19, 20, 21, 22 provide a wide wealth of information, we can confidently say, by looking at all of them, that apart from FID in the lowest NFE-4 case, which is slightly more nuanced, all metrics improve as $\varepsilon$ and $\tau$ decrease (closer to "sharp" OT) and clearly highlight the crucial role of the $\ell_2$ cost (i.e. not "any" OT plan would work, that with shortest paths results in better performance).

### F.3  TRAINING SD-OT ON PIXEL SPACE VS. LATENT SPACE

See Figure 23 and its caption for a comparison between the evolution of the $\chi_2$ convergence criterion when computing SD-OT on either ImgN32 in original pixel space and ImgN256 in latent space.

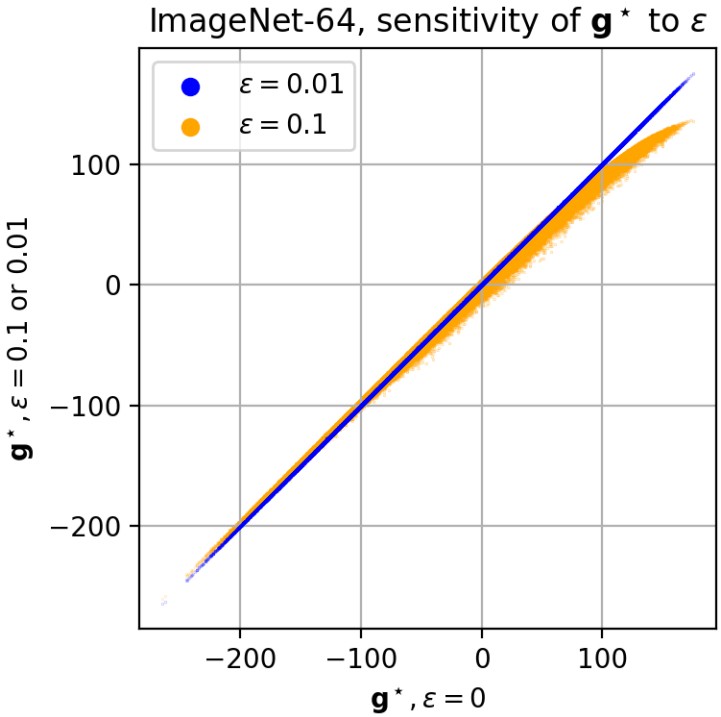

Figure 17: We compare the optimal dual potentials $\mathbf{g}_\varepsilon^\star$ for 3 different levels of $\varepsilon$ using scatter plots. Each scatter plot depicts 2.56M points, where 2D points $((g_\varepsilon^\star)_j, (g_{\varepsilon'}^\star)_j)$ are plotted for different $\varepsilon \neq \varepsilon'$. As can be shown, the potential vector $\mathbf{g}_\varepsilon^\star$ varies smoothly, for each datapoint, with $\varepsilon$. These computations are carried out on full-dimension $d = 12,288$ on ImgN64.

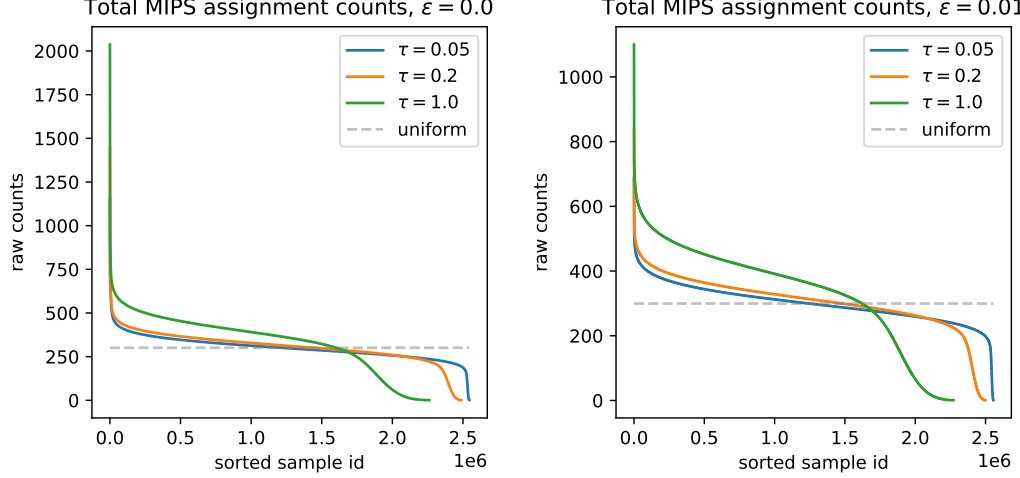

Figure 18: Effect of varying threshold $\tau$ (in Alg.1) when computing $\mathbf{g}_\varepsilon^\star$ on the distribution of selected images in the dataset, for $\varepsilon = 0.0$ on the left, $\varepsilon = 0.01$ right. As can be seen, a loose threshold of $\tau = 1$ (essentially no optimization for $\mathbf{g}$) results in images that are never sampled, whereas a finer threshold of $0.05$ equalizes the distribution of sampled images. These computations are carried out on PCA $k = 1000$ of the data space originally in full-dimension $d = 12,288$, on ImgN64.

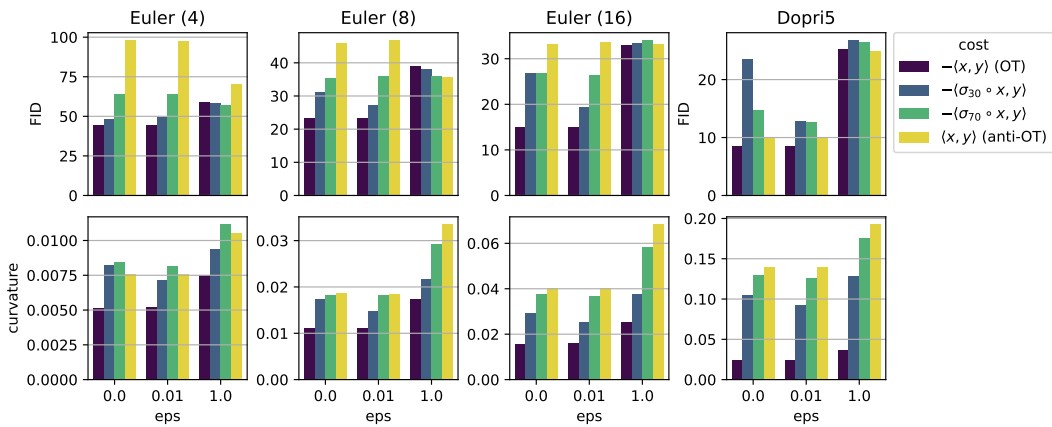

Figure 19: Low $\tau = 0.05$ regime (i.e. close to exact SD-OT computation).

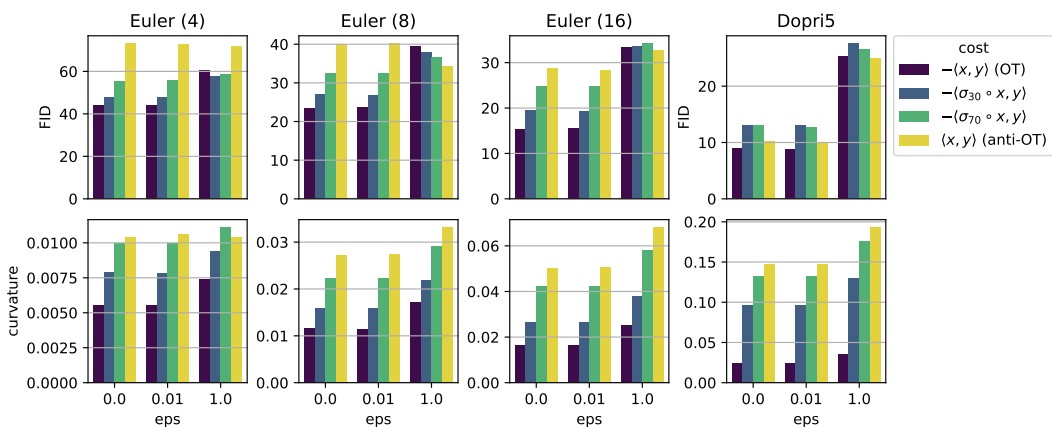

Figure 20: Middle $\tau = 0.2$ regime (i.e. exact SD-OT computation).

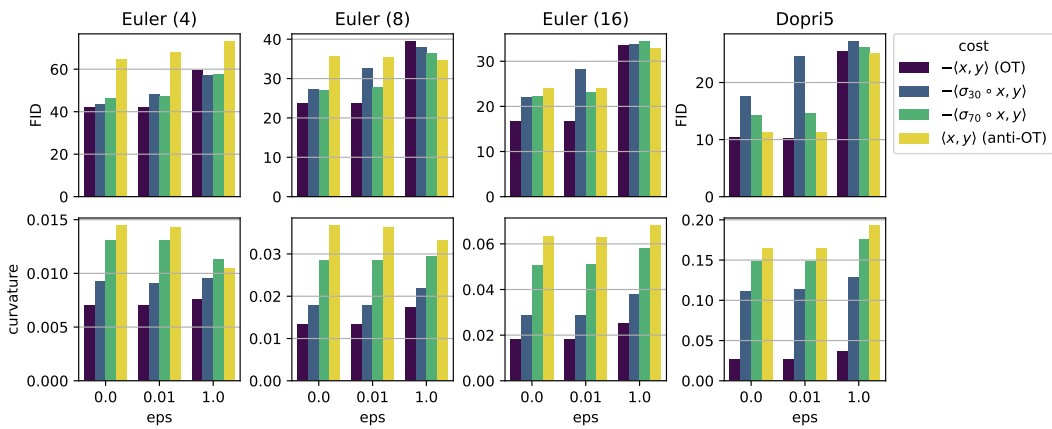

Figure 21: High $\tau = 1.0$ regime (no marginal preservation is guaranteed).

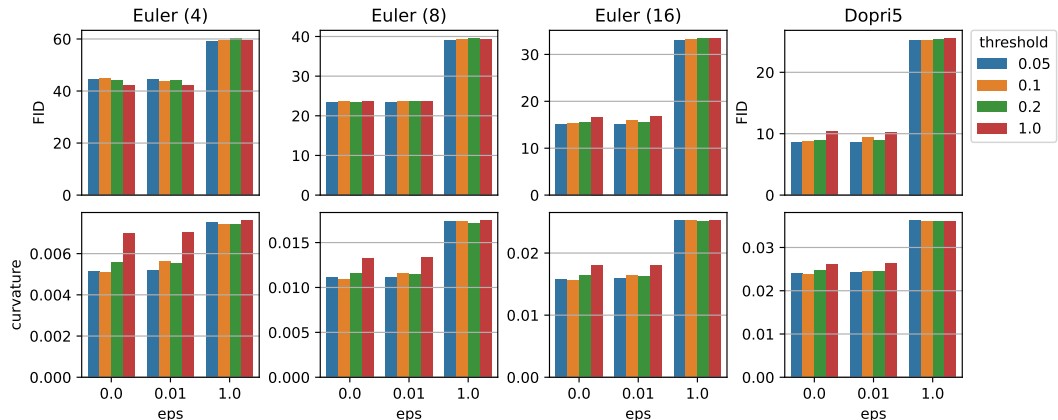

Figure 22: Original cost $c(x, y) = -\langle x, y \rangle$, low $\varepsilon = 0$ regularization, ablating $\tau$.

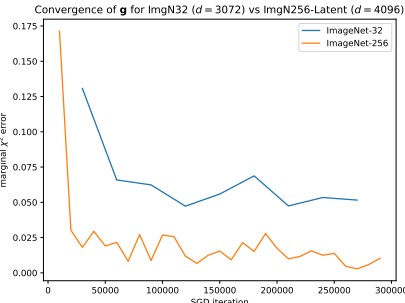

Figure 23: We compare the evolution of the $\chi_2$ convergence criterion when computing SD-OT on either ImgN32 in original pixel space ($d = 3072$) vs. ImgN256 in latent space ($d = 4096$). Here the total dataset size is the same ($N = 2.56$M). While one would expect SD-OT to converge faster for a smaller dimensional space (i.e. ImgN32), this is not the case, because of the far more regular distribution of latents that is designed to resemble a Gaussian.

