# OpenReview forum: "Flow Matching with Semidiscrete Couplings"
_ICLR.cc/2026/Conference — ICLR 2026 Poster_

### Official Review · Reviewer_WFDC · 2025-10-26

**Soundness:** 2
**Presentation:** 2
**Contribution:** 1
**Rating:** 2
**Confidence:** 3

**Summary:**

This paper introduces Semidiscrete Flow Matching (SD-FM), a novel approach to define the noise-to-data coupling in flow matching. The core idea is to perform a offline pre-computation to solve a global, semidiscrete Optimal Transport (OT) problem, which yields a dual potential vector g*. This vector g* implicitly defines a partition of the noise space, where each region in the partition corresponds to a unique data point. During the online FM training phase, each sampled noise vector is coupled to its corresponding data point by performing a Maximum Inner Product Search (MIPS) to identify its region. This "pre-compute then query" strategy is designed to improve the efficiency and scalability of OT-guided flow matching. The authors demonstrate significant performance gains on some benchmarks.

**Strengths:**

- Solid Theoretical Analysis: The paper provides a convergence analysis for the core algorithm, with a systematic comparison with existing optimal transport-based methods. This lends to theoretical importance to the proposed method.

- Clarity of Exposition: The paper is well-written and logically structured. Figure 1, in particular, is highly effective. It intuitively illustrates the core methodological differences between I-FM, OT-FM, and the proposed SD-FM, allowing readers to quickly grasp the paper's central contribution.

**Weaknesses:**

- Limited Experimental Scope: The method was only evaluated on two small-scale benchmarks. This is a significant omission, as the problem of computational inefficiency—which the paper claims to solve—is most critical on large-scale, high-dimensional datasets like ImageNet 256. The absence of such experiments makes it difficult to validate the method's scalability and practical utility.

- Limited Baseline Comparisons: The set of baseline methods is narrowly restricted to only two types of flow-matching models. The paper lacks a comparison against current state-of-the-art (SOTA) generative models in the related domain. To properly contextualize its contributions, the method must be benchmarked against SOTA approaches in terms of both generation quality and efficiency.

- Insufficient Analysis of the Key Hyperparameter ε: The vast majority of experiments are conducted with ε=0 (hard assignment). The parameter ε controls the stochasticity of the coupling (soft vs. hard assignment) and is highly likely to directly influence the diversity of the generated model. The paper lacks any investigation into how ε mediates the trade-off between generation quality (e.g., FID/precision) and diversity (e.g., recall).

**Questions:**

In table 2, the generation quality is better at $\epsilon = 0$, while some theoretical results are only valid when $\epsilon > 0$, e.g., Theorem 2, Proposition 3. Could you provide some explanation for this phenomenon?

---

> ### Author Response · Authors · 2025-11-18
> **Rebuttal to Reviewer WFDC 1/2**
>
> Many thanks for your encouraging comments on the "solid theoretical analysis" and clarity of our work. We are grateful for your time reading our work, and we hope this rebuttal will alleviate some of your concerns.
>
> > **Limited Experimental Scope: The method was only evaluated on two small-scale benchmarks.**
>
> Many thanks for your comment. We would like to clarify an important misunderstanding. Our experiments include 5 datasets: `ImageNet` at 32, 64 and 256 (latent) resolutions, `Petface` and `CelebA`. More precisely
>
> - `ImgN32` $N=2.56 M, d=32\times 32 \times 3 = 3,072$  (Sec. 5.1, 5.3), unconditional **and** class conditional
> - `ImgN64` $N=2.56 M, d=64\times 64\times 3 = 12,288$ (Sec. 5.1), unconditional **and** class conditional
> - `PetFace64` $N=1.27 M, d= 64\times 64\times 3 = 12,288$ (Sec. 5.1), unconditional **and** class conditional
> - `CelebA64` $N=162k, d = 64\times 64\times 3 = 12,288$ (Sec. 5.2), continuous conditional setup (super-resolution)
> - `ImgN256` $N=2.56 M, d=32\times 32 \times 4=4,096$  (Sec. 5.4), unconditional
>
> In total, this counts for **8 different benchmark tasks** where SD-OT couplings always yields better results overall. Following your comment, we added all dataset names in subsection headers (see revision).
>
> > **This is a significant omission, as the problem of computational inefficiency—which the paper claims to solve—is most critical on large-scale, high-dimensional datasets like ImageNet 256.[...]**
>
> Note that we *did include* `ImgN256` in our benchmarks in `S.5.4` following the established practice of using a latent space $d = 32\times 32\times 4=4096$ as done in MeanFlow by [Geng et al., 2025, oral @ Neurips'25]. We get excellent results in that experiment, showing that the simple switch to SD-OT couplings greatly improves performance at low NFEs.
>
> A few more comments on this point:
>
> - our raw-pixel experiments on `ImagN64/PetFace64/CelebA` are thus 3 times higher dimensional than those using `ImgN256` in the FM literature, e.g. MeanFlow.
>
> - we believe that computing flows in raw pixel space is harder than in latent space, but still useful to measure the quality of a flow method. It is harder because the manifold of data is sharp, whereas distributions in latent space resemble a Gaussian prior by design. We also think that SD-OT convergences *faster* in latent space. Following your remark, we added this in `S.5.4` and provide a preliminary evaluation in Fig. 23. While this is not an apple-to-apple comparison (we are comparing SDOT convergence on the "same" data (ImgNet images) but with 2 different representations, one in $d=3072$, the other $d=4096$), this simple result shows that it might be easier to learn an OT map from noise → latents than noise → raw data.
>
> - The reason we provide detailed ablations in raw pixel space is that we know that the flow community will look at them more carefully. When using latents, the good performance of a generative model hinges significantly on the quality of the encoder/decoder.  Because our goal is to isolate the contribution of SD-OT to flow methodology, we chose to consider both raw-pixel generation (with detailed ablations & visual confirmation) **and** latent generation.
>
> > **Limited Baseline Comparisons: The set of baseline methods is narrowly restricted to only two types of flow-matching models. The paper lacks a comparison against current state-of-the-art (SOTA) generative models in the related domain. To properly contextualize its contributions, the method must be benchmarked against SOTA approaches in terms of both generation quality and efficiency.**
>
> Flow matching is a core ML method. FM is **modular**, and our methodological contribution is to replace **one and one module** only, the noise/data coupling step. We explain this in L.11 and the introduction paragraph "Coupling Noise$\times$ data" in L.41. For noise/data coupling, the independent IFM and mini-batch OT (OTFM) couplings are the only competitors we are aware of, as highlighted with colors in Alg.1~5.
>
> To ensure that results are reported for our multitude of controlled experiments, we kept fixed *all* other aspects of flow model training, using default choices for e.g. the time-sampler, training schedule, network size, loss, etc. We ablated entirely all parameters introduced in SD-FM (see e.g. our extended results in Fig. 19~22) and showed a consistent and significant improvement upon other noise-data couplings.
>
> Any other change in the FM approach would be *complementary* to SD-OT. We did focus in `S.5.4` on a very recent SOTA approach, MeanFlow, and showed significant improvements. Of course, this still leaves room to consider a large mix of *any other* SOTA techniques, but that hypothesis space would be far too large to fit in an ICLR paper.
>
> If you have a specific SOTA approach in mind, it would be helpful to share one, **specifically**. We can try to implement it during the rebuttal, to the best of our ability and compute capacity.

---

> ### Author Response · Authors · 2025-11-18
> **Rebuttal to Reviewer WFDC 2/2**
>
> > **Insufficient Analysis of the Key Hyperparameter ε: The vast majority of experiments are conducted with ε=0 (hard assignment).**
>
> The choice of 𝜀 is considered in many experiments (Figures 2, 4, 7, 8, Table 2, 3). These experiments all point to better overall performance in terms of FID and curvature when using smaller 𝜀. Overall, we clarified in the conclusion that we advocate using $𝜀=0$ as a first attempt. Additionally, following your comment and that of Reviewers `48uQ` and `JxQX`, we have added a few ablations more that we found interesting in the new sections that extend significantly Figure 3 (Appendix E/F and below), featuring prominently 𝜀. Our recommendation remains unchanged, 𝜀=0 is likely the easiest choice.
>
> > **The parameter ε controls the stochasticity of the coupling (soft vs. hard assignment) and is highly likely to directly influence the diversity of the generated model. The paper lacks any investigation into how ε mediates the trade-off between generation quality (e.g., FID/precision) and diversity (e.g., recall).**
>
> We observe no qualitative or quantitative influence of 𝜀 on diversity (see e.g. images generated in appendix). In fact, diversity is very similar to that provided by I-FM across all experiments.
>
> Please also note that diversity is tracked by FID, which is a *distributional metric* that incorporates *both* recall and precision (see e.g. https://arxiv.org/pdf/1904.06991 for a detailed analysis): FID also penalizes a lack of diversity in generated samples: Indeed, if a model only generates a subset of the modes present in the real distribution (mode collapse), the covariance matrix of the generated features will differ significantly from the real distribution's covariance, increasing the FID score. This increase in FID would mean that the model fails to "recall" or cover all the variety in the real data. FID is fairly stable for various ε, and by looking at outputs, we have no reason to believe some ε values trade-off better recall in exchange for better precision.
>
> > **In table 2, the generation quality is better at 𝜀=0, while some theoretical results are only valid when 𝜀>0, e.g., Theorem 2, Proposition 3. Could you provide some explanation for this phenomenon?**
>
> **This is an unfortunate and crucial misunderstanding of our paper**. *All* our theoretical results cover explicitly the case $𝜀=0$, which is an important feature of our work. This is explained in:
> - the introduction L.91 ("[results] applicable to both the $𝜀=0$ and $𝜀>0$ cases").
> - Theorem 2 starts with "Suppose either 𝜀>0 or Assumption 1 is satisfied". There are two constants, "Let $L_𝜀\coloneqq 1/𝜀$ for $𝜀> 0$ and $L_0 \coloneqq C_{\max}/\delta$ else.". The analysis is then valid for any $𝜀\geq 0$.
> - In Proposition 3, similarly, there are two constants, either $\delta_𝜀$ for $𝜀>0$ of $\delta_0$. In fact, Proposition 3 demonstrates **an advantage of $𝜀= 0$,** because one recovers the usual Tweedie’s formula *despite the coupling not being independent*. This is *not* the case for $0<𝜀<\infty$, as discussed after Proposition 3.
>
> Following your comment and that of Reviewer `JxQX`, we have emphasized further the importance of obtainaing these results for 𝜀=0 in the introduction of Section 3 (see revision). We can underline this further if you feel this aspect is not stressed enough.

---

> > ### Comment · Reviewer_WFDC · 2025-11-26
> >
> > Thank you for addressing my questions. The clarifications on the experimental scope and the analysis of hyperparameter $\epsilon$ are appreciated.
> >
> > I have raised my score to 4 .
> >
> > To justify an increase to 6, please compare the model's performance against other relevant methods on the **ImageNet 256** dataset on the MeanFlow backbone.

---

> > > ### Author Response · Authors · 2025-11-27
> > > **Many thanks for your score increase**
> > >
> > > Again, we are grateful for your time reviewing our paper and especially for reading our rebuttal.
> > >
> > > > **Thank you for addressing my questions. The clarifications on the experimental scope and the analysis of hyperparameter $\epsilon$ are appreciated. I have raised my score to 4 .**
> > >
> > > We are happy to see that you have moved away from a strong reject stance by raising your score to a borderline reject. Of course, we remain available for any other questions or discussions until the end of the rebuttal period.
> > >
> > > > **To justify an increase to 6, please compare the model's performance against other relevant methods on the ImageNet 256 dataset on the MeanFlow backbone.**
> > >
> > > Many thanks for this suggestion.
> > >
> > > *Let us start by clarifying that Fig. 6, while arguably occupying little space in our draft, required significant compute.*
> > >
> > > We trained for that plot 3 seeds x 2 DiT XL/2 (650M) meanflow models (`SD-MF` and `I-MF`) using `DiT-XL/2`, already the **biggest** model considered in the MeanFlow paper (676M params). Each train run requires a 8-GPU H100 node for ~6 days, namely
> > >
> > > `3 seeds x 2 training procedures (baseline/ours) x 8 GPUs x 6 days x 24 hours ~7k GPU hours.`
> > >
> > > That plot displays a significant performance boost across many NFEs of MeanFlow when using `SD` vs `I` (the default approach used in FM and MF), which was achieved through a simple change in the dataloader step (use MIPS procedure given $\mathbf{g}^\star$ to match noise to data, instead of independent sampling).
> > >
> > > **In the time left before the discussion closes**, training a `DiT XL/2` model would not be feasible. Therefore, our plan is to recreate a similar curve (`FID vs NFE` for `ImgN256 latent space`) across other (smaller) model scales, with the same MeanFlow loss / training loop. We are running against a few obstacles (we no longer have access to a computational resource that is needed to launch new experiments) but we will keep you updated as we progress.
> > >
> > > Many thanks for your patience and for your interest in our work.
> > >
> > > The Authors

---

> ### Author Response · Authors · 2025-12-04
> **update before rebuttal window closes.**
>
> Dear Reviewer
>
> We have tried to carry out the experiments we suggested above (e.g. display gap with M2 and L2 models, in addition to only XL2 as we proposed first) but we ran into difficulties and they haven't finalized at the time of writing.
>
> **For now, the only result we are able to provide for now is a slightly undertrained M/2 model at 40 epochs (200k steps), and was added to the new Figure 6.** The gap between IMF and SDFM remains, though slighlty smaller at that model scale compared to XL-2. This plot will be updated at a later date with more epochs as we are able to sort out these issues.
>
> Some of the jobs we have been running take days, and some of the main contributors for this work no longer have access to our compute infra, which prevented the fast resolution of bugs that arose following changes in the packages we've kept improving since. This has slowed us down.
>
> **This being said, we are very confident to state that the results in our paper are very much representative of SOTA training of flow matching for ImgN-256.** MeanFlow is just being presented now as we speak as an oral at Neurips, and the XL2 model is the largest they consider. The gap that we observe are significant. We do not see any other SOTA approach worth including at the time of writing, and would like to ask you again specifically what SOTA approach you had in mind.
>
> Thanks again for your patience and for your comments.
>
> The authors

---

### Official Review · Reviewer_JxQX · 2025-10-26

**Soundness:** 3
**Presentation:** 2
**Contribution:** 3
**Rating:** 6
**Confidence:** 3

**Summary:**

To overcome the computational cost but fulfill the theoretical promises of OT-FM, this paper leverages the entropy regularized semidiscrete OT (SD-OT) and proposes a SD-OT coupling. In the coupling stage, the SD-OT problem is solved by estimating the dual potential $g$ by SGD; then in the FM training stage, noise samples are pared to data samples by maximum inner product search (MIPS, when $\epsilon = 0$) or SoftMax (when $\epsilon > 0$), using the learned $g$. They also provide the convergence analysis of SGD for SD-OT. The experiments using ImageNet, PetFace and CelebA show that the proposed SD-OT coupling is better than I-FM and OT-FM in main metrics.

**Strengths:**

The paper is well-motivated by practical drawbacks of OT-FM, and the general coupling method SD-FM can potentially integerate into many diffusion-/ flow-based models. The convergence analysis justify the case for both $\epsilon = 0$ and $\epsilon > 0$. The validity of unregularized $\epsilon = 0$ case avoids the choice of tuning parameter.

**Weaknesses:**

1. The paper mainly adapts SD-OT from Genevay et al., 2016 in FM setting, which may raise some novelty issues.
2. The scalabililty of SD-OT precompucate for large $N$ remains a concern.
3. In section 2 (Background and Related Work), are "$\ldots$" in subtites typos?

**Questions:**

1. Although $\epsilon = 0$ is valid, how sensitive are results to $\epsilon$? Also, what's the sensitivity for fitted $g$?

---

> ### Author Response · Authors · 2025-11-17
> **Rebuttal to Reviewer JxQX 1/1**
>
> Many thanks for your sharp review and for recognizing that a strong point of our approach is that our _"convergence analysis justify the case for both $𝜀= 0$ and $𝜀 > 0$"_, which is indeed a core message of our paper.
>
> More generally, we are thankful for your time, and welcome this opportunity to clarify some of the points you raised.
>
> > **The paper mainly adapts SD-OT from Genevay et al., 2016 in FM setting, which may raise some novelty issues.**
>
> While Genevay et al., 2016 is a seminal reference, we would like to clarify crucial differences that should alleviate this concern. These have been added to the revision, L. 193
>
> - Genevay et al. do not provide a convergence criteria to track progress of stochastic optimization (their Alg. 2 is a `for` loop). This is a weakness when $N$ is in the order of millions, in which one cannot simply set a predefined number $K$ of iterations and hope that the algorithm has converged, without tracking progress (L.214).
>
> - Without a useful convergence criterion, Genevay et al do not provide an analysis of the complexity of SGD (# of iterations) to reach a satisfactory *solution*, and do not even consider the case $𝜀=0$.
>
> - Solving these two aspects constitute an important contribution to SD-OT, that extend beyond SD-FM. (1) We proposed an unbiased $\chi_2$ estimator that characterizes convergence in **Eq.10**. The interest of $\chi_2$ is further showcased in  Fig. 2, in which we show that our $\chi_2$ convergence correlates positively with better metrics. (2) We provided a theoretical analysis that proves the convergence rate of $O(1/\sqrt{K})$ **even for the non-smooth case of $𝜀 = 0$.** (Theorem 2)
>
> - Genevay et al. is ~10 years old. Their computations were done on datasets where $N = 10$ and $d=3$. At the time, their end-goal was to approximate a Wasserstein distance. We demonstrate that at a scale of $N \approx 10^6$ and $d\approx10^4$ (a $10^8$ x larger scale overall), SGD for SDOT not only works and is reliable, but also translates into better results in the important downstream task of learning flows.
>
> Note also that our paper provides theoretical novelty in *other* areas, such as generalizing Tweedie’s formula to the OT-coupling regime (Proposition 3), which allows us to perform theoretically sound classifier-free guidance (Section 5.3), and demonstrate that the benefits of OT couplings can be extended to few step generative models such as MeanFlow (Section 5.4).
>
> > **The scalabililty of SD-OT precompute for large N remains a concern.**
>
> Based on our experience running SD-OT, we do not share these concerns, in the sense that SD-OT is both much simpler compared to OT (as a pre-processing task), and cheap compared to the FM training problem (SD-OT is a stochastic concave maximization problem that provably converges).
>
> Our results demonstrate that SD-OT and SD-FM can be applied successfully to high-dimensional datasets ($d=12,288$) of size $N=$2.56M (e.g. Fig. 8). For larger datasets, the memory cost of running SD-OT and then SD-FM grows linearly with dataset size $Nd$. Essentially, the bottleneck becomes simply whether the dataset can be held in memory.
>
> In scenarios where $N$ is truly much larger, we advocate either to use PCA to decrease $d$ to match noise/images, or more aggressively to split the dataset into mega-batches of size $\approx 10^7$ and compute a dual vector on each of them independently.
>
> > **In section 2 (Background and Related Work), are "" in subtites typos?**
>
> We use the ellipsis “...”  to emphasize that the problem of obtaining "Straighter flows..." can be achieved using two different philosophies, either “Straighter flows … using reflow” (post-processing / distillation) or “Straighter flows … using using pairs sampled from OT couplings”, i.e. a pre-processing approach.
>
> > **Although epsilon=0 is valid, how sensitive are results to epsilon? Also, what's the sensitivity for fitted g?**
>
> Thanks for this question. We are not sure to which results you are referring explicitly:
>
> If you are referring to **SD-OT** results and $\mathbf{g}^\star$'s sensitivity to 𝜀, Fig. 2 provides an accurate picture of SGD: small 𝜀 is a bit harder to train (requires more steps to get to a low $\chi_2$ threshold) than large 𝜀. In terms of variations of the fitted $\mathbf{g}$ to 𝜀, we now plot in **Fig. 17** in the revised appendix a *scatter* plot of $\mathbf{g}$ pre-computed for `ImgN64`, full-dimensional case. This shows stability, which is why we are exploring 𝜀 tempering (L.469)
>
> If you are referring to **SD-FM** sensitivity to 𝜀, Fig. 4, 7, 8, Table 2, 3 provide elements. Our broad conclusions are that small 𝜀 or $𝜀=0$ works slightly better, and MIPS used in $𝜀=0$ is arguably simplest. Therefore, we recommend to end-users trying $𝜀=0$ first, and only increasing $𝜀$ if necessary. **We invite you to look more broadly into Fig. 19, 20, 21, 22 in the appendix that fully characterize for `ImgN64` the impact of $\tau$, 𝜀 and cost.**

---

### Official Review · Reviewer_rkmY · 2025-10-30

**Soundness:** 3
**Presentation:** 4
**Contribution:** 3
**Rating:** 6
**Confidence:** 3

**Summary:**

This paper proposes Semidiscrete Flow Matching (SD-FM), a novel approach that addresses the computational bottlenecks of Optimal Transport Flow Matching (OT-FM). By reformulating the noise-data pairing problem as a semidiscrete optimal transport problem from a continuous noise distribution to a discrete dataset, SD-FM fits a dual potential vector via SGD during a precomputation phase. This allows each newly sampled noise vector to be matched to a data point at training time through a simple maximum inner product search, effectively eliminating the quadratic dependency on batch size that plagues batch-OT methods. Extensive experiments demonstrate that SD-FM outperforms both standard FM and OT-FM across various datasets and training settings, achieving superior results in unconditional/conditional generation tasks under all tested inference budget constraints.

**Strengths:**

- The paper introduces a fundamentally different approach to optimal transport in flow matching by leveraging the semidiscrete formulation, effectively circumventing the quadratic complexity that has limited practical adoption of OT-FM methods.
- Beyond the algorithmic contribution, the work provides thorough theoretical analysis including convergence guarantees for both regularized and unregularized settings and extends the Tweedie formula to the semidiscrete case, demonstrating strong mathematical rigor.
- The experimental design covers diverse scenarios including unconditional/conditional generation, super-resolution, and mean-flow models, with consistent improvements across different inference budgets, particularly notable in low-NFE regimes.
- The decoupling of precomputation from training, combined with efficient MIPS-based matching, offers a viable path for scaling OT-based methods to larger datasets while maintaining performance benefits.

**Weaknesses:**

- While the 12-hour precomputation on 2.56M samples is reasonable, the paper lacks systematic analysis of how the method scales to modern large-scale datasets (e.g., >10M samples). The O(N) training cost, though better than quadratic, may still become prohibitive for massive datasets.
- The comparison framework omits important contemporary approaches for improving flow straightness (e.g., Reflow, MinibatchOT variants), making it difficult to assess SD-FM's relative advantages in the broader landscape of efficient flow matching methods.
- The paper provides limited insights into hyperparameter selection, particularly regarding the regularization parameter ε. Practical recommendations for choosing ε based on dataset characteristics (dimensionality, size, noise levels) would significantly enhance reproducibility and usability.
- The experimental validation focuses primarily on lower-resolution datasets, leaving open questions about the method's effectiveness on high-resolution generation tasks (e.g., 256×256 and above) that are increasingly relevant in practical applications.

**Questions:**

See the "Weakness" section.

---

> ### Author Response · Authors · 2025-11-17
> **Rebuttal to Reviewer rkmY 1/2**
>
> Many thanks for your encouraging review stating that our paper introduces a fundamentally different approach to flow matching. We are grateful for your comments (“rigorous theoretical analysis”, “diverse [experimental] scenarios"), and more generally for your time reading this work and our comments below. We welcome this opportunity to clarify a few aspects of our work.
>
> > **While the 12-hour precomputation on 2.56M samples is reasonable, the paper lacks systematic analysis of how the method scales to modern large-scale datasets (e.g., \>10M samples). The O(N) training cost, though better than quadratic, may still become prohibitive for massive datasets.**
>
> We understand this concern, but we argue that our experiments already show *feasibility* of SD-FM approaches at the scale of 10M that you mention.
>
> We state this because Fig. 8 (top row) shows SD-FM results on `ImgN64` where the SD-OT computation and pairing are both done in full pixel-space dimensions ($d=64 \times 64 \times 3 = 12,288$), with $N=2.56M$ images. Whether ran for SD-OT training or SD-FM pairing time, the scale that matters for our approaches is $Nd$ (the total dataset size). To increase $N$, one could e.g. use a PCA projection (onto $k=3,000$ rather than $d=12k$) and use that 4X decrease in memory to increase $N$ accordingly to $N=10M$ (if such a dataset was available). Using e.g.  $k=500$ (as used elsewhere in the paper), $N=$50M would remain feasible. We found consistently that OT with PCA-reduced data (Figs 7, 8) has limited effect on generation quality, since PCA is only used for pairings and generation is always done in pixel space.
>
> For these reasons, we are confident that the 10M scale poses no problem at all to our implementation as it is now. Fundamentally, this is due to the fact that SD-OT relies on simple blocks that can be scaled up in standard ways: stochastic concave maximisation to compute $\mathbf{g}$ and then MIPS search (or softmax sampling). This all scales linearly with $K$ (# of SGD iterations), $N$ (dataset size), $d$ (data dimension), e.g. $KNd$ when computing $\mathbf{g}$. This is in stark contrast to previous attempts that scale in $dn^2$ with batch-size $n$ (see Table 1).
>
> For much larger $N$, we would advocate using mega-batches of a few millions points (see also our response to Reviewer *48uQ*), and solve SD-OT *independently* between $\mathcal{N}(0, I)$ and these batches. This would not solve OT exactly (unlike full-dataset SD-FM) but would remain orders of magnitude faster / better / easier than minibatch-OTFM. FM training would be then done one mega-batch at a time.
>
> > **The comparison framework omits important contemporary approaches for improving flow straightness (e.g., Reflow, MinibatchOT variants), making it difficult to assess SD-FM's relative advantages in the broader landscape of efficient flow matching methods.**
>
> Regarding `Reflow`: We mention `Reflow` in L.133, then L.142, and L.475. To repeat two important points: `Reflow` is a **post-training/distillation** procedure that straightens an existing pre-trained *high-quality* flow model, by training a sequence of “rectified” models that generate {what the model has already pre-learned to generate} using straighter paths. As a result, Reflow can only work if the pre-trained model covers already very well the target distributions. In contrast, `SD-FM` is a **pre-processing** approach on **a specific dataset** to **pre-train** better a model from scratch.
>
> As a result, we believe `SD-FM` and `Reflow` are *complementary*, yet *incomparable*. In that sense, we wish to reuse the argument made recently by the MeanFlow paper (that do not compare to reflow)_**"Notably, our method is self-contained and trained entirely from scratch."**_. Our goal is also to provide directly such a good model, without distillation. While promising, we felt that experiments combining both `Reflow` and `SD-FM` would be far more complex to handle, muddle our message, and would require even more compute.
>
> Regarding `minibatch-OT variants`: What variants would you refer to? Zhang et al., (2025) showed recently that the benefits of minibatch-OT only start panning out for fairly large batch sizes $n$ ($n\approx 10^5$). At the moment, only the Sinkhorn algorithm has been shown to solve these large "sharp" (close to Hungarian) matching problems. Our study reproduces that setting, and shows a clear superiority of SD-FM. More generally, unlike mini-batch OT methods, SD-FM relies on a strong theoretical foundation that states that SD-OT can be solved with sufficient effort in the SGD phase (Algorithm 1, Theorem 2).

---

> ### Author Response · Authors · 2025-11-17
> **Rebuttal to Reviewer rkmY 2/2**
>
> > **The paper provides limited insights into hyperparameter selection, particularly regarding the regularization parameter ε.**
>
> Thanks for this question. Overall, our recommendation is simple: use $\varepsilon=0$. This was our conclusion in L. 467, but we agree it was not clear enough, and have clarified this in our revision.
>
> Why? For now, most results point to $\varepsilon=0$ or $\approx 0$ as the best choice. Zhang et al. (2025) also observed that small $\varepsilon$ yielded better results for OT-FM. In the case of SD-FM, all of the empirical evidence we have shows that performance increases or stays flat when setting $\varepsilon=0$, one get a slightly faster selection of pairs in return through "hard" MIPS. All of this just requires slightly more effort on the SD-OT resolution side (see Fig. 2) to get a similar $\chi_2$ threshold.
>
> Note that setting $\varepsilon$ to zero is only possible thanks to *our* use of the SD formulation of OT, since Sinkhorn’s algorithm requires a strictly positive $\varepsilon$ and blows up in complexity with $\varepsilon\approx 0$. Indeed, we are the first to propose that choice, supported theoretically (Theorem 2) and empirically, across a wide array of experiments.
>
>
> > **Practical recommendations for choosing ε based on dataset characteristics (dimensionality, size, noise levels) would significantly enhance reproducibility and usability.**
>
> Note that results ablating $\varepsilon = 0, 0.01, 0.1$ in Fig. 4 & 5 define $\varepsilon$ as a **relative** quantity selected with respect to a reference scale (here the standard deviation) of the cost matrix. We use Zhang et al. 25’s method as quickly mentioned in L271. In doing so, aspects of the dataset such as dimensionality, noise level, and natural scale are automatically accounted for.
>
> > **The experimental validation focuses primarily on lower-resolution datasets, leaving open questions about the method's effectiveness on high-resolution generation tasks (e.g., 256×256 and above) that are increasingly relevant in practical applications.**
>
> We believe there are two misunderstandings here:
>
> - We provide `ImgN256` results in Section 5.4 using, as done by the MeanFlow paper, latent representation, in dimension $d=32\times 32\times 4=4,096$.
> - The dimensionality of lower resolution versions of Imagenet, `ImgN32` $d=32\times 32\times 3=3,072$ or `ImgN64` $d=64\times 64\times 3=12,288$ is comparable or much bigger than latent space.
>
> To our knowledge, the pixel-space dimension we consider ($d=64 \times 64 \times 3 = 12288$) is the highest considered in the OT flow matching literature.
>
> We prioritised pixel space generation because it is usually seen as more challenging to study flows in raw data space. In that setting, what only matters is the flow performance, not the quality of the encoder/decoder. We kindly refer the reviewer to our response to **48uQ** where we provide more discussion on this.
>
> We would also like to highlight that training in latent space provides additional benefits for SDOT convergence as auto-encoder latents are trained to resemble a Gaussian prior. We now explicitly point this out in Section 5.4

---

### Official Review · Reviewer_48uQ · 2025-10-31

**Soundness:** 3
**Presentation:** 4
**Contribution:** 3
**Rating:** 6
**Confidence:** 3

**Summary:**

This paper proposes Semidiscrete Flow Matching to tackle the training computational cost of OT-FM. It studies a semidiscrete OT problem from a continuous noise to a discrete data distribution. This is solved in two stages: a one-time precomputation of a dual potential vector, followed by a MIPS lookup during training to pair noise with data. Theoretical convergence analysis and empirical results show SD-FM beats I-FM and matches OT-FM at a fraction of the computational cost.

**Strengths:**

1. The core idea of reframing OT-FM as a continuous-to-dataset semidiscrete problem is novel and practically efficient. The associated convergence criterion is also a new contribution.
2. The paper is well-written. The motivation is clear. Figures and tables effectively communicate the method and its advantages.

**Weaknesses:**

1. The paper alludes to fast approximate MIPS but does not investigate the trade-off between approximation error and final model quality, measured by FID. For large $N$, what is the impact of the approximation error from MIPS?
2. The paper lacks validation on high-dimensional data. Current diffusion models often denoise in latent spaces, which have high data dimensions.

**Questions:**

1. Figure 5 presents a nice analysis of diversity control through guidance samples. Could you provide more extensive analysis on how different parameters affect the sample diversity?
2. When applied to datasets of billions of samples, what is the additional memory overhead of SD-FM compared to I-FM?

---

> ### Author Response · Authors · 2025-11-17
> **Rebuttal to Reviewer 48uQ 1/2**
>
> Many thanks for your encouraging and kind comments on novelty / presentation. We are grateful for your time reading our paper & response. We welcome this opportunity to clarify a few important aspects of our contribution.
>
> > **Propose [....] to tackle the training computational cost of OT-FM**
>
> Note that OT-FM cost is not paid at training time, but at preprocessing time. We did our best to explain these trade-offs in Table 1 and Alg. 1~6.
>
> > **The paper alludes to fast approximate MIPS but does not investigate the trade-off between approximation error and final model quality, measured by FID. For large N, what is the impact of the approximation error from MIPS?**
>
> Thanks for this opportunity to clarify a few MIPS related-aspects in the paper.
>
> - All our experiments use **exact** MIPS and this is accounted for in all our time evaluations (`x`-axis in Fig. 4,7,8) when using $\varepsilon=0$ (purple circles). When $\varepsilon>0$, that time is the cost to sample from a softmax of size $N$.
>
> - MIPS means *maximum inner product search* (L.25), which is by default an exact lookup. While more engineering effort *could* result in faster / approximate MIPS using precomputed data structures, for now, early attempts to use, e.g. `jax.lax.approx_top_k`, were not any faster on multi-GPU nodes. This retrieval step remains, however, very cheap compared to backprop (Fig. 4 & 5), and can be easily mitigated (using e.g. PCA-reduced data for MIPS, and sharding) to reach larger scales.
>
> We realise that the term “MIPS” could be misinterpreted as being approximate, and we have clarified this in our current revision, please see lines 286-287 and addition to abstract.
>
> > **The paper lacks validation on high-dimensional data. Current diffusion models often denoise in latent spaces, which have high data dimensions.**
>
> Many thanks for this opportunity to clarify that:
>
> - We *did* provide results in latent space, using exactly the same setup as in the MeanFlows paper (oral @ Neurips25); Our application of the SD coupling to MeanFlow in **Sec. 5.4 and Fig. 6** uses their latent space $d = 32 \times 32 \times 4 = 4096$ for `ImgNet256`. We use their default choices, ablated for unconditional generation in *their* Table 1. We also use their larger model `XL/2`.
>
> - While they reported FID 61.06 for their `B/4` model (Table 1), we report in Fig. 6 a 1-NFE FID of approx. 49 for `XL/2` trained with *independent* coupling (part of their methodology), which drops to approx. 44 when using our coupling approach *SD-OT* for 1-NFE. That gap between SD coupling and independent coupling remains until 8-NFE. We did not evaluate further as mean-flow focuses primarily on low-NFEs (they only show 1 and 2 in their paper).
>
> - Latent spaces are usually *much smaller* than pixel-space. The `ImgN256` latent space dimension of 4k is sandwiched between that of `ImgN32` raw pixels ($d=3,072$) and `ImgN64` / `PetFace64` raw pixels ($d=64 \times 64 \times 3 = 12,288$). We would argue that running OT in latent space dimensions is usually easier: encoders are trained so that data distribution is approximately Gaussian, unlike the sharp manifolds of images described in pixel space.
>
> - The reason we believe pixel space results *also* matter is that latent space generation, while important to obtain better downstream performance, tends to conflate things: the quality of images generated using latent spaces relies mechanically on the encoder/decoder quality, polluting flow accuracy with the ability of the decoder to reconstruct images. Since our paper provides new flow training  *core methodologies*, we chose to follow previous references and do ablations in pixel space.
>
> > **Figure 5 presents a nice analysis of diversity control through guidance samples. Could you provide more extensive analysis on how different parameters affect the sample diversity?**
>
> Thanks for this suggestion. In the context of CFG, two parameters that control the diversity are $\gamma$, the strength of guidance, and $r$, the number of guidance samples used. Increasing $\gamma$ leads to a mode seeking behavior that improves quality (precision) but reduces sample diversity (recall). A priori, the role of $r$ is less clear since higher $r$ simply means more accurately sampling from the geometric mixture. As can be seen from Figure 5, higher $r$ also improves quality at the expense of lower diversity, and it can be seen as a form of picking the “best” sample from $r$ choices. We remark that most prior work consider the case of $r=1$, and providing a theoretically sound way of sampling from the geometric mixture with flow matching *under OT couplings* is a key contribution of our work, which allows better exploring the tradeoff between quality and diversity.

---

> ### Author Response · Authors · 2025-11-17
> **Rebuttal to Reviewer 48uQ 2/2**
>
> > **When applied to datasets of billions of samples, what is the additional memory overhead of SD-FM compared to I-FM?**
>
> The natural bottleneck is that SD-FM in its naive implementation will require storing the dataset in memory.
>
> This means a storage of $Nd$ (or $Nk$ when using $k$ principal components, as we advocate for very high-dimensional datasets), in addition to the dual vector $\mathbf{g}$ (negligible, itself of size $N$).
>
> In our experiments with ~2.5M samples, these operations resulted in very modest overheads, that were orders-of-magnitude cheaper than backprop cost as shown in Fig. 4,5 (this is the "$+ dN$" shown in red in the last line, righmost column of Table 1).
>
> With our current implementation (if we had such a dataset), we could easily go to 30M points in reduced PCA dimension $k=1000$, as evidenced by the fact that we ran `ImgNet64` in full pixel dimensions in Fig. 8, i.e. $N$=2.56M points in $d=12288$.
>
> For *billions* of samples, storing all points in memory (or a projection) of them would indeed become impractical. In that case we would advocate a mixture of known larger scale fixes, e.g. using *mega*-batches of the order of $\approx 10^7$ samples. For the $i$th mega-batch, we would have to pre-compute a SD-OT dual potential $\\mathbf{g}^{(i)}$ corresponding to that batch. We would do the flow training one mega-batch at a time.

---

> > ### Comment · Reviewer_48uQ · 2025-11-26
> >
> > Thank you for the clarifications. I am raising my score to 8.

---

> > > ### Author Response · Authors · 2025-11-27
> > > **Many thanks for your kind score increase.**
> > >
> > > Many thanks for your time, your comments and for your insights on the paper. We are grateful for your score increase. We remain available during the rebuttal period should you have any other questions.
> > >
> > > The Authors

---

### Author Response · Authors · 2025-11-21
**General response to Reviewers and AC**

We thank all reviewers for their time and feedback. We thank the AC and SAC for handling our paper.

**Here is a summary of the salient points we addressed in the rebuttal:**

* **SOTA methods were already present in our initial draft**: Our contribution is on improving the *noise-data coupling* component of flow matching (`L.41`) .
  *  We already compared at great lengths with the **SOTA** method of providing such couplings (using Sinkhorn on large batches of $n\approx 500k$). We provide timing results that were not available in [Zhang et al,25], and show that our algorithm clearly dominate OTFM in both compute cost (time per pair, `x-axis`) and performance (FID, `y-axis`), in `Fig. 4, 7, 8`.
  * Additionally, we had already shown that SD-OT couplings can be added to, and improve, a most recent **SOTA** approach (MeanFlow, Oral @ Neurips 25) in low-NFE unconditional generation in `Fig. 6`.

* **Dimensionality of datasets covered in experiments**: Our experiments covered datasets as large as ImgN-64x64 in pixel space ($d=64 \times 64 \times 3 = 12,288$), and ImgN-256x256 in latent space ($d=32 \times 32 \times 4 = 4096$). While one might be tempted to discount ImgN-64 as low-resolution, it has, in fact, *higher dimension* than latent space ImgN-256 used e.g. in MeanFlow. We focus on *both* pixel and latent space generation because pixel space is known to be more challenging. Using pixel space, we can evaluate progress on the FM training pipeline, ablating the role of the encoder/decoder. We do of course advocate using latent space in for large data spaces, and, remarkably, we believe SD-OT converges faster in latent-space (thanks to the Gaussian-like structure of the prior), as illustrated in a new Figure 23.

* **Ablating the role of 𝜀**: Many of our results already ablated 𝜀. They point to the benefit of choosing 𝜀 as small as possible, and we recommend end-users to choose $𝜀=0$ as default in `L. 471`. We have added a series of parameter wide ablations ($𝜀, 𝜏$, cost) in `Fig. 19~22` for ImgN-64 that confirm the superiority of using a SDOT coupling that is sharp (𝜀=0), recovers data marginal (low 𝜏) and is optimal w.r.t squared Euclidean cost, for better FID / curvature.

* **Scalability to large dataset size $N$**: Our current implementation handled  $N$=2.5M in $d=12k$. $N$ can be increased further using PCA to reduce dimension. Specifically, the cost of computing $\mathbf{g}$ scales linearly with $N$ (dataset size), $K$ (# of SGD iterations), and $d$ (dataset dimension) as $KNd$. Any reduction in $d$ via PCA unlocks the ability of using large $N$. Note that this comes at no cost to the performance of the flow model, which is still trained in the full-dimensional space. For very large datasets, we recommend splitting them into *mega-batches* of a size that would depend on dimension (could be e.g. $10\sim 50M$) and compute the dual vector for each mega-batch independently.

**Additions  to our revision include the following items:**

* **Positioning of our theoretical contributions on SD-OT** `L.192-197` clarify the novelty of our results relative to Genevay et al., 2016, highlighting the importance of the unbiased convergence criterion and the theoretical analysis of SGD.
* **Clarified that MIPS is exact when using 𝜀 = 0**. `L.286 `. In that case, the task of assigning noise to data amounts to an `argmax` lookup. We clarify that our results all used _exact_ lookup. Approximations could bring this to sublinear time but are left for future work.
* **Clarification of datasets for Experiments `Section 5`** Dataset names are included in subsection headings, emphasizing that our experimental results cover unconditional and conditional generation, superresolution, guidance, and one-step generation, and a range of datasets and resolutions up to 64x64 in pixel space and 256x256 in latent space.
* **Recommendation on the choice of 𝜀** In `L.470-472` we are now more upfront about our recommendation to pick $𝜀 = 0$ by default since this effectively eliminates the hyperparameter and is equivalent to solving for exact semidiscrete OT.
* **Additional experimental results** To answer points raised by reviewers, we have added new experiments and ablations to the Appendix (`L.1782` and after).
  * `Fig 17`, we show the stability of the optimal dual potential $\mathbf{g}_𝜀^\\star$ to the choice of $𝜀 \geq 0$ in the case of ImgN64. These results support our earlier conclusions, namely that $𝜀 = 0, 0.01$ differ minimally, while $𝜀 = 0.1$ has a mild impact on the solution.
  * `Fig 18`, we provide an additional illustration on how the selection frequency of images from the ImageNet64 ($N$ = 2.5M) changes as we solve OT. This shows that images are sampled more and more evenly as we get closer to the OT solution.
  * `Figs 19, 20, 21, 22`, we show ablations on (i) the convergence criterion 𝜏 for solving SD-OT, (ii) the choice of cost, and (iii) the choice of 𝜀.

---

### Meta-Review · Area_Chair_AkB8 · 2025-12-25

**Summary:**

This paper introduces a novel semi-discrete optimal transport (OT) coupling method for flow matching. It estimates a dual potential vector, which is then used to sample coupled noise-data pairs at a cost linear to the data size.

Reviewers liked the novel semi-discrete approach, the clear and well-motivated writing, and the theoretical contributions, including convergence analysis and a generalization of the Tweedie formula.

The primary criticism was the lack of validation with high-dimensional or large datasets.

Despite this, the rebuttal successfully addressed the reviewers' concerns, leading one reviewer to raise their score from 6 to 8 and another from 2 to 4. Overall, the paper is deemed strong enough for acceptance at ICLR.

**Reviewer Concerns:**

Please see above.

**Reviewer Scores:**

Please see above.

---

### Decision · Program_Chairs · 2026-01-26

Accept (Poster)